# Single-cell T-cell receptor repertoire profiling in dogs
My H. Hoang [1,2,7], Zachary L. Skidmore[1,2,7], Hans Rindt[3,7], Shirley Chu[1,3], Bryan Fisk[1,2], Jennifer A. Foltz[1], Catrina Fronick[2], Robert Fulton[2], Mingyi Zhou[4], Nathan J. Bivens [4], Carol N. Reinero [3], Todd A. Fehniger[1,5], Malachi Griffith [1,2,5,6], Jeffrey N. Bryan [3] ✉ & Obi L. Griffith [1,2,5,6] ✉

Spontaneous cancers in companion dogs are robust models of human disease. Tracking tumor-specific immune responses in these models requires reagents to perform species-specific single cell T cell receptor sequencing (scTCRseq). scTCRseq and integration with scRNA data have not been demonstrated on companion dogs with cancer. Here, five healthy dogs, two dogs with T cell lymphoma and four dogs with melanoma are selected to demonstrate applicability of scTCRseq in a cancer immunotherapy setting. Single-cell suspensions of PBMCs or lymph node aspirates are profiled using scRNA and dog-specific scTCRseq primers. In total, 77,809 V(D)J-expressing cells are detected, with an average of 3498 (348 - 5,971) unique clonotypes identified per sample. In total, 29/34, 40/40, 22/22 and 9/9 known functional TRAV, TRAJ, TRBV and TRBJ gene segments are observed respectively. Pseudogene or otherwise defective gene segments are also detected supporting re-annotation of several as functional. Healthy dogs exhibit highly diverse repertoires, T cell lymphomas exhibit clonal repertoires, and vaccine-treated melanoma dogs are dominated by a small number of highly abundant clonotypes. scRNA libraries define large clusters of V(D)J-expressing CD8+ and CD4 + T cells. Dominant clonotypes observed in melanoma PBMCs are predominantly CD8 + T cells, with activated phenotypes, suggesting possible anti-tumor T cell populations.

The biomedical research community, as well as the National Cancer Institute, have recognized the value of companion dog spontaneous models of cancer with release of several targeted funding programs. This recognition has led to the completion of translational clinical trials in companion dogs studying osteosarcoma[1–3], lymphoma[4], glioma[5–7], melanoma[3], squamous cell carcinoma, and soft tissue sarcoma[8]. Recently, immunotherapy trials evaluated a human chimeric HER2 Listeria vaccine in osteosarcoma as well as an antibody-linked IL-12 conjugate in melanoma of companion dogs[9]. The outbred genetic variability, body size, cancer-conditioned immune system, and shared environment of companion dogs and humans have been cited as particular advantages of dogs as pre-clinical or interposed post-clinical subjects in veterinary clinical trials of interventions destined for human application[10].

Immunotherapy is a current focus in the treatment of cancer with the rise of checkpoint blockade inhibitors, personalized cancer vaccines, CAR

T-cell therapies, and more. As a result, profiling the T cell receptor (TCR) repertoire has become an important tool for understanding, measuring, tracking and even predicting T cell mediated immune responses in a cancer setting[11]. The ability to accurately profile the TCR repertoire can also provide a powerful means of diagnosing and tracking T cell malignancies (e.g., minimal residual disease monitoring). Finally, TCR profiling may play an important role in understanding and treating a host of other immune-related (e.g., autoimmune) diseases.

It has also become apparent in recent literature that the tumor microenvironment (TME) plays a broad and complex role in tumorigenesis and response to many cancer therapies[12]. In particular the TME is known to be capable of producing an immunosuppressive effect. For example the presence of cytokines such as IL-6, IL-12, and TGFβ can increase PD-L1 expression allowing immune escape of tumor cells via checkpoint pathways[13]. The ability to examine the T cell repertoire, in the context of the

[1]Division of Oncology, Department of Medicine, Washington University School of Medicine, St Louis, MO, USA. [2]McDonnell Genome Institute, Washington University School of Medicine, St Louis, MO, USA. [3]Department of Veterinary Medicine and Surgery, University of Missouri, Columbia, MO, USA. [4]Genomics Technology Core, University of Missouri, Columbia, MO, USA. [5]Siteman Cancer Center, Washington University School of Medicine, St Louis, MO, USA. [6]Department of Genetics, Washington University School of Medicine, St Louis, MO, USA. [7]These authors contributed equally: My H. Hoang, Zachary L. Skidmore, Hans Rindt. ✉e-mail: bryanjn@missouri.edu; obigriffith@wustl.edu

TME, using single cell approaches, in a model with a functionally similar environment to that of humans could yield significant insights into the role of the TME in success or failure of immune therapy. Furthermore, single cell TCR profiling facilitates the reconstruction of the complete TCR by matching the α-β pairs of the TCR. This in turn may allow for refinement of TCR-Neoantigen-MHC binding prediction algorithms. Such refinement could have implications for personalized therapy, allowing researchers to determine if a T cell is reactive to a given neoantigen and opening avenues into engineering T cells matched to an individual tumor profile.

A limitation of the companion dog model of cancer is the relative paucity of molecular profiling reagents available compared to those for human or mouse samples. Dog genome-specific reagents are required to effectively evaluate tumor-specific immune responses in cancer immunotherapy trials. In particular, single cell immune profiling approaches are not well-developed for dogs. A small number of studies have piloted scRNAseq for canine samples[14–16]. However, there are no published protocols or commercially available kits for single cell profiling of the dog TCR repertoire (scTCRseq) from popular platforms like the Chromium 10x. We recently reported the first, to the best of our knowledge, proof-of-principle studies for single cell profiling of the TCR repertoire (scTCRseq)[17,18]. Subsequently, Eschke and colleagues constructed an atlas of sorted TCRαβ + T cells from four healthy experimental dogs using an independent approach[19]. However, to our knowledge, comprehensive single cell profiling of the TCR repertoire (scTCRseq) for client-owned dogs, unsorted tissue samples, or cancer samples has not been demonstrated. Here we address this gap by developing and validating canine α (TRA) and β (TRB) chain TCR single cell profiling for the Chromium 10x platform and apply it to a diverse set of dog samples. We show robust isolation of individual canine cells, including a large proportion of T cells, and establish detailed single cell TCR repertoires for client-owned dogs from unsorted PBMCs and lymph nodes of healthy veterinary patients and cancer patients in an immunotherapy study.

## Results

### Single cell sample preparation and V(D)J enrichment

PBMCs and lymph node aspirates from 5 healthy dogs, PBMCs from 4 dogs with melanoma, and lymph node aspirates from 2 dogs with T cell lymphoma were obtained for profiling, as outlined in Table 1. A total of 16 single cell suspensions were generated. Cell viability was greater than ~80% for all samples. 10x barcoded full-length cDNA was generated and passed QC for all samples. Nested PCR primers were successfully designed for dog TCR α (TRA) and TCR β (TRB) V(D)J chains (Fig. 1; Supplementary Fig. 1; Supplementary Table 1) and used to amplify V(D)J sequences. PCR products of the expected size (approximately 650 bp) were observed by gel electrophoresis (Supplementary Fig. 2). Illumina sequencing libraries from TCR enriched products (TRA and TRB) and unenriched scRNA cDNA were successfully generated, with suitable quantities, that also showed expected fragment size distributions (Supplementary Fig. 3; Supplementary Fig. 4).

### Immune cell composition for the normal dogs

The hematology profile from the normal dogs were within the reference intervals, Supplementary Table 2. The B, T, CD4 + T, CD8 + T and regulatory T cell flow cytometry distributions in the PBMCs and lymph node aspirates are also shown in Supplementary Table 2. As expected, in the normal controls, the CD4 + /CD8+ ratio was greater than 1, the distribution of CD8+ and CD4+ cells were similar in the lymph node and PBMCs, there was a higher % of B cells (CD21+ cells) and Tregs (assessed as (CD4 + )CD25+FoxP3+ cells or (CD25 + CD4 + )FoxP3+ cells) in the lymph node than PBMCs[20]. One dog, Normal_E, had a higher number of B cells and lower number of T cells in the lymph nodes compared to these reference ranges. The PBMCs for Normal_E on the other hand were within the reference ranges. Normal_E remained clinically normal, >309 days (as of 5Dec22) post sampling. Overall, the flow cytometry phenotyping revealed populations of lymphocytes as expected for the sources of these samples.

### Single cell V(D)J and scRNA sequencing results

Sequencing data metrics approximated comparable human data for the TCR enriched libraries (Table 2; Supplementary Data 1)[21,22]. For scTCR libraries, we observed an average of 4,863 (1,850 to 8,000) estimated V(D)J expressing cells. For these cells, we produced an average of 42,312 (range: 26,772 to 86,352) mean total reads per cell with average fraction of reads in cells of 83.3% (39.8% to 96.0%), an average of 14.2 median UMIs/cell (range: 11 to 20), and an average of 2,706 median reads/UMI (range: 735 to 4,157) per cell for each clonotype (Table 2). Note that 10x only recommends 2,000 read pairs per cell for 150 × 150 sequencing. We have sequenced at approximately 21 times the recommended depth. This is a result of our sequencing core's practice of diverting approximately 1/20 of the scRNA

## Table 1 | Overview of dog samples profiled

| Dog (N = 11) | Breed | Sample (N = 16) | Sample Type | scTCR | scRNA | Flow |
|---|---|---|---|---|---|---|
| Normal_A | Australian Shepherd | Normal_A_LN | Lymph Node | Y | | Y |
| Normal_A | Australian Shepherd | Normal_A_PBMC | PBMC | Y | | Y |
| Normal_B | Mixed breed | Normal_B_LN | Lymph Node | Y | | Y |
| Normal_B | Mixed breed | Normal_B_PBMC | PBMC | Y | | Y |
| Normal_C | Great Pyrenees | Normal_C_LN | Lymph Node | Y | | Y |
| Normal_C | Great Pyrenees | Normal_C_PBMC | PBMC | Y | | Y |
| Normal_D | Dachshund | Normal_D_LN | Lymph Node | Y | | Y |
| Normal_D | Dachshund | Normal_D_PBMC | PBMC | Y | | Y |
| Normal_E | Great Dane | Normal_E_LN | Lymph Node | Y | | Y |
| Normal_E | Great Dane | Normal_E_PBMC | PBMC | Y | | Y |
| Melanoma_A | Bouvier des Flandres | Melanoma_A_PBMC | PBMC | Y | Y | |
| Melanoma_B | American Cocker Spaniel | Melanoma_B_PBMC | PBMC | Y | Y | |
| Melanoma_C | Dachshund | Melanoma_C_PBMC | PBMC | Y | Y | |
| Melanoma_D | Mixed breed | Melanoma_D_PBMC | PBMC | Y | Y | |
| Tzone_LSA | Golden Retriever | Tzone_LSA_LN | Lymph Node | Y | Y | Y |
| PTCL_NOS | Golden Retriever | PTCL_NOS_LN | Lymph Node | Y | | |

*scTCR* single cell TCR sequencing, *scRNA* single cell gene expression sequencing.

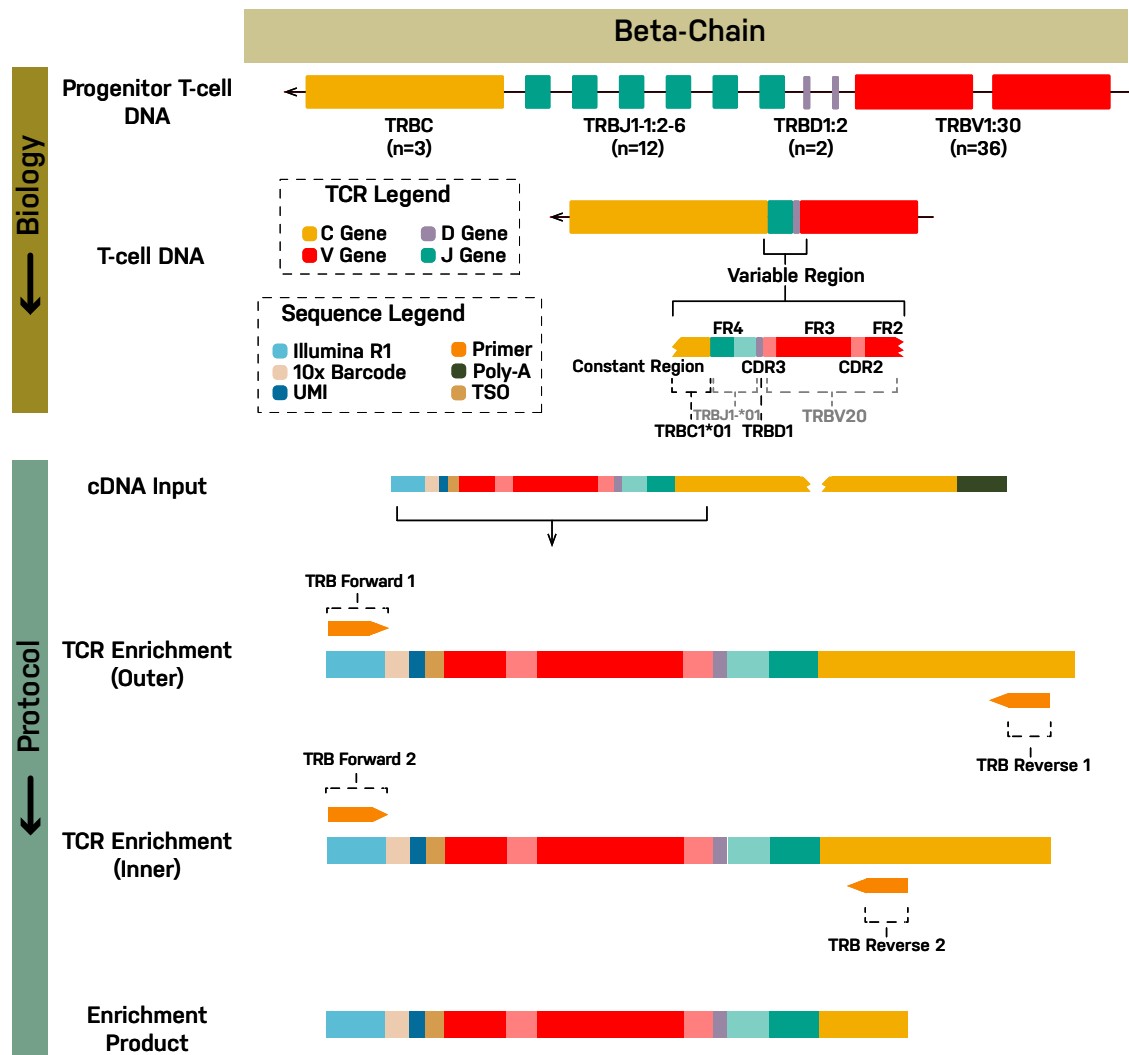

**Fig. 1 | TCR β chain enrichment strategy for use with 10x scRNA sequencing.** The TCR V(D)J enrichment strategy is depicted using the β chain for illustration (see Supplementary Fig. 1 for α chain). At top, the genomic un-rearranged TRB locus is shown (Note: TRB is located on the negative strand in dogs). During T cell development from progenitor T cells to mature T cells, individual V, D, J and C gene segments are rearranged by somatic recombination to produce a functional TRB locus. Transcription and splicing produce a pre-mRNA and then mRNA for the complete V(D)JC transcript sequence (not shown). In the modified 10x protocol, mRNA (including TRB mRNA) is converted to cDNA. TRB cDNA is then amplified using a nested PCR design. The forward primers from the 10x protocol were left unchanged (v2 protocol shown). In the first cycle, the forward primer (TRB Forward 1) primes off the Illumina read 1 (R1) sequencing adapter that is incorporated during generation of cDNA. In the second cycle, the identical forward primer (TRB Forward 2) again primes off the R1 sequence. The first reverse primer (TRB Reverse 1, Outer) primes off the constant (C) region gene segment. The second reverse primer (TRB Reverse 2, Inner) similarly primes off the C region but at an inner, 5' position relative to the outer primer. The β chain primer design was based off a dog TCR β rearranged partial mRNA (GenBank: HE653957.1) which was extended to include (from 3' to 5') the R1 adapter, 10x cell barcode, UMI, TSO, V, D, J, and C gene segments. The constructed cDNA sequence was then used as input to primer3plus (4.0), with forward primers provided as described above, and a target region for reverse primer specified in the C region. The product of the first (outer) design was used as input for the second (inner) design.

sequencing run towards scTCR. In total, an average of 187.7 (87.0 to 236.8) million reads were generated for each of the 16 V(D)J libraries with 96.2% (94.4 to 97.1%) valid barcodes. Average Q30 bases for barcodes, Unique Molecular Identifier (UMI), Read 1, and Read 2 sequences were 91.5, 90.9, 85.7, and 87.8% and respectively.

For the scRNA libraries we also obtained results comparable to human data (Table 3; Supplementary Data 2)[23]. We observed 6,968-9,600 estimated cells with mean reads per cell of 19,952–72,253 and fraction of reads in cells of 95.9–96.9%. In total, 191.5–514.4 million reads were generated for each of the 5 scRNA libraries with 91.9–93.4% valid barcodes. Average Q30 bases for barcodes, Read 1, Read 2, and UMI sequences were 89.3, 82.9, 85.3, and 88.9%, respectively. For gene estimation purposes, 67.1–73.4% of total reads mapped to the genome with 14,090–14,751 total genes detected per sample and 1,162–1,635 median genes per cell.

## Single cell V(D)J repertoire of dogs

In total, across the 16 samples, we profiled 77,809 V(D)J expressing cells and 55,973 clonotypes from 3.0 billion sequence reads. For TCR enriched libraries 72.0%-85.6% of reads mapped to a V(D)J gene with an approximate ratio between β/α chains of 1.5:1 (Table 2; Supplementary Data 1). Between 45.2 and 96.9% (median 83.3%) of cells exhibited a productive TRA contig. Similarly between 83.5 and 99.6% (median 97.6%) of cells exhibited a productive TRB contig. Cells with a productive V-J spanning pair (TRA + TRB) ranged from 34.9-95.3% (median 80.6%). Across all 16 samples, 44,544 (79.6%) clonotypes had at least one TRA/TRB pair (Supplementary Data 1 and 3). Of these, ~7.7% had an extra TRA, 4.4% had an extra TRB and 2.0% had extras of both TRA and TRB. Only 8 clonotypes (0.01%) had more than 2 TRA or TRB chains which would be suggestive of unfiltered duplicates or some other problem. On average each sample had a

**Table 2 | Single cell V(D)J sequencing summary metrics (*N* = 16 samples)**

|  | Metric | Mean | Median | Min | Max | stdev |
|---|---|---|---|---|---|---|
| **Cells** | Estimated number of cells | 4,863.1 | 5,282.5 | 1,850 | 8,000 | 1,702.0 |
|  | Mean reads/cell | 42,312.1 | 40,209.5 | 26,772 | 86,352 | 15,113.3 |
|  | Mean used reads/cell | 24,358.3 | 24,203.0 | 16,322 | 32,044 | 4,264.9 |
|  | Number Cells with Productive V-J Spanning Pair | 3,908.1 | 3,995.2 | 758 | 7,320 | 1,763.5 |
|  | Fraction reads in cells | 83.3% | 87.4% | 39.8% | 96.0% | 16.0% |
| **Enrichment** | Reads Mapped to Any V(D)J Gene | 78.9% | 79.1% | 72.0% | 85.6% | 3.9% |
|  | Reads Mapped to TRA | 32.2% | 33.3% | 21.6% | 41.7% | 5.8% |
|  | Reads Mapped to TRB | 46.5% | 47.4% | 30.9% | 58.0% | 7.4% |
| **V(D)J Expression** | Median TRA UMIs per Cell | 7.6 | 5.5 | 3 | 30 | 6.9 |
|  | Median TRB UMIs per Cell | 13.5 | 10.0 | 9 | 49 | 10.2 |
|  | Median UMIs per Cell per Clonotype | 14.2 | 13.5 | 11 | 20 | 2.4 |
|  | Median reads per UMI per Clonotype | 2,706 | 2,718 | 735 | 4,157 | 907 |
| **V(D)J Annotation** | Cells With Productive V-J Spanning Pair | 76.8% | 80.6% | 34.9% | 95.3% | 15.7% |
|  | Paired Clonotype Diversity | 2,408.1 | 1,986.2 | 1.3 | 5,872.8 | 2,257.8 |
|  | Cells With TRA Contig | 87.3% | 90.0% | 53.3% | 98.0% | 11.1% |
|  | Cells With TRB Contig | 98.1% | 99.2% | 91.2% | 99.7% | 2.5% |
|  | Cells With CDR3-annotated TRA Contig | 85.3% | 88.2% | 48.8% | 97.4% | 12.0% |
|  | Cells With CDR3-annotated TRB Contig | 97.2% | 98.8% | 87.4% | 99.7% | 3.7% |
|  | Cells With V-J Spanning TRA Contig | 86.6% | 89.4% | 50.9% | 98.0% | 11.7% |
|  | Cells With V-J Spanning TRB Contig | 97.3% | 98.8% | 87.9% | 99.7% | 3.6% |
|  | Cells With Productive TRA Contig | 81.0% | 83.3% | 45.2% | 96.9% | 12.2% |
|  | Cells With Productive TRB Contig | 95.8% | 97.6% | 83.5% | 99.6% | 4.3% |
|  | Unique Clonotypes | 3,498.3 | 3,664.0 | 348 | 5,971 | 1,993.4 |
| **Sequencing** | Number of Reads | 187,675,798 | 201,993,022 | 87,026,777 | 236,830,110 | 41,683,334 |
|  | Valid Barcodes | 96.2% | 96.4% | 94.4% | 97.1% | 0.8% |
|  | Q30 Bases in Barcode | 91.5% | 90.9% | 90.4% | 96.8% | 2.1% |
|  | Q30 Bases in RNA Read 1 | 85.7% | 84.7% | 84.0% | 93.8% | 3.2% |
|  | Q30 Bases in RNA Read 2 | 87.8% | 87.7% | 86.0% | 90.0% | 1.2% |
|  | Q30 Bases in UMI | 90.9% | 90.3% | 89.7% | 96.2% | 2.1% |

Paired Clonotype Diversity is computed as the Inverse Simpson Index of the clonotype frequencies. A value of 1 indicates a minimally diverse sample – only one distinct clonotype was detected. A value equal to the estimated number of cells indicates a maximally diverse sample.

median of 69.6% clonotypes with a single TRA/TRB pair, 2.6% TRA only, 16.6% TRB only, 6.3% extra TRA, 3.5% extra TRB, and 1.3% extra TRA and TRB. In samples processed with 10x v1 5′ kits we observed noticeably lower median TRA/TRB pairs (30.9%) compared to samples processed with v2 kits (69.8%). Many v1 clonotypes were TRA only (20.2%) or TRB only (40.9%) compared to only 2.5 and 15.7%, respectively, for v2. The relative amounts of extra TRA (2.3%) and extra TRB (5.4%) in v1 shifted to 6.6% extra TRA and 3.3% extra TRB in v2, more in line with biological expectations for dual TCR expression[24]. These numbers are consistent with those from the Eschke et al. study which also used the v2 10x protocol (62.1% TRA/TRB, 8.7% TRA only, 21.1% TRB only)[19].

In general, specific expanded TRB sequences tended to be consistently paired with a specific TRA and vice versa. This was especially true for samples processed with the 10x v2 protocol. For example, if we consider Melanoma_C, the most dominant clonotype was TRA CDR3:CAMGP-VYSGVGSQLTF (TRAV9-8::TRAJ28) paired with TRB CDR3:CA-SAGQGDPHTQYF (TRBV28::TRBJ2-5) inferred for 147 cells (Supplementary Data 3). We do not see this specific TRA sequence matching any other TRB sequences or vice versa. In other words, the TRA (CAMGPVYSGVGSQLTF) clonotype and TRB (CASAGQGDPHTQYF) are only seen with each other. This is true for at least the top 5 clonotypes for this sample with only minor exceptions. In some cases a beta chain is matched with different alpha chains in separate clonotypes but the CDR3 amino acid sequences are identical and only small nucleotide level differences are observed for single cells.

We observed the expression of 31 TRAV, 47 TRAJ, 24 TRBV and 12 TRBJ known gene segments at some level of support. The top 5 most frequently used genes observed in our dataset were TRAV43-1, TRAV9-6, TRAV43-4, TRAV12, and TRAV9-9 for TRAV; TRAJ21, TRAJ33, TRAJ27, TRAJ28, and TRAJ31 for TRAJ; TRBV20, TRBV16, TRBV7, TRBV3-2, and TRBV5-2 for TRBV; and TRBJ2-6, TRBJ2-1, TRBJ2-3, TRBJ2-5, and TRBJ1-2 for TRBJ. IMGT categorizes all gene segments as either "Functional," "Pseudogene," or "ORF." Functional gene segments have an open reading frame (ORF) without stop codon, and no defects in the splicing sites, recombination signals and/or regulatory elements. A pseudogene segment is one whose coding region has stop codon(s) and/or frameshift mutation(s), and/or a mutation affecting the initiation codon. An ORF segment has an open reading frame, but alterations have been described in the splicing sites, recombination signals and/or regulatory elements, and/or changes of conserved amino acids that may lead to incorrect folding. For simplicity, we refer to these three types as functional, pseudogene, or non-functional ORF. Considering these annotations, we observed 85.3% (29/34), 100.0% (40/40), 100.0% (22/22) and 100% (9/9) of all known functional TRAV, TRAJ, TRBV and TRBJ gene segments respectively (Fig. 2; Supplementary Fig. 5). We also observed expression of 3/44 pseudogene segments (1/23 TRAV, 0/7 TRAJ, 1/13 TRBV, and 1/1 TRBJ) and 11/16 non-functional ORF gene

**Table 3 | Single cell gene expression (scRNA) sequencing summary metrics (*N* = 5 samples)**

|  | Metric | Mean | Median | Min | Max | Stdev |
|---|---|---|---|---|---|---|
| **Cells** | Estimated number of cells | 8,365.4 | 8,554.0 | 6,968 | 9,600 | 1,134.6 |
|  | Fraction Reads in Cells | 1.0 | 1.0 | 95.9% | 96.9% | 0.4% |
|  | Mean Reads per Cell | 43,009.6 | 28,875.0 | 19,952 | 72,253 | 25,462.9 |
|  | Median Genes per Cell | 1,447.0 | 1,531.0 | 1,162 | 1,635 | 186.0 |
|  | Total Genes Detected | 14,549.2 | 14,610.0 | 14,090 | 14,751 | 268.6 |
|  | Median UMI Counts per Cell | 4,237.0 | 4,421.0 | 3,246 | 4,955 | 720.3 |
| **Sequencing** | Number of Reads | 337,945,537 | 267,296,412 | 191,543,369 | 514,405,307 | 158,561,283 |
|  | Number of Short Reads Skipped | 0 | 0 | 0 | 0 | 0 |
|  | Valid Barcodes | 92.6% | 92.7% | 91.9% | 93.4% | 0.6% |
|  | Valid UMIs | 99.8% | 99.8% | 99.8% | 99.9% | 0.1% |
|  | Sequencing Saturation | 64.0% | 59.6% | 49.6% | 80.1% | 13.9% |
|  | Q30 Bases in Barcode | 89.3% | 90.0% | 87.9% | 90.1% | 1.1% |
|  | Q30 Bases in RNA Read 1 | 82.9% | 81.7% | 81.6% | 85.1% | 1.7% |
|  | Q30 Bases in RNA Read 2 | 85.3% | 85.3% | 84.9% | 85.8% | 0.4% |
|  | Q30 Bases in UMI | 88.9% | 89.4% | 87.8% | 89.7% | 0.8% |
| **Mapping** | Reads Mapped to Genome | 70.3% | 69.9% | 67.1% | 73.4% | 2.9% |
|  | Reads Mapped Confidently to Genome | 67.9% | 67.6% | 65.0% | 71.0% | 2.6% |
|  | Reads Mapped Confidently to Intergenic Regions | 7.0% | 6.8% | 5.8% | 8.2% | 0.9% |
|  | Reads Mapped Confidently to Intronic Regions | 6.1% | 6.2% | 5.1% | 7.2% | 0.8% |
|  | Reads Mapped Confidently to Exonic Regions | 54.9% | 55.6% | 52.2% | 56.7% | 1.9% |
|  | Reads Mapped Confidently to Transcriptome | 47.8% | 47.2% | 45.9% | 50.6% | 1.7% |
|  | Reads Mapped Antisense to Gene | 2.3% | 2.2% | 2.1% | 2.6% | 0.2% |

segments (1/1 TRAV, 7/12 TRAJ, 1/1 TRBV, and 2/2 TRBJ). Overall, most expressed pseudogene/ORF segments were observed at low frequency. However, those that were observed were often expressed with a large diversity of different partner gene segments and in a few cases there were clonotypes involving a pseudogene or non-functional segment that were supported by many cell barcodes, including the highly dominant clonotype (TRBV26::TRBJ1-3) for the T zone lymphoma. Only a single combination occurred where both V and J segments were annotated as pseudogenes (TRBV19::TRBJ1-3; *n* = 7 cells, all in Normal_E samples) and only six combinations occurred where both V and J segments were annotated as non-functional ORF or one was a pseudogene and the other a non-functional ORF (Fig. 2; Supplementary Fig. 5). In some cases the use of pseudogene or non-functional ORF segments was unique to a single individual (e.g., TRAJ22 and TRBV19) but in others they were observed in multiple samples (e.g., TRAV9-2, TRAV33, TRAJ10, TRAJ11, TRAJ46, TRAJ59, and TRBJ2-4) or across the entire sample set (e.g., TRAJ52, TRAJ53, TRBV15, TRBJ1-3, and TRBJ1-4) (Fig. 3). The observation of three pseudogene segments (TRAV9-2, TRBV19 and TRBJ1-3), being used in productive CDR3 clonotypes, was unexpected. According to IMGT, the reference sequences for these gene segments each include one or more frameshift or in-frame stop codon mutations that should result in early termination of the TCR ORF. As a result, these would not be expected to generate the productive CDR3 sequences required by cellranger to be reported as a clonotype. To explain this result, we performed additional sequence analyses for each pseudogene as described below.

### Sequence analysis of TRAV9-2 pseudogene usage
The TRAV9-2 pseudogene was identified in 109 unique clonotypes, in combination with 33 different TRAJ gene segments, in 113 cells, for 15 dog samples sequenced (all except Tzone_LSA_LN). IMGT notes an in-frame stop codon in FR1 and a frameshift in FR2. Investigation of 10 randomly selected clonotype sequences, one from each dog, revealed that there were 9 consistent differences (8 SNVs and a single base insertion) in all clonotypes relative to the reference (Supplementary Fig. 6a). In each

clonotype sequence, the first stop (TGA) at amino acid position 45 is "corrected" to an S (TCA) by a single base change (G - > C). An insertion of a single base (C) at amino acid position 57 introduces a frameshift relative to the reference sequence. From this frameshift, the clonotype amino acid sequence diverges completely, with the reference sequence having 3 additional stop codons (Supplementary Fig. 6b) whereas the clonotype sequence continues in an open reading frame for the remaining length of the V gene (Supplementary Fig. 6c). Therefore, we propose an alternate consensus sequence for TRAV9-2 which is not a pseudogene (Supplementary Data 4). This alternate sequence better represents the 11 dogs (1 Bouvier des Flandres, 1 American Cocker Spaniel, 2 Dachshund, 2 mixed breed, 1 Australian Shepherd, 1 Great Pyrenees, 1 Great Dane, 2 Golden Retrievers) that we sequenced, and also has full-length 100% identity, including the inserted C sequence that corrects frame, with more current reference assemblies UU_Cfam_GSD_1.0/canFam4 (German Shepherd, chr8) and UMICH_Zoey_3.1/canFam5 (Great Dane, chrUn_REHQ01000600v1). A BLAST of the alternate TRAV9-2 sequence against 'nt' also reveals near perfect matches, including the frame-correcting insertion, to both wolf (GenBank: HG994390.1) and Labrador (GenBank: CP050567.1) sequences. In contrast, the reference TRAV9-2 does not align without discrepancies to these genomes. Surprisingly, the Dog10K_Boxer_Tasha/canFam6 (chr8) reference still matches the IMGT reference TRAV9-2 perfectly, including the C deletion that shifts frame but does not match the alternate sequence without discrepancies. Altogether, these results suggest that the TRAV9-2 pseudogene reference sequence in IMGT is either incorrect or a rare variant specific to the canFam3/canFam6 Boxer.

### Sequence analysis of TRBV19 pseudogene usage
The TRBV19 pseudogene was identified in 93 unique clonotypes, in combination with all 12 known TRBJ gene segments, in 99 cells, but from only a single dog (Normal_E). IMGT notes an in-frame stop codon in FR3. Investigation of the Normal_E V(D)J sequence data revealed a germline (Donor Reference) T- > C mutation (Supplementary Fig. 7a) that changes a

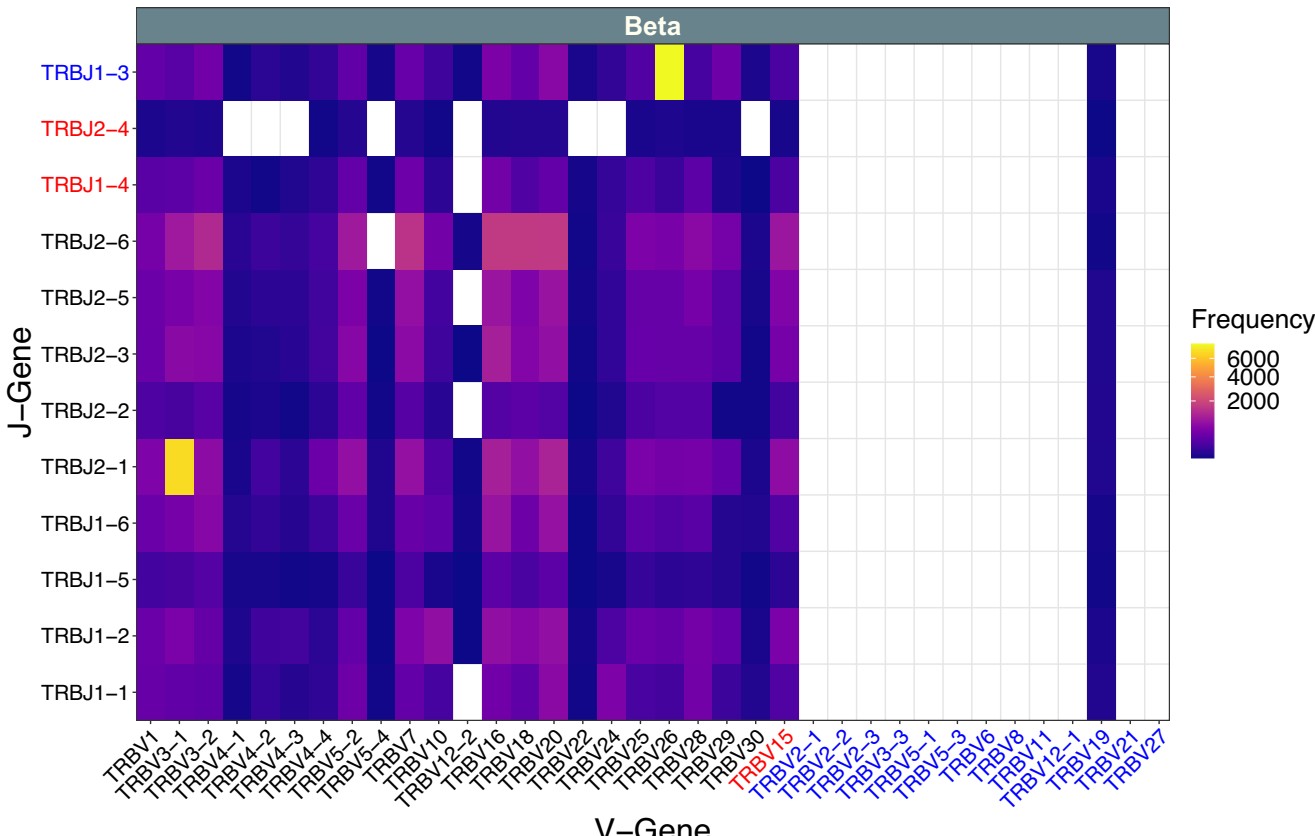

**Fig. 2 | Aggregate VJ Gene Segment Usage for dog T-cell receptor β chain.** The VJ gene combinations identified in samples across all dogs (*n* = 16; normal LN, normal PBMC, melanoma PBMC, lymphoma LN) are plotted along with their observed cell barcode counts (Frequency) for the TRB chain. The corresponding VJ gene usage for the TRA chain is shown in Supplementary Fig. 5. Text in black indicates a functional annotation according to IMGT; blue indicates a pseudogene; red indicates a gene segment that has an ORF but also a defect in the splicing sites, recombination signals and/or regulatory elements or other features disqualifying a functional annotation.

stop codon (TAG) at amino acid position 104 in the reference (Supplementary Fig. 7b) to a Q (CAG) in the clonotype sequence (Supplementary Fig. 7c). This variant was present in both lymph node and PBMC samples for the Normal_E dog and observed in all predicted clonotypes, making use of TRBV19 in this dog. In all other samples, there were no clonotypes making use of TRBV19, presumably because they do not have this "corrective" germline mutation and any rearrangement involving TRBV19 would be truncated, non-productive, and filtered out by cellranger. Analysis of additional dogs will be required to determine how common this corrective variant is and whether this gene segment should retain annotation as pseudogene and/or have a note about occasional function or an alternate allele be added to IMGT.

**Sequence analysis of TRBJ1-3 pseudogene usage**

The TRBJ1-3 pseudogene was identified in 3,413 unique clonotypes, in combination with all 22 known functional TRBV gene segments (and one non-functional ORF and one pseudogene), in 11,711 cells, for all 16 dog samples sequenced. This included the T zone lymphoma (Tzone_LSA_LN) which had a highly dominant clonotype making use of TRBV26 joined to TRBJ1-3 that was expressed in 95.2% (7,615) of cells for this sample. First, we investigated this specific Tzone_LSA_LN clonotype. IMGT notes an in-frame stop codon in the J region of TRBJ1-3. IMGT also notes a stop-codon in the last 3' codon of TRBV26 "which may disappear during rearrangements". Presumably for this reason TRBV26 was not annotated as a pseudogene. In any case, the dominant clonotype of the T zone lymphoma apparently had to overcome two defects. Investigation of the dominant T zone lymphoma clonotype sequence revealed an insertion of 4 bp and 5 other single base pair changes closely flanking the VJ joining boundary (Supplementary Fig. 8a). If TRBV26 reference sequence is joined directly to

the TRBJ1-3 reference sequence, the result includes the IMGT-annotated stop (TAG) at the end of the V segment and the entire J segment is out of frame, resulting in a stop at the end of the J gene, and presumably creating problems even beyond the stop at amino acid position 2 that otherwise defines TRBJ1-3 as pseudogene (Supplementary Fig. 8b). However, in the altered clonotype sequence we observed there are no stop codons and the J-gene portion is in the correct frame (Supplementary Fig. 8c). In this case, multiple changes (insertions and substitutions) upon V-J joining are "correcting" a premature stop both at the end of TRBV26 and beginning of TRBJ1-3 gene while preserving the correct frame to make a productive CDR3 sequence. To further investigate use of TRBJ1-3, an additional 9 randomly selected clonotype sequences, one from each dog, were examined. In all cases, unique sequence changes introduced during VJ joining eliminate the stop codon from the beginning of TRBJ1-3 while preserving the correct frame (Supplementary Fig. 8d). Overall, TRBJ1-3 was the 8th (of 12) most commonly used TRBJ gene segment in terms of median unique clonotypes with a median of 238 unique clonotypes per sample (Supplementary Fig. 9). Given the very extensive use of TRBJ1-3 in productive TCRs we propose that that this gene segment should be re-annotated as functional, perhaps with a note similar to TRBV26, TRBV28, or TRBJ2-1 that while there is a stop-codon in the second 5' codon this may disappear during rearrangements.

**Sequence analysis of ORF gene segment usage**

For the 11 ORF gene segments that we observed to be expressed, the majority (8/11) were annotated as having non-canonical V/J heptamer or nonamer sequences or unexpected spacer lengths in the recombination recognition sequences (Supplementary Data 5). These sequences are recognized by the RAG1/RAG2 enzyme complex during V(D)J

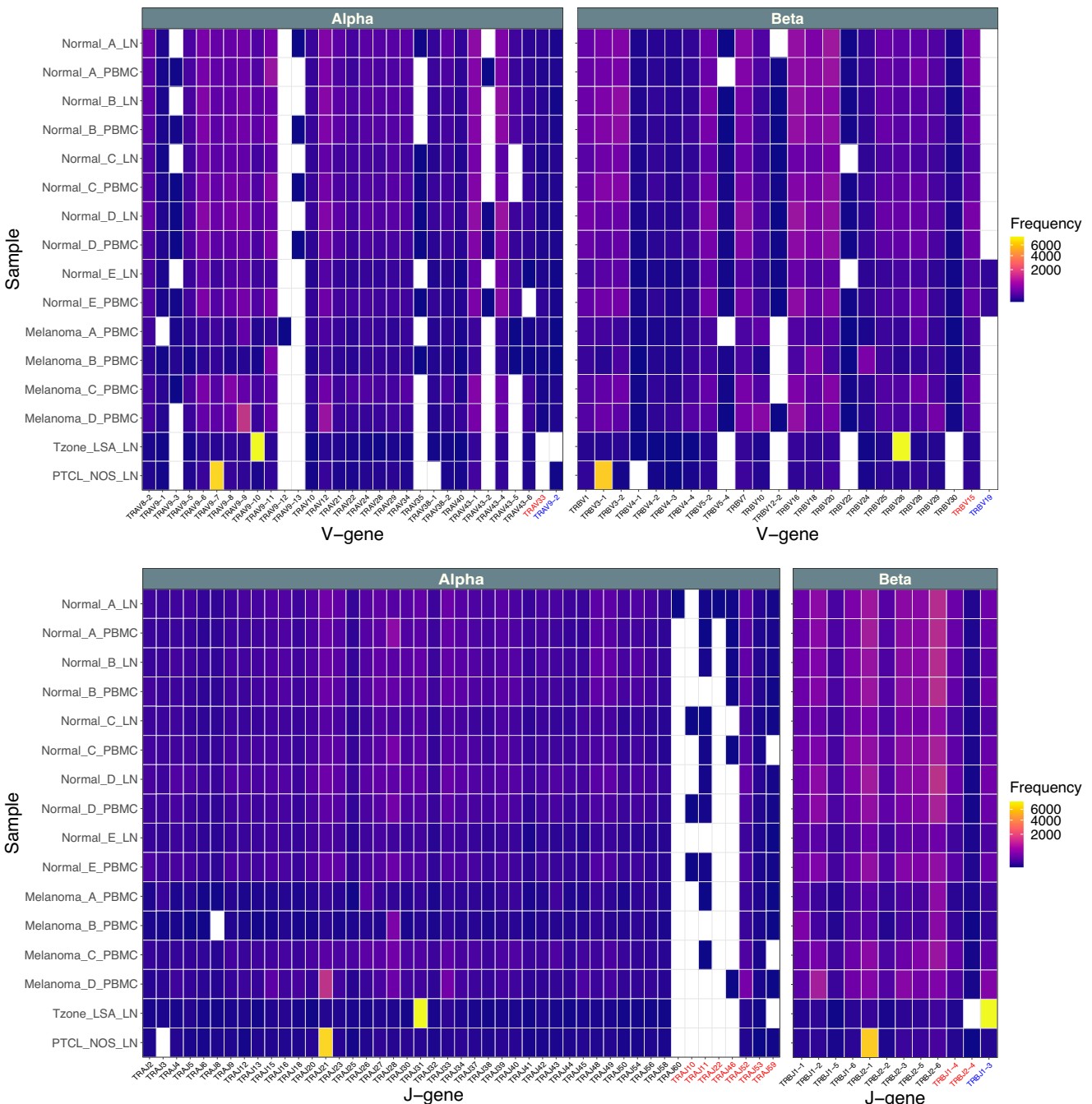

**Fig. 3 | TRA/TRB V and J gene segment usage by sample.** The observed cell barcode counts for the TRA chain (Alpha) and TRB chain (Beta) for V-gene and J-gene segments are shown for each sample. Note, gene segments not detected in any sample are not shown. For observation of the complete set of known gene segments (aggregated across all samples) see Fig. 2 and Supplementary Fig. 5. Text in black indicates a functional annotation according to IMGT; blue indicates a pseudogene; red indicates gene segment that has an ORF but also a defect in the splicing sites, recombination signals and/or regulatory elements or other features disqualifying a functional annotation. Note, in rare cases where multiple alleles are expressed, the same cell/barcode can be counted toward multiple gene segments of the same class or even the same gene segment.

recombination. Non-canonical RSS or spacer sequences are expected to interfere with the efficiency of recombination. Despite this, we clearly observed consistent expression of V(D)J transcripts using these gene segments. There are multiple possible explanations for this. There could be errors in the reference sequences, or germline differences between the reference dog and those we sequenced, or more tolerance to these sequence changes for the RAG1/RAG2 enzyme binding than expected. However, because the recognition sequences are lost during the process of recombination and not included in the final V(D)J transcript sequence, we can not test these theories from scTCR data. Genomic DNA sequence-level analysis

would be required. We also see use of 4 gene segments with non-conserved FGXG sequences and one gene segment with a conserved TRP replaced by ARG. As above, there are multiple possible explanations for use of these non-functional ORF gene segments. At least three ORF gene segments in particular were utilized very frequently based on median number of unique clonotypes observed across the 16 samples, including TRAJ52 (8/59 most commonly used TRAJ segment; median 110 unique clonotypes), TRBV15 (8/36 most commonly used TRBV segment; median 240 unique clonotypes), and TRBJ1-4 (9/12 most commonly used TRBV segment; median 184 unique clonotypes) (Supplementary Fig. 9). Further analyses are

warranted, but based on the frequency of expression of these three ORF gene segments in our data, we propose they could be re-annotated as functional, as above for at least two pseudogene segments.

## V(D)J contig length assessment

The lengths for reference TRA V-J and TRB V-D-J gene segment combinations have a median of 399 (346 to 451) and 406 (366 to 426) nt (nucleotides), respectively. Note that the reference length doesn't take into account the length of the 5' UTR and part of the C region sequenced by the protocol, and therefore will naturally be shorter than observed contig lengths. The observed contig lengths (in nt) have a median of 501 (409 to 1122) for alpha chain, and 511 (406 to 1203) for beta chain. The majority of the contig lengths fall within the expected range, with a few outliers. The outliers, which are longer, could result from: (1) misassembly, where the contig is a chimera between two different transcripts which have been assembled together by accident; (2) translation of a long (potentially alternate) 5' UTR; (3) amplification of a larger section of C region due to mis-priming; (4) sequencing of intronic regions from incompletely processed

RNA that was amplified; or other explanations. The median observed CDR3 lengths are 42 (15 to 72) nt, or 14 (5 to 24) aa (amino acids) for alpha chain; and 42 (18 to 66) nt, or 14 (6 to 22) aa for beta chain (Supplementary Fig. 10).

## V(D)J clonality and diversity assessment

Overall, a large diversity of clonotypes and range of clonality was observed with an average of 3498 (348 to 5,971) unique TRA/TRB clonotypes identified per sample (Fig. 4; Supplementary Data 3). All healthy normal PBMC and lymph node samples displayed nearly total V(D)J diversity with an almost complete absence of clonal expansion. The single most dominant clonotype represented an average of only 0.09% (0.06 to 0.13%) of cells for healthy lymph nodes with a very high average paired clonotype diversity (Inverse Simpson Index) of 4,470.9 (2,210.8 to 5,872.8) and average total paired or unpaired unique TRA/TRB clonotypes of 4,562 (2,250 to 5,971) identified for an average of 4,642 (2,290 to 6,040) total V(D)J expressing cells. For healthy PBMCs, the single most dominant clonotype was on average only 0.91% (0.32 to 1.73%) of cells, average diversity was 3128.4 (1265.8 to 4,719.9) and average unique clonotypes was 4,910 (3,889 to 5,512)

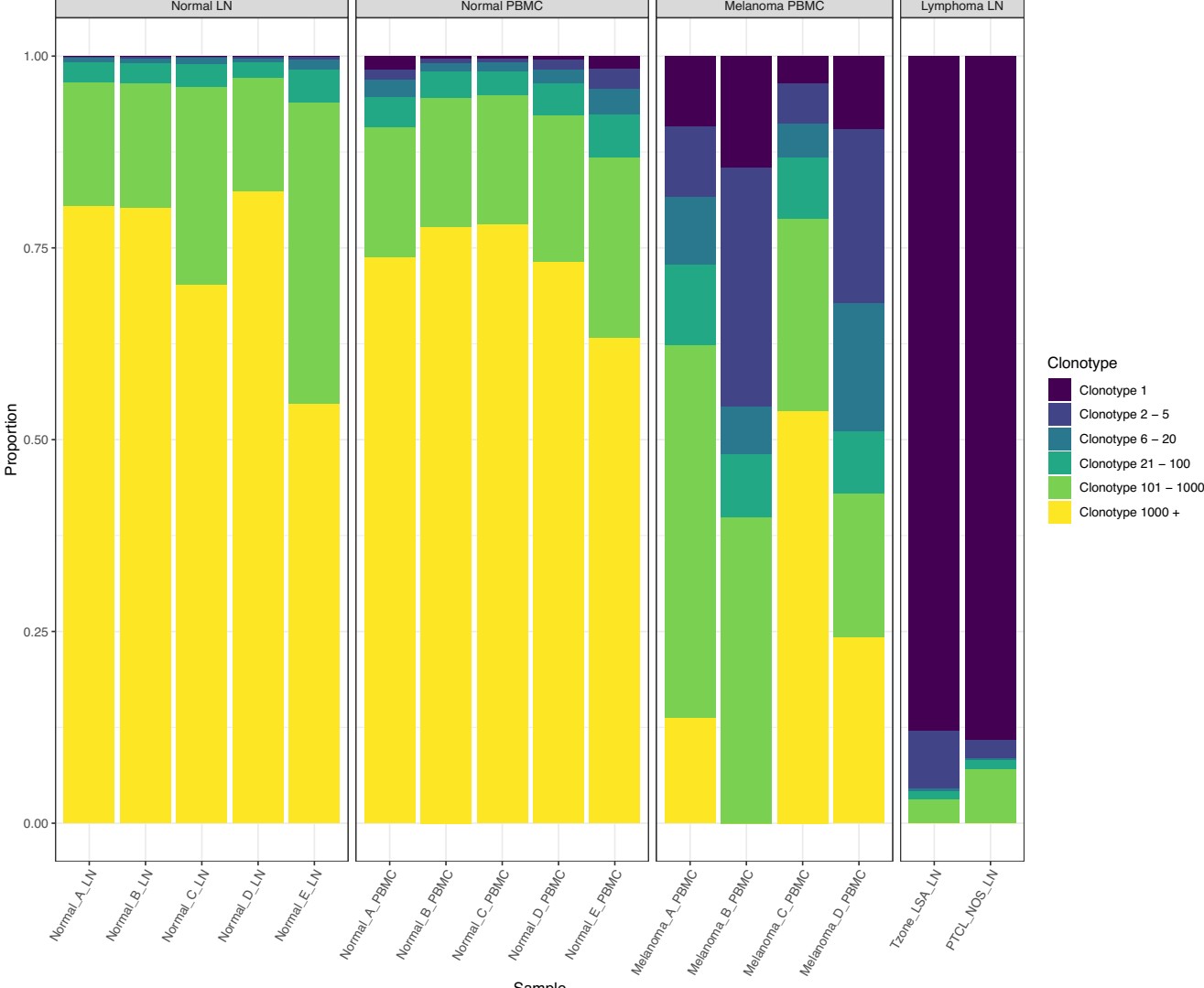

**Fig. 4 | Single cell clonotype distribution for TRA and TRB chains for all samples.** The proportion of total barcodes, for all clonotypes, is shown for the lymph node aspirates (Normal_LN) and PBMCs (Normal_PBMC) from five healthy dogs, PBMCs from four dogs with melanoma (Melanoma_PBMC) and lymph node aspirates from two dogs with T cell lymphoma (Lymphoma_LN). Proportion was estimated by calculating the fraction of cells in each bin (Clonotype 1, Clonotype

2–5, etc where the clonotypes are sorted in descending order of cell counts). The healthy normal samples are characterized by highly diverse clonotypes, with even the most frequent clonotype observed in only a very small proportion of cells. The melanoma PBMC cases are characterized by a small number of dominant clonotypes with higher frequency. The T cell lymphoma cases are characterized by one dominant clonotype in each case.

for an average of 5,320 (4,574 to 5,806) total cells. In contrast, the T zone lymphoma and PTCL had completely dominant clonotypes representing 88.0 and 89.2% of cells, very low diversity of 1.28 and 1.26, and total unique clonotypes of only 348 and 562 for 8,000 and 6,649 total cells respectively. All PBMCs from dogs with melanoma exhibited intermediate but relatively clonal TCR repertoires, with the top clonotype representing on average 9.2% (3.5 to 14.5%) of cells and average paired clonotype diversity of 132.7 (19.4 to 390.6). Each melanoma PBMC displayed one or a few CDR3 clonotypes accounting for a substantial fraction of all cell barcodes. For example, for Melanoma_B, just two TRA and two TRB clonotypes account for approximately 20% and 40% of all cell barcodes respectively. At the same time, a large diversity of clonotypes was captured with an average of 1,925 (966 to 3,232) unique clonotypes identified for 3,338 (1,850 to 5,176) total cells.

A potential limitation of the analysis of clonotype diversity (Fig. 4) is that cell numbers across samples were not equal (range 1,850 to 8,000). To address this concern, we computed average cell proportions, across 100 permutations, randomly downsampled, with replacement, to the lowest number of clonotyped T cells observed in any sample ($n = 1,850$ cells). As shown in Supplementary Fig. 11, the downsampled data show that the patterns of TCR diversity are highly concordant with the non-downsampled data. This analysis suggests that our conclusions regarding different clonality/diversity between sample groups (healthy vs melanoma vs lymphoma) are not confounded by differences in T cell number.

We also assessed the clonotype diversity for CD4+ and CD8+ subsets (defined by singleR cell typing) for the 5 samples with gene expression data (Supplementary Fig. 12). The CD8+ subset of the T zone lymphoma sample ($n = 6,938$ cells) is characterized by a single dominant clonotype that accounts for 92% of the cells. In contrast, the CD4+ subset of the T zone lymphoma ($n = 158$ cells) was more diverse with the most dominant clone accounting for only 19.6%. This suggests that the T zone lymphoma malignant clone is predominantly CD8+. Similarly, for three of four melanoma PBMCs, the CD8 subset of cells were generally more oligoclonal (less diverse) compared to CD4 cells. However, both had some evidence of clonotype expansion as well as a diversity of different clonotypes. Melanoma_D_PBMC had similar patterns of CD4 and CD8 expansion. As above, downsampling to a common minimum number of cells did not affect these conclusions.

As expected, individual clonotypes were characterized by evidence of germline variation, V(D)J recombination diversity (usage of different V, D and J gene segments in different combinations), as well as the recombination-related mutations at V(D)J junctions which occur during gene segment joining and contribute to TCR diversity (see Fig. 5 for a representative example clonotype).

### Germline analysis
On average, each sample had 26.75 (range: 17 to 36) alternate allele sequences compared to reference in 21 (range 11 to 28) TRAV or TRBV genes (Supplementary Data 6). There were 23 TRAV and 13 TRBV genes with at least one alternate allele observed in at least one sample (Supplementary Data 7). Note that germline variant assessment for J genes is currently not performed by cellranger.

### Integration of single cell V(D)J repertoire and gene expression data
For all 5 dogs with scRNA data (Table 1), t-SNE clustering based on single cell gene expression patterns identified a number of distinct clusters of cells (Fig. 6). Expression of *CD3E* (a general T cell marker) clearly demarcated a subset of major and minor clusters, representing a substantial fraction of all cells of ~50% for melanoma PBMCs and >90% for the T zone lymphoma (Fig. 6a). Cell typing based on established blood cell type gene expression signatures (mapped from human genes to dog orthologs) for Melanoma PBMCs identified an average of 51.3% T cells (41.1 to 60.3%), 23.5% CD4 + T cells (19.3 to 26.6%), 27.8% CD8 + T cells (21.8 to 33.7%), 29.6% monocytes (23.6 to 35.6%), 12.0% B cells (4.6 to 21.0%), 2.3% erythroid cells

(0.4 to 4.3%), 1.6% dendritic cells (0.1 to 3.2%), 0.6% NK cells (0.3 to 1.2%), 0.7% megakaryocyte cells (0.2 to 1.4%) and less than 1% maximum for any other cell type (Fig. 6b; Supplementary Table 6). These cell type proportions were consistent with previously reported results in human PBMC samples[25], expectations for dog PBMC samples, and cell fractions observed for normal PBMCs by flow cytometry (Supplementary Table 2)[26]. For the T zone lymphoma, cell typing identified 94.4% T cells (2.4% CD4 +; 92.0% CD8 +), 4.2% B cells, and less than 1% of any other cell type (Fig. 6b; Supplementary Table 6). Cells identified as CD8+ or CD4 + T cells largely overlapped with *CD3E* expressing cells (Fig. 6a). Similarly, for all 5 samples, cells identified as expressing productive V(D)J transcripts (Fig. 6c) overlapped almost completely with those identified as *CD3E*-positive (Fig. 6a) and as CD4/CD8 T cells (Fig. 6b).

For melanoma PBMCs, cells corresponding to the single most dominant V(D)J clonotypes largely overlapped with CD8 + T cells (Fig. 6b, d; Supplementary Fig. 13). For the T zone lymphoma, the vast majority of cells correspond to a single dominant clonotype, which also was almost entirely CD8 + T cells (Fig. 6b, d; Supplementary Fig. 13). This correlated with the CD8+ lymphocytosis immunophenotype seen on the commercial flow cytometry results, Supplementary Table 2, and with the immunophenotypes that have been described for T zone lymphoma in dogs[27]. A monoclonal T cell population was also seen in the PTCL lymph node sample scTCR data but scRNA sequencing and flow cytometry were not done in this case.

We next sought to determine the phenotype of expanded compared to non-expanded T cells for melanoma PBMCs. Differential expression analysis showed that expanded CD8 + T cells had significantly increased expression of known T cell activation markers *GZMA, GZMK,* or *CD38* (adjusted p-value < 0.05) and significantly decreased expression of exhaustion-related markers *CTLA4, TOX,* or *NFATC1* compared to non-expanded CD8 + T cells (adjusted p value < 0.05) (Fig. 7, Supplementary Fig. 14, Supplementary Data 8). The inverse pattern (significantly decreased activation or increased exhaustion) was not observed. Additionally, we noted decreased expression of *TCF7*, encoding for *TCF1* protein, which when co-expressed with *TOX* supports a progenitor exhausted fate, and *TCF7* is subsequently downregulated upon terminal exhaustion[28].

We next sought to characterize the expanded and non-expanded (CD4/CD8) cell populations by memory status. We visualized the same t-SNE projections with *CD4, CD8,* and markers of effector memory (*CCL5, ZEB2, GZMK*) or naive status (*LEF1, TCF7, CCR7*) (Supplementary Fig. 15). Individual *CD4* and *CD8* marker expression was largely concordant with singleR cell typing. For the melanoma samples, the expanded (dominant clone) CD8+ population mainly expressed effector memory markers, whereas the non-expanded population predominantly expressed naive markers. In contrast, for the T zone lymphoma sample, the expanded population mainly expressed naive markers. This hints that the malignant CD8 + T cells expanded but never matured.

We next attempted to determine why a subset of non-expanded CD8+ cells seemed to co-cluster with CD4+ cells (Fig. 6b). Supplementary Figure 16 shows that the subset of CD8 + T-cells co-clustering with the CD4+ population primarily expressed naive markers (*LEF1, TCF7, CCR7*). A portion of this CD8+ subset was also CD4 +. Hence, co-clustering of CD4+ and CD8+ populations can be explained by at least two independent phenomena. First, it marks the existence of double positive (CD4 + / CD8 +) T-cells. Second, it demonstrates CD4+ and CD8+ populations with shared naive status drives clustering more than CD4 vs. CD8 lineage.

## Discussion
The biomedical community increasingly recognizes the value of companion dogs as a model system for human cancer. However, there remains a lack of the necessary reagents and methods for molecular profiling of canine clinical samples. Here we define a protocol for single cell TCR sequencing that should enhance the utility of canine samples as they relate to the study of cancer and disease. In this work we were able to successfully adapt existing protocols to perform single cell TCR profiling

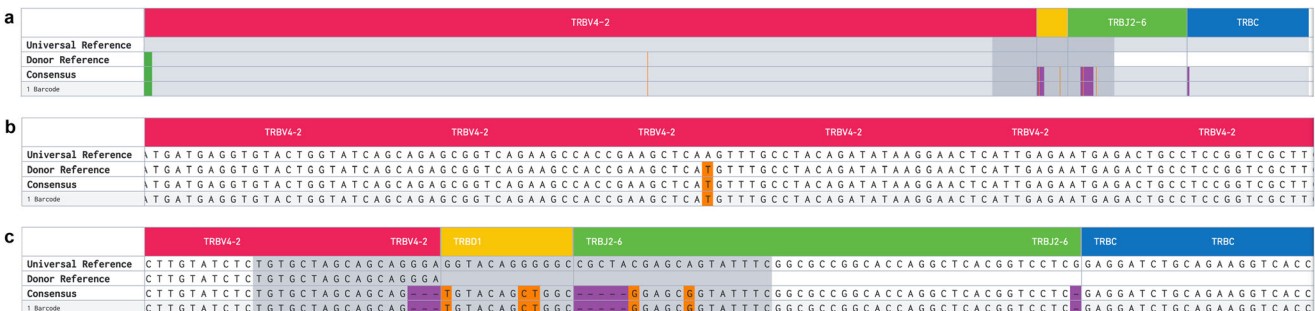

**Fig. 5 | Example individual clonotype.** Individual cell-specific TRA and TRB clonotypes are resolved to the nucleotide level. Illustrated here is a single such TRB clonotype from Melanoma_B (CDR3: CASSSVQLAERYF). **a** A specific VDJ recombination with complete TRBV4-2, TRBD1, and TRBJ2-6 and a portion of TRBC is depicted. The Universal Reference (based on IMGT V(D)JC reference sequences) is shown in the first row. Germline variants in the analyzed sample, relative to the Universal Reference, are depicted in the Donor Reference line. The germline/donor sequence is inferred by the cellranger software, by determining shared sequence between multiple clonotypes, in different cells, that use the same gene segment. The Consensus row and subclonotype row(s) show additional variants from the Donor and Universal Reference that were presumably introduced during joining of gene segments. The consensus represents the sequence of the first exact subclonotype for a receptor chain within the clonotype. In this case, only a single subclonotype for a single barcode is shown. But, in other cases, multiple subclonotypes may be grouped together, each represented by one or more cell barcodes. Subclonotypes may have small nucleotide differences or share a TRA chain but have missing TRB chain or vice versa. Single nucleotide changes are shown in orange and small deletions shown in purple. **b** and **c** are zoomed into the base pair level to visualize the variants.

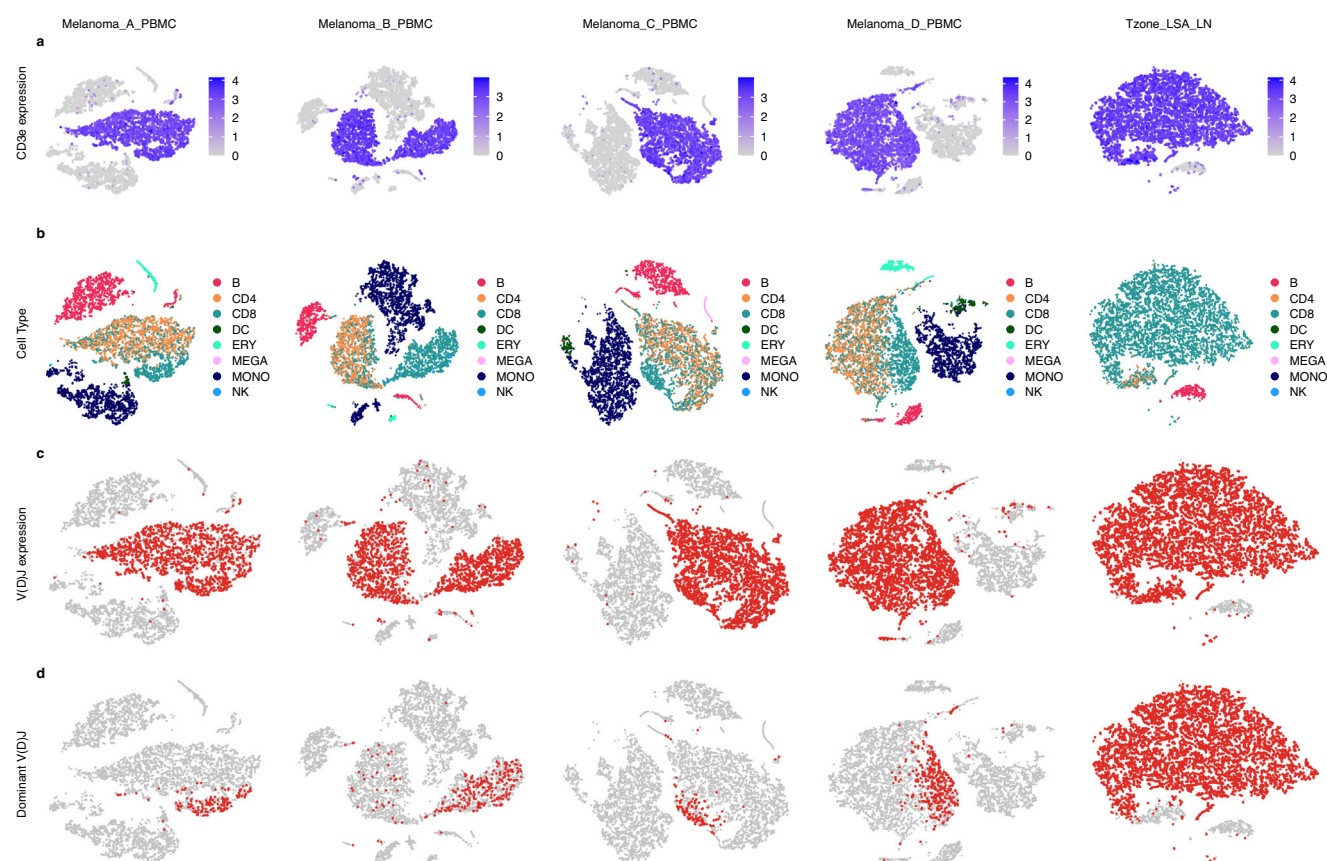

**Fig. 6 | Gene expression based t-SNE clustering for dogs with melanoma or T zone lymphoma with cells annotated by CD3E expression, inferred cell type or V(D)J transcript detection.** t-SNE clustering based on global gene expression patterns identified a number of distinct clusters of cells from Melanoma_A, Melanoma_B, Melanoma_C, and Melanoma_D PBMCs and T zone lymphoma lymph node. Genes present in fewer than 10 cells and cells with fewer than 100 genes were filtered out of the dataset. Cells whose mitochondrial expression was in the top 5% across all cells were filtered out and DoubletFinder was used to identify and filter out expected doublets. **a** A T cell marker (CD3E; colored gray to blue) identifies several large clusters. **b** Cell types, inferred based on published expression signatures of blood cell types, identified CD4 (orange) and CD8 (teal) T cell clusters largely overlapping with the CD3E-positive clusters identified in (**a**) as well as large monocyte (dark blue) and B cell (red) clusters and smaller clusters of several other cell types. Cell types without an assignment or with a population of less than 1% of all cells identified were excluded. **c** Cells identified with V(D)J rearrangements overlap strongly with those identified as CD4/CD8 T cells in (**b**) or CD3E-positive shown in (**a**). **d** Cells corresponding to the single most dominant clonotype largely cluster together in the CD8 T cell clusters shown in (**b**).

**Fig. 7 | Expression of T cell activation and exhaustion markers in expanded clonotypes versus non-expanded clonotypes for Melanoma_B_PBMC.** Heatmap of single cell expression values ($\log_e(x + 1)$ normalized and scaled for all cells in the sample) for expanded CD8 + T cells vs non-expanded CD8 + T cells for known markers of T cell activation and exhaustion. Expanded cells are those with a clonotype frequency greater than 1%. Marker gene names are colored blue if their expression is significantly increased or red if significantly decreased in expanded vs non-expanded CD8 + T cells (adjusted $p$ value < 0.05) for this dog sample (Melanoma_B_PBMC) (Supplementary Data 8).

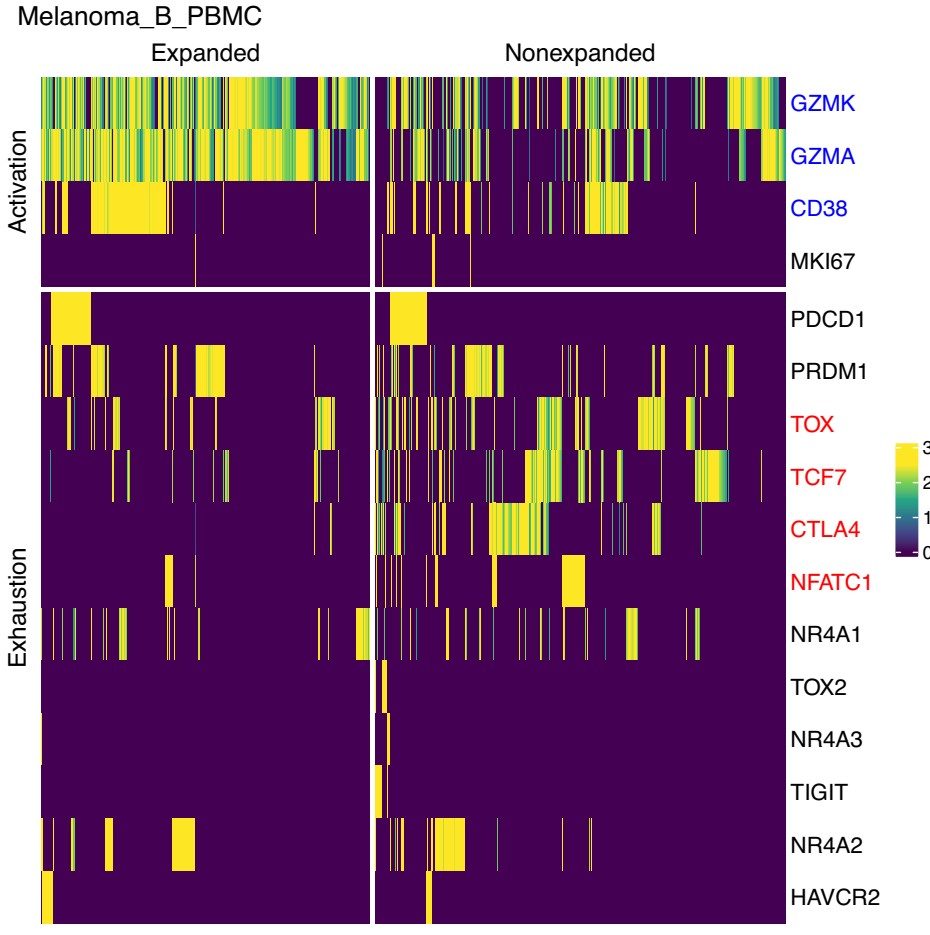

for 16 samples from 11 individual companion dogs. We were able to detect nearly all known functional V(D)J gene segments for both α and β TCR chains. We also identify several pseudogene or non-functional ORF segments which could be annotated as functional. These are at least in part related to a large amount of germline variation observed in our data, not currently represented in the reference database for dogs. Integration with single cell transcriptome sequencing for 5 samples demonstrated the expected relationship between cells expressing rearranged V(D)J sequences and T cell markers or cell type inferred from global gene expression patterns. A spectrum of T cell clonality was observed in the samples, from monoclonal (T cell lymphoma), to oligoclonal (melanoma) or polyclonal (healthy normal) (Fig. 4). There was slightly more diversity in healthy lymph nodes than healthy PBMCs but both showed near complete absence of clonal expansion. The PBMC samples from dogs with melanoma showed strong evidence of dominant clonotypes. Furthermore, cells classified as CD8 + T cells with expanded clonotypes showed consistently increased expression of known activation markers and decreased expression of exhaustion markers. As these samples were collected from melanoma patients undergoing active immunotherapy treatments, taken together, these patterns might indicate active T cell populations responding to their tumors. However, such conclusions await further immune studies of these canine patient samples.

Overall, we observed a broad representation of the known canine V(D)J repertoire. We detected all the known functional TRBV/TRBJ gene segments, all functional TRAJ and most, but not all, functional TRAV gene segments (Fig. 2, Supplementary Fig. 5). D gene segments are difficult to determine due to their small size and the introduction of junction recombination mutations that make mapping to reference D gene sequences difficult, therefore we did not report on their detection. The vast majority of the missing V/J gene segments were pseudogenes (IMGT

accessed 15-Aug-2022)[29]. The only missing V/J gene segments which were predicted to be functional on IMGT were TRAV8-1, TRAV9-4, TRAV19, TRAV23, and TRAV43-3. In theory, all V, D, or J gene segments may participate in V(D)J rearrangement and form functional TCRs. However, there are a number of plausible explanations for these unobserved gene segments. The unobserved gene segments may simply be the result of insufficient sampling and could be the reason all of the TRBV gene segments ($n = 36$) were observed but some of the larger number of TRAV segments ($n = 57$) were not present in the samples. Despite the very comprehensive sequencing performed, only 77,809 total cells were sequenced. Furthermore, several of the samples showed evidence of dominant clonotypes. As a result, a substantial fraction of cells and sequence reads were associated with these few dominant clonotypes, decreasing our power to detect rare clonotypes, and increasing the chance that some gene segments might be missed due to insufficient sampling. In human BCR repertoires it has been suggested that V(D)J gene segment usage is not random with some gene segments, segment combinations, and CDR3 lengths preferentially used and others rarely used[30–35]. The same could also be true in dog TCR repertoires.

A prior study querying whole genome sequences of 19 dog breeds reported block duplications of TRAV9, TRAV43, and TRAV13 genes in the C-distal end of the TRA locus[29]. TRAV43-1, TRAV9-6, TRAV43-4, TRAV12, and other TRAV9 genes were the most commonly used TRAV genes found in our dataset. Thus abundant usage of TRAV43 and TRAV9 gene families may be a natural consequence of these duplications. Interestingly, TRAV9 and TRAV43 were also reported as the most commonly expressed V gene subgroup in cats[36]. Overall, the most frequently observed TRA and TRB genes in this study were also consistent with those reported as highly utilized in sorted blood T cells of healthy experimental dogs by Eschke et al.[19].

A number of gene segments annotated as pseudogene or non-functional ORF were also detected, often in many cells and across multiple dogs (see Results for details). TRBV19 was identified as a pseudogene in IMGT due to an in-frame stop codon in FR3-IMGT but a functional CDR3 was predicted in combination with all functional TRBJ gene segments in a single dog. TRBV19 is functional or has an ORF in numerous other species including humans and the domestic cat[37]. We showed that in at least one dog a germline SNP corrects the stop, making this pseudogene functional. In another case, TRBJ1-3, another pseudogene in IMGT with an in-frame stop codon in the J-region, also produced a functional CDR3 in combination with all TRBV gene segments, and for all dogs. In this case, the stop codon was very near the beginning of the J gene and random changes introduced during recombination appear to frequently correct/obviate the stop. TRAV9-2, another pseudogene in IMGT due to a frameshift in FR2-IMGT and in-frame stop codon in FR1-IMGT also produced productive CDR3s with many different TRAJ gene segments and in all dogs but one. In this case, extensive but consistent germline differences were observed between the reference and all dogs in this study using the gene segment (as well as other wolf and dog reference sequences) which corrected both the stop and frameshift. This observation suggests that the commonly used CanFam3.1/CanFam6 Boxer reference has either a very distinct/unique version of this gene segment or there are errors in its assembly. Taken together these results suggest the re-annotation of at least two (TRAV9-2 and TRBJ1-3) and possibly a third (TRBV19) pseudogene segment as functional. A number of non-functional ORFs (especially TRAJ52, TRBV15 and TRBJ1-4) should also be re-evaluated with whole genome data based on the frequency of their usage in productive TCRs in our data. Most of these were also observed by Eschke et al.[19].

In addition to missing some known segments and observing productive usage of pseudogene/non-functional segments, it is also possible that some completely unannotated gene segments were missed. Currently the IMGT database effectively has only a single reference sequence for each known V(D)J gene segment, based almost entirely on the CanFam3.1 reference sequence. It is possible that some as yet unknown gene segments are entirely missing due to incompleteness of this reference. Even more likely is the absence of alternate alleles for known gene segments. We identified a large number of possible alternate alleles in TRAV and TRBV genes with cell-ranger. However, these sequences have not been validated to a level sufficient for submission to IMGT. Modern canine genome assemblies[38], future annotation work, and comprehensive germline analysis of data like ours should allow identification of novel gene segments and alleles in the future.

The methods developed in this study could be immediately applied to current canine immunotherapies studies to help identify predictors of response and evaluate candidate immunotherapeutic targets and efficacy of immune stimulating protocols. It should be noted that the protocols described in this paper, as with human and mouse protocols, are limited to 5'-based 10x (v1 or v2) protocols. They can not be easily adapted to 3' GEX (v1, v2 or v3), Multiome, or other 3'-based 10x approaches. Interpretation of this data may also benefit from advanced tools for TCR sequence analysis such as improved TCR clonotype merging not assessed here[39–41]. We observed improved TRA/TRB pairing using 10x v2 5' kits compared to v1 kits with less TRA/TRB only clonotypes and percentages of extra TRA/TRB clonotypes more in line with expectations for dual TCR expression[24]. Therefore we recommend using v2 kits if possible. Future work should include adapting the protocols herein to be applicable for additional TCR loci (e.g., TRD and TRG) as well as B-cell receptor (BCR) sequencing. The ability to perform BCR sequencing would further increase the utility of canine samples, allowing for a more complete understanding of adaptive immunity in cancer, auto-immune diseases and B-cell malignancies.

## Methods
### Sample collection
Collection of samples at the University of Missouri was approved by the ACUC under protocol #30721. We have complied with all relevant ethical regulations for animal use.

### Normal samples and single cell suspension
To demonstrate the TCR repertoire in PBMCs and lymph nodes of normal client-owned dogs we identified 5 dogs (Normal_A, Normal_B, Normal_C, Normal_D, and Normal_E) that were clinically healthy (Table 1). These dogs did not have any masses on physical exam and did not have a history of or current inflammatory or immune mediated disease. These dogs were also not vaccinated within 6 months of sample collection and were not on immune modulating drugs. The normal dogs were followed for >309 days (as of 5 Dec 2022) and remained clinically healthy. Ficoll-separated peripheral blood mononuclear cells (PBMCs) were obtained from each dog and cryopreserved. Aspirates were collected from the mandibular lymph nodes of each dog at the same time as peripheral blood and cryopreserved. Aliquots from each sample were analyzed via flow cytometry, cytology and hematology. Complete clinical information and the results from a CBC and lymph node cytology are included in Supplementary Table 2. Cryopreserved PBMCs or aspirates were thawed and a single cell suspension of approximately 1000/uL was generated according to the 10x Genomics Demonstrated Protocol (CG00039 Rev D). Viability was assessed by trypan blue exclusion staining and/or flow cytometry.

### Lymphoma samples and single cell suspension
To demonstrate the ability of scTCR-seq to identify expected T cell clonality, the TCR repertoire from lymph node aspirates from 2 Golden Retrievers with T cell lymphoma, one indolent (T zone) and one aggressive (PTCL), were evaluated (Table 1; See Supplementary Table 3 for complete clinical information). Aspirates were prepared for single cell sequencing as described above.

### Melanoma samples and single cell suspension
To demonstrate applicability of canine TCR profiling in a cancer immunotherapy setting we identified 4 individual dogs (Melanoma_A, Melanoma_B, Melanoma_C, and Melanoma_D) with metastatic melanoma from an ongoing immune therapy trial (Table 1; see Supplementary Table 4 for complete clinical information). All dogs were treated with autologous deglycosylated vaccines derived from primary and/or metastatic tumor cultures and showed evidence of clinical response. Melanoma_A had a progression free interval of 280 days after the first dose of the autologous vaccine. Melanoma_B had resolution of progressive pulmonary nodules following the vaccine series and survived more than a year without clinical recurrence. Melanoma_C and Melanoma_D were euthanized for non-melanoma related reasons >1205 days from diagnosis and were free from melanoma on necropsy. PBMCs were obtained from each dog and cryopreserved. PBMCs for Melanoma_A were collected 1355 days after the initial autologous vaccine treatment, in the presence of lymph node metastasis and at the time of the first tyrosinase vaccine treatment (Oncept). PBMCs for Melanoma_B and Melanoma_D were collected 14 days after the initial autologous vaccine treatment. Melanoma_B, Melanoma_C and Melanoma_D were free of macroscopic disease at the time of blood draw. PBMCs for Melanoma_C were collected 30 days after the initial autologous vaccine treatment. PBMCs were prepared for single cell sequencing as described above.

### Flow cytometry
The proportion of cells expressing the TCRα/β chains was confirmed by flow cytometry for normal PBMCs. PBMCs were stained with an antibody against the pan canine T cell marker CD5 (BioRad) and an antibody against either TCRα/β, or TCRγ/δ (Washington State University). Phenotypic analysis of normal PBMCs and cells from lymph node aspirates of the T zone lymphoma was performed using commercially available dog-reactive antibodies (Supplementary Table 5) to identify the proportion of B cells (CD21+), T cells (CD5+), T helper (CD5+ CD4+) and cytotoxic (CD5+ CD8+) T cells, and T regulatory cells (CD4+ (CD25+FoxP3+)). Cells were analyzed using a Beckman Coulter Fortessa X-20 flow cytometer and FACSDiva 8.0 Software.

## Primer design

Primer design and application (Fig. 1; Supplementary Fig. 1) was modeled from the Chromium Single Cell V(D)J Reagent Kits User Guide (CG000086 Rev L) for v1 and Chromium Next GEM Single Cell 5' Reagent Kits v2 (Dual Index) User Guide (CG000331 Rev C). Note, only 10x 5' protocols are amenable to V(D)J amplification and sequencing with single cell resolution. This protocol uses a nested PCR design. The forward primers from the 10x protocols were left unchanged for v1 and v2 kits respectively. In the first round, the forward primer (TRA/TRB Forward 1) primes off the Illumina read 1 (R1) sequencing adapter sequence that is incorporated during generation of 10x barcoded, full-length cDNA from polyadenylated mRNA. In the second round, the forward primer (TRA/TRB Forward 2) primes off the same R1 sequence as previously. The forward primers correspond to the 5' end of 10x cDNA fragments. In the case of full length TCR cDNA this would represent the V gene segment end of the cDNA fragment. The forward primers do not have any specificity for the TCR which comes entirely from the reverse primers. The first reverse primer (TRA/TRB Reverse 1, Outer) primes at the constant (C) region gene segment. The second reverse primer (TRA/TRB Reverse 2, Inner) similarly primes at the C region at an inner, 5' position relative to the outer primer. In order to design appropriate 3' reverse primers for use with canine cells we first constructed (in silico) a complete reference V(D)JC TCR cDNA sequence along with 10x adapter sequences for both α (TRA) and β (TRB) chains. The α chain was based on a representative dog TCR α rearranged partial mRNA (GenBank: M97511.1). The closest V gene segment to this partial mRNA was determined by blast alignment against canine V gene sequences in the IMGT database. The constructed cDNA was extended to include (from 3' to 5') the Illumina R1 adapter, 16 nucleotide (nt; 16 x N) 10x cell barcode, 10 nt (10 x N) unique molecular identifier (UMI), 13 nt template switch oligo (TSO), complete V gene segment, complete J gene segment, and complete C gene segment. The β chain was based on a representative dog TCR β rearranged partial mRNA (GenBank: HE653957.1). The closest V and C gene segments to this partial mRNA were determined by blast alignment against canine V and C gene sequences in the IMGT database. The constructed cDNA was extended to include (from 3' to 5') the R1 adapter, 10x cell barcode, UMI, TSO, V, D, J, and C gene segments as described above. These constructed cDNA sequences were then used as input to primer3plus (4.0)[42,43], with forward primers provided as described above, and a target region for reverse primer design specified in the C region. The product of the first (outer) design was used as input for the second (inner) design. Primer oligonucleotides were ordered from Integrated DNA Technologies (Coralville, IA). Final primer sequences for both v1 and v2 kits are included in Supplementary Table 1.

## cDNA generation

cDNA generation was performed according to the Chromium Single Cell V(D)J Reagent Kits User Guide (CG000086 Rev L) or Chromium Next GEM Single Cell 5' Reagent Kits v2 (Dual Index) User Guide (CG000331 Rev C), with the exception of the TCR amplification step (described below). Briefly, cells were suspended in phosphate-buffered saline containing 0.4% bovine serum albumin at approximately 1000/uL. The appropriate volume for a targeted cell recovery of 10,000 (taken from 10x manual) was used for Gel Bead-in-Emulsion (GEM) generation, barcoding, post-GEM clean-up, and cDNA amplification. Quality controls were performed using the Agilent Bioanalyzer high sensitivity chip. 2 uL of cDNA were used to separately amplify TCR α and β chains using custom dog-specific primers (see Primer design; TCR amplification). The remaining cDNA were set-aside for gene expression (scRNA) profiling.

## TCR amplification

In order to amplify dog TCR sequences, the nested PCR approach utilized for human and mouse protocols was adopted, with dog-specific reverse primers located in the constant region of the TCR cDNA (see Primer design; Supplementary Table 1). Briefly, 2 uL of cDNA were amplified in a Mastercycler Gradient instrument (Eppendorf) in 100 uL total reaction volume using Phusion high fidelity polymerase (Thermo Fisher). The first reaction consisted of an initial denaturation step (98 °C for 45 s) followed by 12 cycles of 98 °C for 20 s, 65 °C for 30 s, and 72 °C for 60 s. An additional extension step (72 °C for 60 s) was added at the end. The PCR product was processed using the double-sided SPRI bead purification protocol described in the 10x Genomics user manual for single cell 5' reagent kits v2, CG000331 Rev C. The resulting product was amplified in the second reaction. Amplification conditions were identical except for the annealing temperature which was 62 °C. The amplification product was purified using SPRI beads before quality control on the Agilent Bioanalyzer high sensitivity chip and further processing for library generation.

## Library generation

TCR enriched PCR products and set-aside scRNA cDNAs underwent library construction and final quality control as described in the Chromium Single Cell V(D)J Reagent Kits User Guide (CG000086 Rev L) or Chromium Next GEM Single Cell 5' Reagent Kits v2 (Dual Index) User Guide (CG000331 Rev C).

## Sequencing

scRNA and TCR VDJ enriched libraries, produced by the MU Core (as described above), were shipped to the McDonnell Genome Institute for sequencing or sequenced at the University of Missouri Genomics Technology Core. The concentration of each 10x single cell and V(D)J library was determined through qPCR (Kapa Biosystems) to produce cluster counts appropriate for the NovaSeq 6000 platform. Approximately 500 M read pairs were targeted for each gene expression library and 25 M read pairs were targeted for each V(D)J library. The libraries were sequenced on the S4 300 cycle kit flow cell (2×151 paired end reads) using the XP workflow as outlined by Illumina.

## Data pre-processing and analysis

All data processing and subsequent analysis made use of Cell Ranger 5.0.1 unless otherwise noted. Raw fastq files were generated from Illumina instrument data using *cellranger mkfastq*. A canine-specific reference package was created using *cellranger mkref* with CanFam3.1 (INSDC Assembly GCA_000002285.2, Sep 2011) canine reference genome and *cellranger mkgtf* filtered CanFam3.1 GTF from Ensembl v102 as input. The GTF file was filtered according to the example provided in the documentation on the 10x Genomics site[44]. A canine-specific V(D)J reference package was created using the *fetch-imgt* script and *cellranger mkvdjref*. The *fetch-imgt* script extracts V(D)JC reference sequences from IMGT which are based almost entirely on the CanFam3.1 reference except for a handful of TRBJ segments which are based on genbank accession HE653929. The *fetch-imgt* script failed to obtain C-REGION sequences causing errors in subsequent steps. These sequences (IMGT000004 | TRAC*01, IMGT000005 | TRBC1*01, HE653929 | TRBC1*02, and HE653929 | TRBC2*01) were manually retrieved from IMGT[45], artificially spliced, and added to the *fetch-imgt* file before completing V(D)J reference generation (Supplementary Data 9). Single cell feature counts were generated with *cellranger count*. Counts were combined, normalized, and batch corrected across multiple libraries with *cellranger aggregate*. Single cell VDJ sequence assembly and paired clonotype calling were performed with *cellranger vdj*. Each unique clonotype is defined by *cellranger* as one or more TRA and/or one or more TRB resolved CDR3 sequences which are expressed in one or more cells per sample. This definition of clonotype is used throughout the paper. V(D)J gene segment annotations (functional, non-functional ORF, pseudogene) were obtained from IMGT (Supplementary Data 5, accessed 20-Mar-2023). Genes present in fewer than 10 cells and cells with fewer than 100 genes were filtered out of the dataset. Cells whose mitochondrial expression was in the top 5% across all cells were filtered out and DoubletFinder[46] (version 2.0.3) was used to identify and filter out expected doublets. DoubletFinder was run with parameters derived

using the example provided at https://github.com/chris-mcginnis-ucsf/DoubletFinder. Additional ad hoc analysis and figure generation were performed in the Loupe Browser (version 5.0.1), Loupe VDJ Browser (version 5.0.0) and Seurat[47] (version 4.3.0). Sequence analysis of observed pseudogene and non-functional ORF gene segments was performed using the IMGT database, Loupe VDJ Browser, BLAST, UCSC Genome Browser, Blat, Clustal Omega, and Expasy Translate. Gene segment usage, clonotype distribution, and gene length analyses were performed using built-in math functions in R, and associated figures were generated using ggplot2 (version 3.4.2), data.table (version 1.14.6), viridis (version 0.6.3), gtools (verion 3.9.2), forcats (version 0.5.2), dplyr (version 1.1.2), plyr (version 1.8.8), and cowplot (1.1.1).

## Cell typing
Cells were annotated to cell types with SingleR (version 1.0)[48] using expression profiles derived from the DMAP dataset available on Haemopedia, a hematopoiesis cell expression database[49]. The DMAP dataset was converted from Affymetrix probes to human gene IDs and then to dog gene names using the probe to human gene name mapping available on Haemopedia and the human to dog gene mapping acquired from Ensembl v102 BioMart. If different probes mapped to the same human gene, the probe with the highest coefficient of variance across all samples in the dataset was kept and all other probes that mapped to that gene were discarded. If a probe entry mapped to multiple human genes then that probe-to-gene mapping was duplicated for each gene it mapped to. For the human gene ID to dog gene name conversion all many-to-many mappings were removed, mappings that had multiple human genes to a single dog gene were removed, and all the mappings from one human gene to multiple dog genes were duplicated for each dog gene. Cell types were manually simplified to B cell (B), early B-cell (PreB), basophil (BASO), CD4 + T cell (CD4), CD8 + T cell (CD8), common myeloid progenitor (CMP), dendritic cell (DC), eosinophil (EOS), erythroid (ERY), granulocyte/monocyte progenitor (GMP), granulocyte (GRAN), hematopoietic stem cell (HSC), megakaryocyte (MEGA), megakaryocyte/erythroid progenitor (MEP), monocyte (MONO), and NK cell (NK).

## Identification of markers of T cell phenotype
A set of known human T cell activation (n = 8) and exhaustion (n = 14) markers was identified from the literature[28,50–53] (Supplementary Data 10). Of these, 4 activation (CD38, GZMA, GZMK, MKI67) and 12 exhaustion (CTLA4, HAVCR2, NFATC1, NR4A1, NR4A2, NR4A3, PDCD1, PRDM1, TCF7, TIGIT, TOX, TOX2) could be reliably mapped to dog orthologs (Ensembl 102, one-to-one). Similarly, a set of known T cell effector memory (n = 5) and naive (n = 4) status markers were identified[19] (Supplementary Data 10). Of these, 3 effector memory (CCL5, ZEB2, GZMK) and 4 naive (LEF1, TCF7, CCR7, SELL) could be reliably mapped to dog orthologs (Ensembl 102, one-to-one). SELL was ultimately excluded due to non-specific expression leaving 3 markers of effector memory and 3 markers of naive status.

## Differential gene expression analysis of exhaustion/activation status of expanded versus non-expanded CD8 + T cells
CD8+ T cells (see Cell typing) were categorized as expanded if they had a clonotype represented in >1% of all cells. Otherwise they were considered non-expanded. Differential expression was tested for all 16 exhaustion/activation markers, using the Seurat package, with a Wilcoxon Ranked Sum test (two-sided). A marker was considered significant if the adjusted p-value (Bonferroni correction) was < 0.05.

## Statistics and reproducibility
Single cell VDJ sequencing was conducted on 16 samples (5 healthy PBMCs, 5 healthy lymph node aspirates, 4 melanoma PBMCs, and 2 lymphoma lymph node aspirates). Single cell RNA sequencing was conducted on 5 samples (4 melanoma PBMCs and 1 T zone lymphoma lymph node aspirate). All primary data processing and analysis was performed with Cell Ranger 5.0.1. Additional ad hoc analysis was performed with the R statistical programming language built-in math functions and additional packages including DoubletFinder46 (version 2.0.3), Seurat (version 4.3.0), ggplot2 (version 3.4.2), data.table (version 1.14.6), viridis (version 0.6.3), gtools (verion 3.9.2), forcats (version 0.5.2), dplyr (version 1.1.2), plyr (version 1.8.8), and cowplot (1.1.1). Differential gene expression analysis of exhaustion/activation status of expanded versus non-expanded CD8 + T cells was conducted with a two-sided Wilcoxon Ranked Sum test. 16 marker genes were tested for differential expression, and a marker was considered significantly upregulated/downregulated if the adjusted p-value (Bonferroni correction) was < 0.05. Additional details are provided above for each individual Methods section.

## Reporting summary
Further information on research design is available in the Nature Portfolio Reporting Summary linked to this article.

## Data availability
All raw scRNAseq and scTCRseq data have been deposited with BioProject: PRJNA742469 and SRA:SRP326193. Custom canine-specific reference files (CanFam3.1, Ensembl v102, see Methods) for use with cellranger count and cellranger vdj are available at http://genomedata.org/10X_canine_ref/.

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

## Acknowledgements

This research was supported by the Alvin J. Siteman Cancer Center Siteman Investment Program (supported by The Foundation for Barnes-Jewish Hospital, Cancer Frontier Fund and Barnard Trust) and jointly funded by the Ellis Fischel Cancer Center, University of Missouri. This work was also supported by the University of Missouri Research and Creative Works Strategic Investment Program. Malachi Griffith was supported by the V Foundation for Cancer Research under Award Number V2018-007 and by the NCI under Award Number U01CA248235.

## Author contributions

Z.L.S., H.R., M.H., M.G., J.N.B. and O.L.G. designed the study. H.R., C.F., R.F., M.Z. and N.J.B. completed laboratory experiments and sequence data production. Z.L.S., H.R., M.H., S.C., B.F., J.A.F., M.G., J.N.B. and O.L.G. performed data analysis and interpretation. Z.L.S., H.R., M.H., S.C., B.F., J.A.F., C.F., M.Z., N.J.B., M.G., J.N.B. and O.L.G. helped draft the manuscript. C.N.R., T.A.F., M.G., J.N.B. and O.L.G. supervised the study.

## Competing interests

The authors declare no competing interests.
