## [Peer Review File · Communications Biology]

Reviewers' comments:

Reviewer #1 (Remarks to the Author):

Skidmore and colleagues present a successful modification of the 10x GE + VDJ platform to the sequencing of alpha/beta TCRs from canines. They design canine specific reverse primers for the TCR cDNA enrichment step and apply these to two animals who have undergone cancer immunotherapy. The manuscript demonstrates the successful amplification of TRA and TRB chains and that the cells carrying these chains fall within the T cell populations by the GE analysis. The modifications to the assay are thoroughly and clearly presented. All supporting data is provided or deposited in an appropriate repository.

Overall, it is a robust report of a methodology that is likely to be of interest to those working with canines, both as model animals and within the veterinary field, and is an important contribution to increasing the availability of species-specific assays. The limitation of the manuscript is that it doesn't extend to a detailed characterisation of the assay outputs.

Major comments:

As a proof of concept for the method, the manuscript is excellent. It is well written and clearly described. The ability of the method to produce scTCR-seq data is well supported by the data provided. Methods are well described. Key results are clearly reported.

The manuscript is, however, largely limited to the reporting of the methodology and there is limited analysis of the outputs from the application of the method. The claim in the Abstract, repeated in the Discussion, that the dominant clonal expansions suggest the presence of an anti-tumour population are not supported by the Results in the current form. Given that the scTCR-seq is from PBMCs and that these clonotypes appear to map to CD8+ populations, they could represent memory T cell populations. In the absence of any attempt to characterise the cells using the GE to determine if they do represent an "active T-cell population" the results presented only show that they tend to be CD8+ and nothing about anti-tumour response. This statement should be removed. Removal doesn't impact on the quality of the manuscript in terms of reporting a new method.

In the Results section, sub-heading "Single cell V(D)J repertoire" in the final sentence, there is mention of somatic hypermutation (SHM) being observed. TCRs are not subject the SHM, this is unique to the immunoglobulin loci. It is also no clear what "recombination-related mutations at V(D)J" are. The elements described in Figure 6 are explained by the mentioned germline variation and V(D)J recombination diversity.

With regards to "germline variation" was any attempt made to determine TRA/B genes that were missing from the IMGT reference directory? For example, in Figure 6, a proposed germline variant (or polymorphism/allele) for TRBV4-2 is shown. Was this variation present in all TRBV4-2 from this dog? If

so, this would mean this dog likely carries a version of TRBV4-2 that isn't currently captured in the IMGT reference directory. There is mention of how many of the genes from IMGT are observed, but the flip side, i.e. how many novel genes from the two animals weren't in IMGT, isn't considered. Similarly in Figure 6, there is also potential for an unreported TRBJ gene. This isn't required for a report on the method, but the approach does permit the identification of this type of germline variation that can contribute to more complete germline gene reference directories.

Minor comments:

- Please include line numbers to assist with review process.
- Table 1 – given that the forward primers are the standard 10x VDJ kit primers it would provide clarity if they were named as such in the Table, or only report the novel TCRA/B primers designed for the study here.
- Table 2 – what is the percentage of cells that have paired TRA/B from each animal?
- Figure 4/S3 – legend label, count rather than frequency? Is it count for cell barcodes or TRA/B contigs?
- Figure 5/S4 – why not include both animals in a single figure?
- Figure 6 – is the 'universal reference' for the TRA/B the customised IMGT-derived reference set used for cellranger vdj rather than the genomic reference (CanFam3.1) used for the gene expression analysis?
- Figure 7/S5
 - o A) doesn't show the overlap, just shows the CD3E expression, by comparing A and C the overlap is indicated.
 - o B) Why not remove cells from the TSNE that were filtered due to mito/doublets/<5 cells in the cluster rather than showing them in gray?
 - o B) CD4/8 colours are hard to delineate. Is it possible to change them to less similar colours?
 - o D) Could you quantify the number of dominant clone cells in each cluster as a bar chart or similar?
 - o For future, could consider integrating the two animals into a single analysis?
- CDR3 length distributions might be of interest to demonstrate that you are capturing transcripts of diverse lengths.
- For the supplemental tables, animal names rather than Dog_A/B are used in the tab names (Rose/Geddy).
- Supplemental Data 1 (the VDJ reference for cell ranger) wasn't available from the manuscript submission system. Are there issues in distributing this as part of the manuscript due to IMGT licensing? Agree that if it is OK to include it should be. It is important to capture the reference used as repeating your well described process for generating the set in the future may result in a different set of reference genes due to changes in the IMGT references.
- Supplemental data 2 – 4 out of the top 5 paired clonotypes from Dog_A have two TRB chains reported for the clonotypes. Whereas, for Dog_B's expanded clonotypes there's only single TRB chains (as expected). Have you looked at why? Is this caused by doublets, stray RNA in the emulsions picking up barcodes, some quirk of the cellranger software?

Reviewer #2 (Remarks to the Author):

In this manuscript, the authors present single-cell TCR sequencing profile data for two different dogs under treatment with autologous deglycosylated vaccines derived from tumor cultures, using 10X technology and described the detected TCR sequences and their distribution. To my understanding, a novelty of this work is providing new primer sets to adapt 10X scTCR-seq framework for dog immune cell analysis. Nevertheless, I am not sure how significant the contribution of this work to the research field. If this work has significance in the aspect of methodology development, I suggest the authors highlight more on it.

As the authors mentioned, there is a value of companion dogs as a model system for human cancer. Then, they should conduct more extensive comparison between human and dogs in terms of TCR diversity. I believe such a comparative analysis will substantially increase the impact of this work. Currently, the manuscript is descriptive overall.

What are characteristics of T cell subsets enriched by dominant TCR clones? Does it similar to the canonical type of T cells that undergo activation and clonal expansion in human immunity?

Reviewers #3-4 (Remarks to the Author):

Skidmore et al. present a primer scheme for the amplification of canine TCR sequences from cDNA generated with 10x-Chromium 5' single-cell gene expression kits. Using this approach, they perform concurrent single-cell RNA and TCR sequencing on PBMC samples obtained from two dogs enrolled in immunotherapy trials for cancer. They demonstrate recovery of TCR sequences that map primarily to clusters of T cells.

As a technical report, this manuscript reports useful advances for the field though some revisions to strengthen and clarify the results are needed prior to publication. The manuscript could be strengthened by the addition of further quality control metrics, and by the addition of an integrated analysis of T cell phenotype and repertoire, which would illustrate the translational potential of such a methodology for profiling both the clonotypes and their gene expression profiles for the same cells.

Major points:

The authors should provide additional quality metrics obtained using their approach, such as:

- 1) What is the level of TCR sequence consensus produced by cellranger? Are individual reads for the same UMI in agreement with each other?
- 2) For expanded TRB sequences, is pairing with TRA sequences consistent from the same dog?
- 3) For expanded TRB/TRA sequences, is mapping consistently to either CD8+ or CD4+ T cells?
- 4) For cells with multiple UMI mapping to TRB/TRA, are these sequences in agreement with each other?

It would be expected that some fraction of sequences are artifacts generated by factors such as mispriming during the RT reaction or the generation of chimeric PCR byproducts.

These technical features would strengthen the understanding of the quality of the pairings recovered and the methods presented.

The authors should explicitly mention the level of agreement between the TRBV/TRAV/TRBJ/TRAJ sequences they obtain and those currently in canine reference genomes and on IMGT. The authors are correct that these references are incomplete for non-model species (i.e. human and mouse). The annotation of novel V/J segments and alleles in this species would be a valuable addition to the field. Are these gaps in the reference genome a factor in the limited coverage of TRBV gene segments observed in the data shown in Figure 4? It would appear to be consistent gaps across both dogs evaluated.

“As expected, individual clonotypes were characterized by evidence of germline variation, V(D)J recombination diversity, as well as somatic hypermutation and/or recombination-related mutations at V(D)J junctions (See Figure 6 for a representative example clonotype).”

The authors should quantify and present these factors rather than showing data for a single representative clonotype. Expanding the analyses in Figure 6 for a plurality of clonotypes would further address these factors and show further generalizability of the method.

Can the authors demonstrate a relationship between repertoire and phenotype – for example, do “dominant clonotypes” express greater levels of activation signatures than other clonotypes? This analysis would further motivate the simultaneous profiling of gene expression and TCR repertoire. The data presented in Figure 7 appear suited to consider these connections further.

Discussion: “Indeed, we observed that a number of the more frequent clonotypes could be merged based on CDR3 sequence similarity, even further emphasizing the dominant clonotype pattern (Suppl Table 2-3)”

It is not clear how “clonotypes were merged based on CDR3 sequence similarity” from Supplemental Tables 2 and 3. This is currently an active area of investigation in the field, with many proposed solutions (Dash et al Nature 2017, Huang et al Nature Biotechnology 2020, Zhang et al Clin Cancer Res 2020), none of which seemed to be cited or utilized here. Better justification for this statement needs to be provided and clarification on how the consolidation was performed.

Minor points:

The authors should mention how “dominant clonotypes” are defined.

The authors’ approach for TCR library preparation presented here relies on 5`-template switching, which is primarily compatible with 10x Chromium 5` v2 kits, not newer 10x v3 GE kits or other many other approaches for construction of 3`-barcoded libraries. This limitation should be explicitly acknowledged in the Discussion of the results to contextualize this work with other related studies in the literature in other systems (e.g., human, mouse, etc.). What is required to adapt this method to other library chemistries?

Figures 4 and 5 are not very informative. At the very least, these figures should be presented on a log-scale to better visualize the quantitative spectrum of their data. These could also be consolidated to one figure to provide space for additional analyses as noted above.

“As expected, individual clonotypes were characterized by evidence of germline variation, V(D)J recombination diversity, as well as somatic hypermutation and/or recombination-related mutations at V(D)J junctions (See Figure 6 for a representative example clonotype).”

It is unexpected that TCR repertoires would exhibit somatic hypermutation, as this process typically only occurs in B cells. Can the authors provide further evidence for this event? If not, it would make sense to revise the statement in accordance with the current understanding of T cell development.

Re: COMMSBIO-21-2112-T

First, we would like to thank the editor and reviewers for their careful reading and thoughtful suggestions. In response to this feedback, we have now completed a major revision that we believe substantially improves the manuscript and addresses their concerns. We will first briefly summarize the nature and extent of these revisions. Then we will address each comment individually. The original reviewers' comments are in *black italics*, our responses in plain blue text, and any new text added to the manuscript is in plain black text, highlighted yellow. The manuscript itself has been highlighted with changes in yellow as well. New figures, referred to in this response, are embedded in this document for convenience of the reviewers and also included in the revised submission.

Summary of response:

In response to the reviewers comments, extensive new data and analyses were generated. We updated the protocol from 10X 5' v1 to v2 to use the latest 10x forward primers and other reagents and show improved performance. We significantly expanded the set of individual dogs and samples profiled. Initially we piloted our scTCRseq approach on just two samples of PBMCs from dogs with melanoma on a vaccine trial. With this revision, we added two additional dogs from that trial. We also added paired lymph node biopsies and PBMCs from 5 healthy dogs (10 new normal samples), and lymph node biopsies from 2 dogs with T cell malignancies. These act as a kind of negative and positive control respectively, relative to the melanoma dogs, in terms of clonotype diversity. Overall we increased from 2 samples (2 dogs) to 16 samples (11 dogs) with scTCR data and 5 samples (5 dogs) with scRNA data. In total, we now profile at single cell resolution the TCR of 77,809 V(D)J-expressing cells (4,022 -previously), identifying 55,539 unique clonotypes (2,219 previously) with 3.0 billion sequence reads (128.7M previously). These additional data allow us to much more comprehensively survey the TCR repertoire of dogs, compare between healthy dogs and dogs with different kinds of malignancy, and assess the robustness of our protocol. We add new analysis related to TRA/TRB pairing. We now present our VJ gene segment usage in the context of their functional annotations (functional, non-functional ORF, pseudogene) and show overall very good sensitivity (nearly all known functional segments observed) and specificity (few non-functional/pseudogene segments observed). Where non-functional or pseudogene segments were observed to be expressed we performed extensive sequence analysis and show that several are likely to in fact be misannotated in the reference database. We added additional statistics and figures related to V(D)J sequence coverage/quality. We now analyze V(D)J contig length, CDR3 length and summarize germline differences across the cohort. Finally, we further characterize the phenotype of T cells with dominant clonotypes in PBMCs of

melanoma on the vaccine trial. All data are freely available at BioProject: PRJNA742469 and SRA:SRP326193.

The text of the abstract, introduction, methods, results, and discussion were all substantially supplemented or revised to reflect these new data and analyses. A new **Table 1** was added to provide an overview of the expanded set of individuals and samples profiled. **Table 2** (previously Table 1) describes custom dog TCR enrichment primers. **Tables 3 and 4** (previously Tables 2 and 3) now provide aggregate scRNA and scTCR sequencing metrics for the expanded set of individuals and samples. **Figures 1 and 2** received only minor modifications. The original Figure 3 showing Agilent traces for scTCR libraries was moved to **Suppl Figure 2**. The original Figure 4 (now **Figure 3**) showing dog TRB VJ gene usage was revised to present data for the expanded set of 16 samples and annotations for gene segment type were added. A new **Figure 4** shows TRA/TRB gene segment usage by sample. Figure 5 and Suppl Figure 4 previously showing the TRA/TRB clonotype distribution for two dogs were replaced to now show clonotype distributions across all 16 samples and 11 dogs in a new **Figure 5**. **Figure 6** showing a representative clonotype sequence remains unchanged. Previous Figure 7 and Suppl Figure 5 showed scRNA data overlaid with scTCR data for two representative samples and now show data for 5 samples in a new **Figure 7**. **Suppl Data 1** is unchanged but we added **Suppl Data 2** to provide our alternate consensus reference sequence for TRAV9-2 and **Suppl Data 3** to provide alternate germline alleles detected by cellranger across the cohort. Suppl Table 1 providing clinical history of samples was replaced by **Suppl Tables 1 - 3** which provide clinical histories for the expanded set of 16 samples (11 dogs) including new flow cytometry results. The previous Suppl Tables 2 and 3 with clonotypes observed in two dogs were replaced with **Suppl Table 9** which now provides all clonotypes observed across all 16 samples (11 dogs). Cell typing results in the previous Suppl Table 4 for 2 dogs are now in **Suppl Table 11** for all 5 dogs with scRNA. A new **Suppl Table 4** lists all antibodies used to generate new flow cytometry results. A new **Suppl Table 5** provides complete TRA/TRB VJ gene segment functional annotations from IMGT. New **Suppl Tables 7 and 8** provide raw sequencing QC metrics. **Suppl Figure 1** received only minor modifications. Suppl Figure 2 (now **Suppl Figure 3**) is unchanged. The original Suppl Figure 3 (now **Suppl Figure 4**) showing dog TRA VJ gene usage was revised to present data for the expanded set of 16 samples and annotations for gene segment type were added. New **Suppl Figures 5, 6, and 7** show detailed sequence analysis of three pseudogenes found to be expressed in our data. New **Suppl Figure 8** shows the TRA/TRB VJ gene usage ranked by median number of unique clonotypes. New **Figure 8** and **Suppl Figure 11** show expression of T cell activation and exhaustion markers for clusters of cells identified in scRNA and scTCR data. New **Suppl Table 6** shows T cell activation and exhaustion markers in dog and their corresponding markers in human

investigated in this project. New **Suppl Table 10** summarizes numbers of alternate V gene alleles observed for 16 dogs. New **Suppl Table 12** summarizes expression levels of T cell activation and exhaustion markers in expanded clonotypes versus non-expanded clonotypes. New **Suppl Figure 9** shows the length distribution of reference contig, observed contig, and observed CDR3 for both alpha and beta chain. New **Suppl Figure 10** details the number and proportion of cells of each cell type for the dominant clonotype for each sample.

Reviewers' comments:

Reviewer #1 (Remarks to the Author):

Skidmore and colleagues present a successful modification of the 10x GE + VDJ platform to the sequencing of alpha/beta TCRs from canines. They design canine specific reverse primers for the TCR cDNA enrichment step and apply these to two animals who have undergone cancer immunotherapy. The manuscript demonstrates the successful amplification of TRA and TRB chains and that the cells carrying these chains fall within the T cell populations by the GE analysis. The modifications to the assay are thoroughly and clearly presented. All supporting data is provided or deposited in an appropriate repository.

Overall, it is a robust report of a methodology that is likely to be of interest to those working with canines, both as model animals and within the veterinary field, and is an important contribution to increasing the availability of species-specific assays. The limitation of the manuscript is that it doesn't extend to a detailed characterisation of the assay outputs.

Thank you for these positive words of encouragement. Our primary motivation was indeed to provide a proof-of-concept and working protocol to allow characterization of T cell populations in canine models. Specific concerns will be addressed below.

Major comments:

As a proof of concept for the method, the manuscript is excellent. It is well written and clearly described. The ability of the method to produce scTCR-seq data is well supported by the data provided. Methods are well described. Key results are clearly reported.

Thank you again for these positive remarks.

The manuscript is, however, largely limited to the reporting of the methodology and there is limited analysis of the outputs from the application of the method. The claim in the Abstract, repeated in the Discussion, that the dominant clonal expansions suggest the presence of an anti-tumour population are not supported by the Results in the current form. Given that the scTCR-seq is from PBMCs and that these clonotypes appear to map to CD8+ populations, they could represent memory T cell populations. In the absence of any attempt to characterise the cells using the GE to determine if they do represent an “active T-cell population” the results presented only show that they tend to be CD8+ and nothing about anti-tumour response. This statement should be removed. Removal doesn't impact on the quality of the manuscript in terms of reporting a new method.

Thank you for this critique. We have removed this claim from the abstract. The statement in the discussion was already in our opinion fairly soft where we stated “As melanoma patients undergoing active immunotherapy treatments these patterns might indicate active T cell populations responding to their tumors. However, such conclusions await further immune studies of these canine patient samples.” In this revision we have added additional data which show that all (now n=4) melanoma PBMCs have more clonal TCR repertoires (with a small number of dominant clonotypes) than healthy PBMCs and less clonal (more diverse) repertoires than a T cell malignancy where near total clonality would be expected.

We also assess the expression of a number of known activation and exhaustion markers. Briefly, we consulted with immunology experts at our institute. With their help we identified 21 biomarkers of T cell activation or exhaustion. Using Ensembl we identified the subset (n=16) of these markers with one-to-one orthology. We performed a differential expression analysis comparing expanded (clonotypes at >1% of cells) CD8+ T cells to non-expanded CD8+ T cells. We show that the T cell clusters with the dominant clonotype tend to express more activation markers compared to other T cell clusters which tend to express more exhaustion markers (**Figure 8** and **Suppl Figure 9**). Taken together, these patterns are suggestive of a T cell population responding to a tumor (induced or not by their cancer vaccines). However, without functional studies, such conclusions are still very preliminary. We have tried to cautiously present these results in the discussion with the following:

[Furthermore,] cells classified as CD8+ T cells with expanded clonotypes showed consistently increased expression of known activation markers and decreased expression of exhaustion markers. As these samples were collected from melanoma patients undergoing active immunotherapy treatments, taken together, these patterns

might indicate active T cell populations responding to their tumors. However, such conclusions await further immune studies of these canine patient samples.

In the Results section, sub-heading “Single cell V(D)J repertoire” in the final sentence, there is mention of somatic hypermutation (SHM) being observed. TCRs are not subject to the SHM, this is unique to the immunoglobulin loci. It is also not clear what “recombination-related mutations at V(D)J” are. The elements described in Figure 6 are explained by the mentioned germline variation and V(D)J recombination diversity.

We thank the reviewer for pointing out this error with regard to the mention of SHM. We have sometimes observed small mutations that differentiate otherwise identical CDR3 sequences in large populations of cells and speculate that these may represent somatic mutations (to which any cell is prone) occurring during T cell expansion. But, we did not mean to attribute these to somatic hypermutation which is a specific kind of somatic mutation mediated by AID in B cells.

With regard to “V(D)J recombination diversity” and “recombination-related mutations at V(D)J” we were trying to refer separately to the two main ways that diversity is introduced to TCRs: (1) combinatorial diversity - through the random combination/selection of different V and J gene segments (TRAV8-1/TRAJ1, TRAV8-1/TRAJ2, etc) and (2) junctional diversity - the introduction of errors (insertions, deletions, substitutions) that happen during this recombination, typically in close proximity to the V(D)J joining region. These are the result of double-strand breaks mediated by the RAG1/RAG2 complex followed by non-homologous end-joining repair. We have removed mention of SHM and tried to clarify the language around the introduction of recombination related sequence diversity in the Results.

As expected, individual clonotypes were characterized by evidence of germline variation, V(D)J recombination diversity (usage of different V, D and J gene segments in different combinations), as well as the recombination-related mutations at V(D)J junctions which occur during gene segment joining and contribute to TCR diversity (See **Figure 6** for a representative example clonotype).

With regards to “germline variation” was any attempt made to determine TRA/B genes that were missing from the IMGT reference directory? For example, in Figure 6, a proposed germline variant (or polymorphism/allele) for TRBV4-2 is shown. Was this variation present in all TRBV4-2 from this dog? If so, this would mean this dog likely carries a version of TRBV4-2 that isn’t currently captured in the IMGT reference directory. There is mention of how many of the genes from IMGT are observed, but the

flip side, i.e. how many novel genes from the two animals weren't in IMGT, isn't considered. Similarly in Figure 6, there is also potential for an unreported TRBJ gene. This isn't required for a report on the method, but the approach does permit the identification of this type of germline variation that can contribute to more complete germline gene reference directories.

These are very interesting thoughts/questions. The reviewer implicitly raises two distinct possibilities of: (1) entirely novel V(D)J gene segments/loci which are missing from the IMGT database; and (2) novel V(D)J gene segment alleles which are missing from the IMGT database. Currently the IMGT database effectively has a only single reference sequence for each known V(D)J gene segment based almost entirely off the CanFam3.1 reference sequence. It is possible that some as yet unknown gene segments are entirely missing due to incompleteness of this reference. Even more likely is the absence of alternate alleles for known gene segments. In the human IMGT database you might see TRAV1-2*01, TRAV1-2*02 and TRAV1-2*03 representing three alleles of the same gene segment with polymorphisms differentiating them. For canine there is only ever a single "*01" allele represented. This means that the reviewer is absolutely correct that there are likely versions (alleles) of many canine TRA/TRB V(D)J gene segments not currently captured in the IMGT reference directory. The germline variant shown in **Figure 6** for TRBV4-2 is indeed present in all 17 unique clonotypes which make use of TRBV4-2 from this dog. In fact, that is how the germline/donor sequence is being inferred by the cellranger software, by determining shared sequence between multiple clonotypes, in different cells, that use TRBV4-2 from this dog. Furthermore, this particular germline difference from the reference is observed in all 11 dogs that we sequenced. This suggests that the allele represented in the IMGT reference is either incorrect, relatively rare or private to the CanFam3.1 Boxer from which it was derived. Other germline differences that we observed, inferred by cellranger, are unique to individual dogs or subsets of dogs in our study. Later in this response we describe new results that show other germline differences which suggest the incorrect annotation of several pseudogenes in the IMGT database. It should be noted that the TRBJ gene (TRBJ2-6) in **Figure 6** did not have any such germline differences from the reference (nothing shown in the donor/germline track). Therefore, this does not present evidence for an unreported TRBJ gene. In this case the differences are in the consensus track and represent the unique clonotype produced from recombination and recombination related errors. We have added new text and modified the **Figure 6** caption to better explain this.

Figure 6. Example individual clonotype

Individual cell-specific TRA and TRB clonotypes are resolved to the nucleotide level. Illustrated here is a single such TRB clonotype from Melanoma_B (CDR3:

CASSSVQLAERYF). (A) a specific VDJ recombination with complete TRBV4-2, TRBD1, and TRBJ2-6 and a portion of TRBC is depicted. The Universal Reference (based on IMGT V(D)JC reference sequences) is shown in the first row. Germline variants in the analyzed sample, relative to the Universal Reference, are depicted in the Donor Reference line. The germline/donor sequence is inferred by the cellranger software, by determining shared sequence between multiple clonotypes, in different cells, that use the same gene segment. The Consensus row and subclonotype row(s) show additional variants from the Donor and Universal Reference that were presumably introduced during joining of gene segments. The consensus represents the sequence of the first exact subclonotype for a receptor chain within the clonotype. In this case, only a single subclonotype for a single barcode is shown. But, in other cases, multiple subclonotypes may be grouped together, each represented by one or more cell barcodes. Subclonotypes may have small nucleotide differences or share a TRA chain but have missing TRB chain or vice versa. Single nucleotide changes are shown in orange and small deletions shown in purple. (B) and (C) are zoomed into the base pair level to visualize the variants.

The reviewer suggests the possibility of contributing to a more complete germline gene reference database. This is an inspiring suggestion. However, there are several challenges. The sequence data we have represent the end product of V(D)J recombination at the RNA transcript level. They reflect not only the underlying germline sequence of each dog but also the subsequent somatic changes introduced during recombination. A consensus donor/germline sequence can only be inferred under certain circumstances where there are sufficient distinct cells with different clonotypes such that somatic differences can be subtracted out. We have completed a new cohort level analysis of germline differences observed in our data. We summarize the results of this germline analysis in the **Results (Germline Analysis)** and **Suppl Table 10** and provide the novel predicted alternate allele sequences in **Suppl Data 3**.

Germline Analysis

On average, each sample had 26.75 (range: 17 to 36) alternate allele sequences compared to reference in 21 (range 11 to 28) TRAV or TRBV genes (**Suppl Table 10**). There were 23 TRAV and 13 TRBV genes with at least one alternate allele observed in at least one sample (**Suppl Data 3**). Note that germline variant assessment for J genes is currently not performed by cellranger.

However, these potential alternate alleles should be considered highly provisional. The ends of gene segments where recombination/joining occurs are too variable to confidently know what the true germline sequence might be. We are well positioned to identify specific differences from or errors in the reference sequence but not to propose

full-length novel gene segments or alternate allele sequences for known gene segments. Even with 11 dogs sequenced, we can not easily distinguish between common germline variation worthy of submitting a new reference allele to IMGT from de novo germline variation that might be private to a single dog or its immediate lineage. We can not be certain that an individual germline variation isn't part of a larger haplotype which should be included in the full length sequence of a proposed new allele. Put another way, we can feel confident that we are seeing evidence of alternate alleles but we can't confidently determine their full sequence for submission to a reference database. In recent years there have been a number of new modern genome assemblies and also increasing numbers of whole genome/exome sequences of dog populations. It is these data which might best be used to expand the set of known IMGT genes/alleles. This would be the subject of another paper. However, we do feel confident that our data can be used to re-annotate several IMGT gene segments or identify problematic sequences that should be treated with caution. An entirely new results section has been added in which detailed sequence analysis is used to suggest possible re-annotation of at least 2 pseudogenes and 3 non-functional ORFs. A new discussion of this area of future work has also been added.

New Results text:

Single cell V(D)J repertoire of dogs

In total, across the 16 samples, we profiled 77,809 V(D)J expressing cells and 55,973 clonotypes from 3.0 billion sequence reads. For TCR enriched libraries 72.0%-85.6% of reads mapped to a V(D)J gene with an approximate ratio between β/α chains of 1.5:1 (**Table 3; Suppl Table 7**). Between 45.2-96.9% (median 83.3%) of cells exhibited a productive TRA contig. Similarly between 83.5-99.6% (median 97.6%) of cells exhibited a productive TRB contig. Cells with a productive V-J spanning pair (TRA + TRB) ranged from 34.9-95.3% (median 80.6%). Across all 16 samples, 44,544 (79.6%) clonotypes had at least one TRA/TRB pair (**Suppl Table 7 and 9**). Of these, ~7.7% had an extra TRA, 4.4% had an extra TRB and 2.0% had extras of both TRA and TRB. Only 8 clonotypes (0.01%) had more than 2 TRA or TRB chains which would be suggestive of unfiltered duplicates or some other problem. On average each sample had a median of 69.6% clonotypes with a single TRA/TRB pair, 2.6% TRA only, 16.6% TRB only, 6.3% extra TRA, 3.5% extra TRB, and 1.3% extra TRA and TRB. In samples processed with 10x v1 5' kits we observed noticeably lower median TRA/TRB pairs (30.9%) compared to samples processed with v2 kits (69.8%). Many v1 clonotypes were TRA only (20.2%) or TRB only (40.9%) compared to only 2.5 and 15.7% respectively for v2. The relative amounts of extra TRA (2.3%) and extra TRB (5.4%) in v1 shifted to 6.6% extra TRA and 3.3% extra TRB in v2, more in line with biological expectations for dual TCR expression³⁷. These numbers are consistent with those from the Eschke et al study

which also used the v2 10x protocol (62.1% TRA/TRB , 8.7% TRA only, 21.1% TRB only)¹⁹.

In general, specific expanded TRB sequences tended to be consistently paired with a specific TRA and vice versa. This was especially true for samples processed with the 10x v2 protocol. For example, if we consider Melanoma_C, the most dominant clonotype was TRA CDR3:CAMGPVYSGVGSQ LTF (TRAV9-8::TRAJ28) paired with TRB CDR3:CASAGQGDPHTQYF (TRBV28::TRBJ2-5) inferred for 147 cells (**Suppl Table 9**). We do not see this specific TRA sequence matching any other TRB sequences or vice versa. In other words, the TRA (CAMGPVYSGVGSQ LTF) clonotype and TRB (CASAGQGDPHTQYF) are only seen with each other. This is true for at least the top 5 clonotypes for this sample with only minor exceptions. In some cases a beta chain is matched with different alpha chains in separate clonotypes but the CDR3 amino acid sequences are identical and only small nucleotide level differences are observed for single cells.

We observed the expression of 31 TRAV, 47 TRAJ, 24 TRBV and 12 TRBJ known gene segments at some level of support. The top 5 most frequently used genes observed in our dataset were TRAV43-1, TRAV9-6, TRAV43-4, TRAV12, and TRAV9-9 for TRAV; TRAJ21, TRAJ33, TRAJ27, TRAJ28, and TRAJ31 for TRAJ; TRBV20, TRBV16, TRBV7, TRBV3-2, and TRBV5-2 for TRBV; and TRBJ2-6, TRBJ2-1, TRBJ2-3, TRBJ2-5, and TRBJ1-2 for TRBJ. IMGT categorizes all gene segments as either “Functional”, “Pseudogene”, or “ORF”. Functional gene segments have an open reading frame (ORF) without stop codon, and no defects in the splicing sites, recombination signals and/or regulatory elements. A pseudogene segment is one whose coding region has stop codon(s) and/or frameshift mutation(s), and/or a mutation affecting the initiation codon. An ORF segment has an open reading frame, but alterations have been described in the splicing sites, recombination signals and/or regulatory elements, and/or changes of conserved amino acids that may lead to incorrect folding. For simplicity, we refer to these three types as functional, pseudogene, or non-functional ORF. Considering these annotations, we observed 85.3% (29/34), 100.0% (40/40), 100.0% (22/22) and 100% (9/9) of all known functional TRAV, TRAJ, TRBV and TRBJ gene segments respectively (**Figure 3; Suppl Figure 4**). We also observed expression of 3/44 pseudogene segments (1/23 TRAV, 0/7 TRAJ, 1/13 TRBV, and 1/1 TRBJ) and 11/16 non-functional ORF gene segments (1/1 TRAV, 7/12 TRAJ, 1/1 TRBV, and 2/2 TRBJ). Overall, most expressed pseudogene/ORF segments were observed at low frequency. However, those that were observed were often expressed with a large diversity of different partner gene segments and in a few cases there were clonotypes involving a pseudogene or

non-functional segment that were supported by many cell barcodes, including the highly dominant clonotype (TRBV26::TRBJ1-3) for the T zone lymphoma. Only a single combination occurred where both V and J segments were annotated as pseudogenes (TRBV19::TRBJ1-3; n=7 cells, all in Normal_E samples) and only six combinations occurred where both V and J segments were annotated as non-functional ORF or one was a pseudogene and the other a non-functional ORF (**Figure 3; Suppl Figure 4**). In some cases the use of pseudogene or non-functional ORF segments was unique to a single individual (e.g., TRAJ22 and TRBV19) but in others they were observed in multiple samples (e.g., TRAV9-2, TRAV33, TRAJ10, TRAJ11, TRAJ46, TRAJ59, and TRBJ2-4) or across the entire sample set (e.g., TRAJ52, TRAJ53, TRBV15, TRBJ1-3, and TRBJ1-4) (**Figure 4**). The observation of three pseudogene segments (TRAV9-2, TRBV19 and TRBJ1-3), being used in productive CDR3 clonotypes, was unexpected. According to IMGT, the reference sequences for these gene segments each include one or more frameshift or in-frame stop codon mutations that should result in early termination of the TCR ORF. As a result, these would not be expected to generate the productive CDR3 sequences required by cellranger to be reported as a clonotype. To explain this result, we performed additional sequence analyses for each pseudogene as described below.

Sequence analysis of TRAV9-2 pseudogene usage

The TRAV9-2 pseudogene was identified in 109 unique clonotypes, in combination with 33 different TRAJ gene segments, in 113 cells, for 15 dog samples sequenced (all except Tzone_LSA_LN). IMGT notes an in-frame stop codon in FR1 and a frameshift in FR2. Investigation of 10 randomly selected clonotype sequences, one from each dog, revealed that there were 9 consistent differences (8 SNVs and a single base insertion) in all clonotypes relative to the reference (**Suppl Figure 5A**). In each clonotype sequence, the first stop (TGA) at amino acid position 45 is “corrected” to an S (TCA) by a single base change (G -> C). An insertion of a single base (C) at amino acid position 57 introduces a frameshift relative to the reference sequence. From this frameshift, the clonotype amino acid sequence diverges completely, with the reference sequence having 3 additional stop codons (**Suppl Figure 5B**) whereas the clonotype sequence continues in an open reading frame for the remaining length of the V gene (**Suppl Figure 5C**). Therefore, we propose an alternate consensus sequence for TRAV9-2 which is not a pseudogene (**Suppl Data 2**). This alternate sequence better represents the 11 dogs (1 Bouvier des Flandres, 1 American Cocker Spaniel, 2 Dachshund, 2 mixed breed, 1 Australian Shepherd, 1 Great Pyrenees, 1 Great Dane, 2 Golden Retrievers) that we sequenced, and also has full-length 100% identity, including the inserted C sequence that corrects frame, with more current reference assemblies UU_Cfam_GSD_1.0/canFam4 (German Shepherd, chr8) and UMICH_Zoey_3.1/canFam5 (Great Dane, chrUn_REHQ01000600v1). A BLAST of the

alternate TRAV9-2 sequence against 'nt' also reveals near perfect matches, including the frame-correcting insertion, to both wolf (GenBank: HG994390.1) and Labrador (GenBank: CP050567.1) sequences. In contrast, the reference TRAV9-2 does not align without discrepancies to these genomes. Surprisingly, the Dog10K_Boxer_Tasha/canFam6 (chr8) reference still matches the IMGT reference TRAV9-2 perfectly, including the C deletion that shifts frame but does not match the alternate sequence without discrepancies. Altogether, these results suggest that the TRAV9-2 pseudogene reference sequence in IMGT is either incorrect or a rare variant specific to the canFam3/canFam6 Boxer.

Sequence analysis of TRBV19 pseudogene usage

The TRBV19 pseudogene was identified in 93 unique clonotypes, in combination with all 12 known TRBJ gene segments, in 99 cells, but from only a single dog (Normal_E). IMGT notes an in-frame stop codon in FR3. Investigation of the Normal_E V(D)J sequence data revealed a germline (Donor Reference) T->C mutation (**Suppl Figure 6A**) that changes a stop codon (TAG) at amino acid position 104 in the reference (**Suppl Figure 6B**) to a Q (CAG) in the clonotype sequence (**Suppl Figure 6C**). This variant was present in both lymph node and PBMC samples for the Normal_E dog and observed in all predicted clonotypes, making use of TRBV19 in this dog. In all other samples, there were no clonotypes making use of TRBV19, presumably because they do not have this "corrective" germline mutation and any rearrangement involving TRBV19 would be truncated, non-productive, and filtered out by cellranger. Analysis of additional dogs will be required to determine how common this corrective variant is and whether this gene segment should retain annotation as pseudogene and/or have a note about occasional function or an alternate allele be added to IMGT.

Sequence analysis of TRBJ1-3 pseudogene usage

The TRBJ1-3 pseudogene was identified in 3,413 unique clonotypes, in combination with all 22 known functional TRBV gene segments (and one non-functional ORF and one pseudogene), in 11,711 cells, for all 16 dog samples sequenced. This included the T Zone lymphoma (Tzone_LSA_LN) which had a highly dominant clonotype making use of TRBV26 joined to TRBJ1-3 that was expressed in 95.2% (7,615) of cells for this sample. First, we investigated this specific Tzone_LSA_LN clonotype. IMGT notes an in-frame stop codon in the J region of TRBJ1-3. IMGT also notes a stop-codon in the last 3' codon of TRBV26 "which may disappear during rearrangements". Presumably for this reason TRBV26 was not annotated as a pseudogene. In any case, the dominant clonotype of the T Zone lymphoma apparently had to overcome two defects. Investigation of the dominant T zone lymphoma clonotype sequence revealed an insertion of 4bp and 5 other single base pair changes closely flanking the VJ joining boundary (**Suppl Figure 7A**). If TRBV26 reference sequence is joined directly to the

TRBJ1-3 reference sequence, the result includes the IMGT-annotated stop (TAG) at the end of the V segment and the entire J segment is out of frame, resulting in a stop at the end of the J gene, and presumably creating problems even beyond the stop at amino acid position 2 that otherwise defines TRBJ1-3 as pseudogene (**Suppl Figure 7B**). However, in the altered clonotype sequence we observed there are no stop codons and the J-gene portion is in the correct frame (**Suppl Figure 7C**). In this case, multiple changes (insertions and substitutions) upon V-J joining are “correcting” a premature stop both at the end of TRBV26 and beginning of TRBJ1-3 gene while preserving the correct frame to make a productive CDR3 sequence. To further investigate use of TRBJ1-3, an additional 9 randomly selected clonotype sequences, one from each dog, were examined. In all cases, unique sequence changes introduced during VJ joining eliminate the stop codon from the beginning of TRBJ1-3 while preserving the correct frame (**Suppl Figure 7D**). Overall, TRBJ1-3 was the 8th (of 12) most commonly used TRBJ gene segment in terms of median unique clonotypes with a median of 238 unique clonotypes per sample (**Suppl Figure 8**). Given the very extensive use of TRBJ1-3 in productive TCRs we propose that that this gene segment should be re-annotated as functional, perhaps with a note similar to TRBV26, TRBV28, or TRBJ2-1 that while there is a stop-codon in the second 5' codon this may disappear during rearrangements.

Sequence analysis of ORF gene segment usage

For the 11 ORF gene segments that we observed to be expressed, the majority (8/11) were annotated as having non-canonical V/J heptamer or nonamer sequences or unexpected spacer lengths in the recombination recognition sequences (**Suppl Table 5**). These sequences are recognized by the RAG1/RAG2 enzyme complex during V(D)J recombination. Non-canonical RSS or spacer sequences are expected to interfere with the efficiency of recombination. Despite this, we clearly observed consistent expression of V(D)J transcripts using these gene segments. There are multiple possible explanations for this. There could be errors in the reference sequences, or germline differences between the reference dog and those we sequenced, or more tolerance to these sequence changes for the RAG1/RAG2 enzyme binding than expected. However, because the recognition sequences are lost during the process of recombination and not included in the final V(D)J transcript sequence, we can not test these theories from scTCR data. Genomic DNA sequence-level analysis would be required. We also see use of 4 gene segments with non-conserved FGXG sequences and one gene segment with a conserved TRP replaced by ARG. As above, there are multiple possible explanations for use of these non-functional ORF gene segments. At least three ORF gene segments in particular were utilized very frequently based on median number of unique clonotypes observed across the 16 samples, including TRAJ52 (8/59 most commonly used TRAJ segment; median 110 unique clonotypes), TRBV15 (8/36 most commonly used TRBV segment; median 240 unique clonotypes), and TRBJ1-4 (9/12

most commonly used TRBV segment; median 184 unique clonotypes) (**Suppl Figure 8**). Further analyses are warranted, but based on the frequency of expression of these three ORF gene segments in our data, we propose they could be re-annotated as functional, as above for at least two pseudogene segments.

New Suppl Figures:

Suppl Figure 5. Sequence analysis of TRAV9-2 pseudogene

(A) A Clustal Omega multisequence alignment shows 10 randomly selected clonotypes from 10 different dogs compared to the reference sequence for amino acid positions 41 to 60 of the TRAV9-2 gene segment. A stop codon (TGA) and single base frameshift deletion, as annotated in IMGT, and unique to the reference sequence, are highlighted in red. Four consistent nucleotide differences (3 SNVs and 1 insertion) between clonotypes and reference are indicated along the bottom. (B) The reference TRAV9-2 coding sequence is shown with a stop codon (TGA) at amino acid position 45 and additional stop codons resulting from frameshift deletion are highlighted in red. (C) A representative TRAV9-2 clonotype coding sequence is shown with the alterations that correct nonsense and frameshift mutations observed in the reference.

A.

```

IMGT000004|TRAV9-2*01|Canis          AATTGTACTTACtgaATGTCAGGGTATCCTGTCCFTTTCTGGTGTGTgCAGTATCTGGGG 179
Normal_B_PBMC|clonotype3371         AATTGTACTTACTCAACGTCAGGGTATCCTGTCCFTTTCTGGTATGTCCAGTATCTGGGG 180
Normal_A_PBMC|clonotype134          AATTGTACTTACTCAACGTCAGGGTATCCTGTCCFTTTCTGGTATGTCCAGTATCTGGGG 180
Normal_C_LN|clonotype1431           AATTGTACTTACTCAACGTCAGGGTATCCTGTCCFTTTCTGGTATGTCCAGTATCTGGGG 180
Melanoma_B_PBMC|clonotype956        AATTGTACTTACTCAACGTCAGGGTATCCTGTCCFTTTCTGGTATGTCCAGTATCTGGGG 180
Normal_E_LN|clonotype1418           AATTGTACTTACTCAACGTCAGGGTATCCTGTCCFTTTCTGGTATGTCCAGTATCTGGGG 180
Melanoma_A_PBMC|clonotype616        AATTGTACTTACTCAACGTCAGGGTATCCTGTCCFTTTCTGGTATGTCCAGTATCTGGGG 180
PTCL_NOS_LN|clonotype332           AATTGTACTTACTCAACGTCAGGGTATCCTGTCCFTTTCTGGTATGTCCAGTATCTGGGG 180
Normal_D_LN|clonotype164            AATTGTACTTACTCAACGTCAGGGTATCCTGTCCFTTTCTGGTATGTCCAGTATCTGGGG 180
Melanoma_C_PBMC|clonotype322        AATTGTACTTACTCAACGTCAGGGTATCCTGTCCFTTTCTGGTATGTCCAGTATCTGGGG 180
Melanoma_D_PBMC|clonotype403        AATTGTACTTACTCAACGTCAGGGTATCCTGTCCFTTTCTGGTATGTCCAGTATCTGGGG 180
***** ** *****

```

B.

```

>IMGT000004|TRAV9-2|Canis lupus familiaris boxer|L-PART1+V-EXON
atg aag tgt tct cca ggg atc gtg att gtc cta ttc tta atg ctt gga caa acc cat gga
M K C S P G I V I V L F L M L G Q T H G 20
aac tca gtg aac cag act gaa ggc cag gtg acc gtc tca gaa gag gct tcc ttc aca atg
N S V N Q T E G Q V T V S E E A S F T M 40
aat tgt act tac tga atg tca ggg tat cct gtc ctt ttc tgg tgt gtc agt atc tgg gga
N C T Y * M S G Y P V L F W C V S I W G 60
atg gtc cac agc tcc tcc tga aag cat cag gag aca agg aga agg gaa gta aca aag ggt
M V H S S S * K H Q E T R R R E V T K G 80
ttg aag cca ctt tgg aca gtt cat cca aat cct tcc act tga aga aag gct cac tgc aag
L K P L W T V H P N P S T * R K A H C K 100
tgt cag act cag ctg tgt act act gtg tca tga gtg
C Q T Q L C T T V S * V 112

```

C.

```

>Normal_A_PBMC|clonotype134
atg aag tgt tct cca ggg atc gtg att gtc cta ttc tta atg ctt gga caa acc cat gga
M K C S P G I V I V L F L M L G Q T H G 20
aac tca gtg aac cag act gaa ggc cag gtg acc ctc tca gaa gag gct tcc ttg act atg
N S V N Q T E G Q V T L S E E A S L T M 40
aat tgt act tac tca acg tca ggg tat cct gtc ctt ttc tgg tat gtc cag tat ctg ggg
N C T Y S T S G Y P V L F W Y V Q Y L G 60
aat ggt cca cag ctc ctc ctg aaa gca tca gga gac aag gag aag gga agt aac aaa ggg
N G P Q L L L K A S G D K E K G S N K G 80
ttt gaa gcc act ttg gac agc tca tcc aaa tcc ttc cac ttg aag aaa ggc tca ctg cac
F E A T L D S S S K S F H L K K G S L H 100
gtg tca gac tca gct gtg tac tac tgt gtc atg
V S D S A V Y Y C V M 111

```

Suppl Figure 6. Sequence analysis of TRBV19 pseudogene

(A) A portion of a representative clonotype sequence is shown for Normal_E_PBMC. A single base substitution (T->C) is highlighted in orange and shown to be present in the germline (Donor Reference) of this dog. Note, an insertion (GTTCCCAT), likely related to end-joining, is also shown with a blue bar. (B) The TRBV19 reference coding sequence is shown with stop codon (TAG) at amino acid position 104, highlighted in red. (C) A representative TRBV19 clonotype coding sequence is shown with the T->C alteration that corrects the nonsense mutation observed in the reference to a (CAG) Q amino acid.

A.

Normal_E_PBMC clonotype73

Show Starred Only	TRB	TRB	TRB	TRB	TRB	TRB	TRB	TRB	TRB	TRB													
	TRBV19		TRBV19		TRBV19		TRBV19		TRBJ2-2														
Universal Reference	T	G	A	C	A	T	C	G	G	T	G	C	A	A	A	A	G	A	A	C	T	G	
Donor Reference	T	G	A	C	A	T	C	G	G	T	G	C	A	A	A	G	A	A	C	T	G	T	G
Consensus	T	G	A	C	A	T	C	G	G	T	G	C	A	A	A	G	A	A	C	T	G	T	G
3 Barcodes	T	G	A	C	A	T	C	G	G	T	G	C	A	A	A	G	A	A	C	T	G	T	G

B.

```
>IMGT000005|TRBV19|Canis lupus familiaris_boxer|P|L-PART1+V-EXON
atg ggt aac cag gtg atc tgc tgt gtg gcc ctt tgt ctc ctc gga gca gga aca gca agt
M G N Q V I C C V A L C L L G A G T A S 20
ggt gga atc act cag acc ccc aaa tat ttg ttc aga gag gaa gga cga ggt gtg act ctg
G G I T Q T P K Y L F R E E G R G V T L 40
gaa tgt gaa cag gat ttt aat cat gac tct atg tac tgg tac cga caa gac cca ggg caa
E C E Q D F N H D S M Y W Y R Q D P G Q 60
ggg ctg aga ctg atc tac tac tcg ctg gta gaa aat gat gct cag aaa gga gac ata cct
G L R L I Y Y S L V E N D A Q K G D I P 80
gaa ggc tac agt gcc tct cgg atg aag aag gca ttc ttc tct ctc acc atg aca tcg gtg
E G Y S A S R M K K A F F S L T M T S V 100
caa aag aac TAG aca gct cta tat ctc tgt gcc agt ggt aga
Q K N * T A L Y L C A S G R 114
```

C.

```
>Normal_E_PBMC|clonotype73
atg ggt aac cag gtg atc tgc tgt gcg gcc ctt tgt ctc ctc gga gca gga aca gca agt
M G N Q V I C C A A L C L L G A G T A S 20
ggt gga atc act cag acc ccc aaa tat ttg ttc aga gag aaa gga cga ggc gtg act ctg
G G I T Q T P K Y L F R E K G R G V T L 40
gaa tgt gaa cag gat ttt aat cat gac tct atg tac tgg tac cga caa gac cca ggg caa
E C E Q D F N H D S M Y W Y R Q D P G Q 60
ggg ctg aga ctg atc tac tac tcg ctg gta gaa aat gat gct cag aaa gga gac ata cct
G L R L I Y Y S L V E N D A Q K G D I P 80
gaa ggc tac agt gcc tct cgg atg aag aag gca ttc ttc tct ctc acc atg aca tcg gtg
E G Y S A S R M K K A F F S L T M T S V 100
caa aag aac cag aca gct cta tat ctc tgt gcc agt tcc cat gtg ggg tcg
Q K N Q T A L Y L C A S S H V G S 117
```

Suppl Figure 7. Sequence analysis of TRBJ1-3 pseudogene.

(A) A portion of the dominant clonotype sequence is shown for TZone_LSA_LN revealing a 4bp insertion and 5 other single base pair changes closely flanking the VJ joining boundary. (B) The TRBV26/TRBJ1-3 reference coding sequence is shown. Without modifications these reference sequences include a stop (TAG) at the end of the V segment (first red highlight) and the entire J segment (blue bases) is out of frame. A stop early in the TRBJ1-3 pseudogene is avoided with this frame (second red highlight) but another stop is introduced at the end of the J gene (third red highlight). (C) The dominant TRBV26/TRBJ1-3 clonotype coding sequence from TZone_LSA_LN is shown with no stop codons and the J-gene portion is now in the correct frame. (D) A Clustal Omega multisequence alignment shows the dominant TZone_LSA_LN clonotype along with 10 randomly selected clonotypes from 10 different dogs compared to the reference sequence for the first ~50 bp of the TRBJ1-3 gene segment. In all cases, unique sequence changes introduced during VJ joining eliminate the stop codon from the beginning of TRBJ1-3 while preserving the correct frame.

A.
Tzone_LSA_LN_clonotype1

	TRB	TRB	TRB	TRB	TRB	TRB	TRB	TRB	TRB
	TRBV26		TRBV26		TRBJ1-3		TRBJ1-3		
Universal Reference	FCTGTGTACCTCTGTGCCAGCAGTTAGCCTTTTGA AACACCTTGC ACTTTGGGGACGGGAGCCGGCTCACTG								
Donor Reference	FCTGTGTACCTCTGTGCCAGCAGTTAGCCTTTTGA AACACCTTGC ACTTTGGGGACGGGAGCCGGCTCACTG								
Consensus	FCTGTGTACCTCTGTGCCAGCAGTTAGCCTTTTGA AACACCTTGC ACTTTGGGGACGGGAGCCGGCTCACTG								
☆ 6988 Barcodes	FCTGTGTACCTCTGTGCCAGCAGTTAGCCTTTTGA AACACCTTGC ACTTTGGGGACGGGAGCCGGCTCACTG								
☆ 21 Barcodes	FCTGTGTACCTCTGTGCCAGCAGTTAGCCTTTTGA AACACCTTGC ACTTTGGGGACGGGAGCCGGCTCACTG								
☆ 19 Barcodes	FCTGTGTACCTCTGTGCCAGCAGTTAGCCTTTTGA AACACCTTGC ACTTTGGGGACGGGAGCCGGCTCACTG								
☆ 2 Barcodes	FCTGTGTACCTCTGTGCCAGCAGTTAGCCTTTTGA AACACCTTGC ACTTTGGGGACGGGAGCCGGCTCACTG								
☆ 1 Barcode	FCTGTGTACCTCTGTGCCAGCAGTTAGCCTTTTGA AACACCTTGC ACTTTGGGGACGGGAGCCGGCTCACTG								
☆ 1 Barcode	FCTGTGTACCTCTGTGCCAGCAGTTAGCCTTTTGA AACACCTTGC ACTTTGGGGACGGGAGCCGGCTCACTG								
☆ 1 Barcode	FCTGTGTACCTCTGTGCCAGCAGTTAGCCTTTTGA AACACCTTGC ACTTTGGGGACGGGAGCCGGCTCACTG								
☆ 1 Barcode	FCTGTGTACCTCTGTGCCAGCAGTTAGCCTTTTGA AACACCTTGC ACTTTGGGGACGGGAGCCGGCTCACTG								
☆ 1 Barcode	FCTGTGTACCTCTGTGCCAGCAGTTAGCCTTTTGA AACACCTTGC ACTTTGGGGACGGGAGCCGGCTCACTG								
☆ 1 Barcode	FCTGTGTACCTCTGTGCCAGCAGTTAGCCTTTTGA AACACCTTGC ACTTTGGGGACGGGAGCCGGCTCACTG								

B.
>IMGT000005|TRBV26/TRBJ1-3|Canis lupus familiaris_boxer|L-PART1+V-EXON+J-REGION

```

atg agc aac agg ttg ctc tgc tgt gtt gtc att tgt ctt gtc aaa gta ggt ctc aag gat
M S N R L L C C V V I C L V K V G L K D 20
gct ctg gtc aat cag ttc cca aga cat agg atc ttg ggg aca gga aag aaa tta acc cta
A L V N Q F P R H R I L G T G K K L T L 40
cag tgt ttg cag gat atg aat cat gtt tca atg ttc tgg tat cgc caa gac cca gga ttt
Q C L Q D M N H V S M F W Y R Q D P G F 60
ggg cta cag ctg atc tac tac tca act ggt act gac aac ttt gaa aaa gga gat gcc cct
G L Q L I Y Y S T G T D N F E K G D A P 80
gag ggg tat gat gtc tct cga aat gag ctg aaa tct ttt ccc ctg acc ctg gtc tct gcc
E G Y D V S R N E L K S F P L T L V S A 100
agc acc aac cag aca tct gtg tac ctc tgt gcc agc agt tag cct ttt gaa aca cct tgc
S T N Q T S V Y L C A S S * P F E T P C 120
act ttg ggg acg gga gcc ggc tca ctg ttg tag
T L G T G A G S L L * 131

```

C.
>Tzone_LSA_LN|clonotype1

```

atg agc aac agg ttg ctc tgc tgt gtt gtc att tgt ctt gtc aaa gta ggt ctc aag gat
M S N R L L C C V V I C L V K V G L K D 20
gct ctg gtc aat cag ttc cca aga cat agg atc ttg ggg aca gga aag aaa tta acc cta
A L V N Q F P R H R I L G T G K K L T L 40
cag tgt ttg cag gat atg aat cat gtt tca atg ttc tgg tat cgc caa gac cca gga ttt
Q C L Q D M N H V S M F W Y R Q D P G F 60
ggg cta cag ctg atc tac tac tca act ggt act gac aac ttt gaa aaa gga gat gcc cct
G L Q L I Y Y S T G T D N F E K G D A P 80
gag ggg tat gat gtc tct cga aat gag ctg aaa tct ttt ccc ctg acc ctg gtc tct gcc
E G Y D V S R N E L K S F P L T L V S A 100
agc acc aac cag aca tct gtg tac ctc tgt gcc agc agt tat ggg ggg tgg gga aac acc
S T N Q T S V Y L C A S S Y G G S G N T 120
ttg cac ttt ggg gac ggg agc cgg ctc act gtt gta
L H F G D G S R L T V V 132

```

D.

```

IMGT000005|TRBJ1-3*01|Canis -----CTTTTGA AACACCTTGC ACTTTGGGGACGGGAGCCGGCTCACTGTTGTAG 50
Melanoma_B_PBMC|clonotype65 -----GACGGGGCTGGCTTGC ACTTTGGGGACGGGAGCCGGCTCACTGTTGTAG 49
Normal_D_PBMC|clonotype13 CCGACTACAGGTAACACACCTTGC ACTTTGGGGACGGGAGCCGGCTCACTGTTGTAG 58
Normal_E_PBMC|clonotype6 -----ATAGTGGGCTTGACACCTTGC ACTTTGGGGACGGGAGCCGGCTCACTGTTGTAG 53
Normal_B_LN|clonotype4 -----TATGGACACCTTGC ACTTTGGGGACGGGAGCCGGCTCACTGTTGTAG 46
Normal_C_PBMC|clonotype36 -----GGGTAGCAACACCTTGC ACTTTGGGGACGGGAGCCGGCTCACTGTTGTAG 49
PTCL_NOS_LN|clonotype38 -----ACACGAGAACACCTTGC ACTTTGGGGACGGGAGCCGGCTCACTGTTGTAG 49
Normal_A_LN|clonotype16 -----GGACGTAGAAACCTTGC ACTTTGGGGACGGGAGCCGGCTCACTGTTGTAG 49
Melanoma_D_PBMC|clonotype3 -----ATTAGGAAACACCTTGC ACTTTGGGGACGGGAGCCGGCTCACTGTTGTAG 49
Tzone_LSA_LN|clonotype1 -----GTGGGAAACACCTTGC ACTTTGGGGACGGGAGCCGGCTCACTGTTGTAG 50
Melanoma_C_PBMC|clonotype18 ACACTCAGGGGGAGACACCTTGC ACTTTGGGGACGGGAGCCGGCTCACTGTTGTAG 58
Melanoma_A_PBMC|clonotype23 -----GGGAAACACCTTGC ACTTTGGGGACGGGAGCCGGCTCACTGTTGTAG 47
*****

```

New Discussion material:

A number of gene segments annotated as pseudogene or non-functional ORF were also detected, often in many cells and across multiple dogs (see Results for details). TRBV19 was identified as a pseudogene in IMGT due to an in-frame stop codon in FR3-IMGT but a functional CDR3 was predicted in combination with all functional TRBJ gene segments in a single dog. TRBV19 is functional or has an ORF in numerous other species including humans and the domestic cat⁴⁹. We showed that in at least one dog a germline SNP corrects the stop, making this pseudogene functional. In another case, TRBJ1-3, another pseudogene in IMGT with an in-frame stop codon in the J-region, also produced a functional CDR3 in combination with all TRBV gene segments, and for all dogs. In this case, the stop codon was very near the beginning of the J gene and random changes introduced during recombination appear to frequently correct/obviate the stop. TRAV9-2, another pseudogene in IMGT due to a frameshift in FR2-IMGT and in-frame stop codon in FR1-IMGT also produced productive CDR3s with many different TRAJ gene segments and in all dogs but one. In this case, extensive but consistent germline differences were observed between the reference and all dogs in this study using the gene segment (as well as other wolf and dog reference sequences) which corrected both the stop and frameshift. This observation suggests that the commonly used CanFam3.1/CanFam6 Boxer reference has either a very distinct/unique version of this gene segment or there are errors in its assembly. Taken together these results suggest the re-annotation of at least two (TRAV9-2 and TRBJ1-3) and possibly a third (TRBV19) pseudogene segment as functional. A number of non-functional ORFs (especially TRAJ52, TRBV15 and TRBJ1-4) should also be re-evaluated with whole genome data based on the frequency of their usage in productive TCRs in our data. Most of these were also observed by Eschke et al.¹⁹

In addition to missing some known segments and observing productive usage of pseudogene/non-functional segments, it is also possible that some completely unannotated gene segments were missed. Currently the IMGT database effectively has only a single reference sequence for each known V(D)J gene segment, based almost entirely on the CanFam3.1 reference sequence. It is possible that some as yet unknown gene segments are entirely missing due to incompleteness of this reference. Even more likely is the absence of alternate alleles for known gene segments. We identified a large number of possible alternate alleles in TRAV and TRBV genes with cellranger. However, these sequences have not been validated to a level sufficient for submission to IMGT. Modern canine genome assemblies⁵⁰, future annotation work, and comprehensive germline analysis of data like ours should allow identification of novel gene segments and alleles in the future.

Minor comments:

Please include line numbers to assist with review process.

We have added line numbers.

Table 1 – given that the forward primers are the standard 10x VDJ kit primers it would provide clarity if they were named as such in the Table, or only report the novel TCRA/B primers designed for the study here.

We have indicated that the TRA/TRB Forward 1/2 primer sequences are identical and as provided by 10x in the primer table (now Table 2) and with explanation in a table note.

Table 2. Dog TCR amplification primer sequences

Target	Reaction	Primer Name	Primer Sequence (5' to 3')
TCR α chain (TRA)	First reaction (Outer)	TRA Forward 1 (10x 5' v1)	AATGATACGGCGACCACCGA- GATCTACTCTTTCCCTACACG ACGCTC
		TRA Forward 1 (10x 5' v2)	GATCTACTCTTTCCCTACACG ACGC
		TRA Reverse 1	TCGGTGAACAGGCAGACAGTCC
	Second reaction (Inner)	TRA Forward 2 (10x 5' v1)	AATGATACGGCGACCACCGA- GATCT
		TRA Forward 2 (10x 5' v2)	GATCTACTCTTTCCCTACACG ACGC
		TRA Reverse 2	TGGTACACAGAGGGGTCAGG
TCR β chain (TRB)	First reaction (Outer)	TRB Forward 1 (10x 5' v1)	AATGATACGGCGACCACCGA- GATCTACTCTTTCCCTACACG ACGCTC
		TRB Forward 1 (10x 5' v2)	GATCTACTCTTTCCCTACACG ACGC
		TRB Reverse 1	TTCTGGGTCCGCGAGATCTC
	Second reaction (Inner)	TRB Forward 2 (10x 5' v1)	AATGATACGGCGACCACCGA- GATCT
		TRB Forward 2 (10x 5' v2)	GATCTACTCTTTCCCTACACG ACGC

		TRB Reverse 2	GGTTCAAACACTGTGACCGT
--	--	---------------	----------------------

Note: *TRA/TRB Forward 1/2 primer sequences are identical and as provided by the 10x Chromium Single Cell V(D)J Reagent Kits User Guide (CG000086 Rev L) or 10x Chromium Next GEM Single Cell 5' Reagent Kits v2 (Dual Index) User Guide (CG000331 Rev C) for v1 and v2 kits respectively.

Table 2 – what is the percentage of cells that have paired TRA/B from each animal?

This was reported in Table 2 as “Cells With Productive V-J Spanning Pair”. The summarized statistics of “Cells With Productive V-J Spanning Pair” are now in **Table 3** and raw values in **Suppl Table 7**. We have added a statement to the paper summarizing this result to draw attention to this.

Cells with a productive V-J spanning pair (TRA + TRB) ranged from 34.9-95.3% (median 80.6%).

Figure 4/S3 – legend label, count rather than frequency? Is it count for cell barcodes or TRA/B contigs?

These data for the original two pilot samples are now shown in **Figure 3** and **Suppl Figure 4** for all 16 samples profiled in aggregate. The legend labeled “Frequency” indeed refers to the count of cell barcodes which had expression of each VJ combination. This has been clarified in the figure captions.

The VJ gene combinations identified in samples across all dogs (n = 16; normal LN, normal PBMC, melanoma PBMC, lymphoma LN) are plotted along with their observed cell barcode counts (Frequency) for the TRB chain.

Figure 5/S4 – why not include both animals in a single figure?

These figures have been replaced by a single new **Figure 5** which shows the clonotype distribution of all 16 samples (including both original animals).

Figure 5. Single cell clonotype distribution for TRA and TRB chains for all samples

The proportion of total barcodes, for all clonotypes, is shown for the lymph node aspirates (Normal_LN) and PBMCs (Normal_PBMC) from five healthy dogs, PBMCs from four dogs with melanoma (Melanoma_PBMC) and lymph node aspirates from two dogs with T cell lymphoma (Lymphoma_LN). The healthy normal samples are

characterized by highly diverse clonotypes, with even the most frequent clonotype observed in only a very small proportion of cells. The melanoma PBMC cases are characterized by a small number of dominant clonotypes with higher frequency. The T cell lymphoma cases are characterized by one dominant clonotype in each case.

Figure 6 – is the ‘universal reference’ for the TRA/B the customised IMGT-derived reference set used for cellranger vdj rather than the genomic reference (CanFam3.1) used for the gene expression analysis?

This is a subtle but important distinction. The IMGT TRA/TRB gene segment reference sequences are based almost entirely on the dog reference genome sequence (IMGT000004 and IMGT000005 sequences are directly based on *CanFam3.1*) except for a handful of TRBJ segments which are based on genbank accession HE653929.

The “Universal Reference” depicted in **Figure 6** is indeed based on these IMGT-derived reference sequences used by cellranger vdj. These are the artificially spliced transcript sequences from each V(D)JC gene segment, extracted from IMGT with the cellranger fetch-imgt script, as described in the methods. These were slightly customized because the fetch-imgt script failed to obtain some gene segments (for unknown reasons) and we added them manually, as described in methods. This has been corrected/clarified in **Figure 6** caption and methods.

The Universal Reference (based on IMGT V(D)JC reference sequences) is shown in the first row.

Figure 7/S5

A) doesn't show the overlap, just shows the CD3E expression, by comparing A and C the overlap is indicated.

This point has been clarified in a revised Figure caption for the new **Figure 7** and in corresponding results.

Figure 7. Gene expression based TSNE clustering for dogs with melanoma or T zone lymphoma with cells annotated by CD3E expression, inferred cell type or V(D)J transcript detection.

TSNE clustering based on global gene expression patterns identified a number of distinct clusters of cells from Melanoma_A, Melanoma_B, Melanoma_C, and Melanoma_D PBMCs and T-zone lymphoma lymph node. Genes present in fewer than 10 cells and cells with fewer than 100 genes were filtered out of the dataset. Cells whose mitochondrial expression was in the top 5% across all cells were filtered out and DoubletFinder was used to identify and filter out expected doublets. (A) A T cell marker (CD3E; colored gray to blue) identifies several large clusters. (B) Cell types, inferred based on published expression signatures of blood cell types, identified CD4 (orange) and CD8 (teal) T cell clusters largely overlapping with the CD3E-positive clusters identified in (A) as well as large monocyte (dark blue) and B cell (red) clusters and smaller clusters of several other cell types. Cell types without an assignment or with a population of less than 1% of all cells identified were excluded. (C) Cells identified with V(D)J rearrangements overlap strongly with those identified as CD4/CD8 T cells in (B) or CD3E-positive shown in (A). (D) Cells corresponding to the single most dominant clonotype largely cluster together in the CD8 T cell clusters shown in (B).

Results text

Cells identified as CD8+ or CD4+ T cells largely overlapped with CD3E expressing cells (**Figure 7A**). Similarly, for all 5 samples, cells identified as expressing productive V(D)J

transcripts (**Figure 7C**) overlapped almost completely with those identified as CD3E-positive (**Figure 7A**) and as CD4/CD8 T cells (**Figure 7B**). For melanoma PBMCs, cells corresponding to the single most dominant V(D)J clonotypes largely overlapped with CD8+ T cells (**Figure 7B,D; Suppl Figure 10**).

B) Why not remove cells from the TSNE that were filtered due to mito/doublets/<5 cells in the cluster rather than showing them in gray?

This is a good idea. The TSNE clustering plots have all been replaced in a new **Figure 7**. Cells with fewer than 100 genes, cells whose mitochondrial expression was in the top 5% across all cells, or cells identified as doublets are now filtered out from the dataset completely as described in the methods. These filtered cells and cells with rare cell type assignments are no longer shown in gray. The Figure and caption have been revised accordingly.

Methods (Data pre-processing)

Genes present in fewer than 10 cells and cells with fewer than 100 genes were filtered out of the dataset. Cells whose mitochondrial expression was in the top 5% across all cells were filtered out and DoubletFinder²⁴ was used to identify and filter out expected doublets. DoubletFinder was run with parameters derived using the example provided at <https://github.com/chris-mcginnis-ucsf/DoubletFinder>.

Figure 7. Gene expression based TSNE clustering for dogs with melanoma or T zone lymphoma with cells annotated by CD3E expression, inferred cell type or V(D)J transcript detection.

TSNE clustering based on global gene expression patterns identified a number of distinct clusters of cells from Melanoma_A, Melanoma_B, Melanoma_C, and Melanoma_D PBMCs and T-zone lymphoma lymph node. Genes present in fewer than 10 cells and cells with fewer than 100 genes were filtered out of the dataset. Cells whose mitochondrial expression was in the top 5% across all cells were filtered out and DoubletFinder was used to identify and filter out expected doublets. (A) A T cell marker (CD3E; colored gray to blue) identifies several large clusters. (B) Cell types, inferred based on published expression signatures of blood cell types, identified CD4 (orange) and CD8 (teal) T cell clusters largely overlapping with the CD3E-positive clusters identified in (A) as well as large monocyte (dark blue) and B cell (red) clusters and smaller clusters of several other cell types. Cell types without an assignment or with a population of less than 1% of all cells identified were excluded. (C) Cells identified with V(D)J rearrangements overlap strongly with those identified as CD4/CD8 T cells in (B) or CD3E-positive shown in (A). (D) Cells corresponding to the single most dominant

clonotype largely cluster together in the CD8 T cell clusters shown in (B)..

o B) CD4/8 colours are hard to delineate. Is it possible to change them to less similar colours?

The new **Figure 7** has been revised to use a color scheme with greater contrast. The CD4/8 colors are now very distinct.

o D) Could you quantify the number of dominant clone cells in each cluster as a barchart or similar?

In a new **Suppl Figure 10** we now show the numbers and proportion of dominant clone cells for each cell type cluster.

Suppl Figure 10. Number and proportion of cells corresponding to the dominant clonotype for each cell type.

Plots show cell type assignment for cells corresponding to dominant clonotypes, in number (left) and proportion (right), for Melanoma_A_PBMC, Melanoma_B_PBMC,

Meloma_C_PBMC, Melanoma_D_PBMC, and Tzone_LSA_LN. Dominant clonotype here is defined as the single most frequent clonotype in each sample. Cell types summarized include: B cell (B), CD4+ T cell (CD4), CD8+ T cell (CD8), Dendritic Cell (DC), erythrocyte (ERY), megakaryocyte (MEGA), monocyte (MONO), and Natural Killer cell (NK). Cells corresponding to dominant clonotypes in the scTCRseq experiment were almost exclusively assigned as T cells in scRNAseq experiment, but occasionally assigned as other cell types, potentially due to sample impurity (doublets) or uncertainty/errors in the cell assignment algorithm. Cells corresponding to dominant clonotypes were more often CD8+ than CD4+ T cells.

For future, could consider integrating the two animals into a single analysis?

The manuscript now reports on 16 samples from 11 animals. Where we felt it made sense these have been summarized or integrated into a single analysis. For example, the VJ gene segment usage analyses for TRA and TRB chains are aggregated across all samples (**Figure 3, Suppl Figure 4**). Similarly scTCR and scRNA sequencing metrics are now summarized across the entire cohort in **Tables 3 and 4** (with sample level data in **Suppl Tables 7 and 8**). For most other analyses, sample-level data was important to report and visualize. This gives a better sense of reproducibility and variability in the data and also helps visualize differences between the distinct types of samples we now include (healthy PBMC/LN vs Melanoma PBMC vs Lymphoma LN).

CDR3 length distributions might be of interest to demonstrate that you are capturing transcripts of diverse lengths.

Thank you for the suggestion. We have added **Suppl Figure 9** and text describing CDR3 length distribution to the manuscript.

V(D)J contig length assessment

The lengths for reference TRA V-J and TRB V-D-J gene segment combinations have a median of 399 (346 to 451) and 406 (366 to 426) nt (nucleotides), respectively. Note that the reference length doesn't take into account the length of the 5' UTR and part of the C region sequenced by the protocol, and therefore will naturally be shorter than observed contig lengths. The observed contig lengths (in nt) have a median of 501 (409 to 1122) for alpha chain, and 511 (406 to 1203) for beta chain. The majority of the contig lengths fall within the expected range, with a few outliers. The outliers, which are longer, could result from: (1) misassembly, where the contig is a chimera between two different transcripts which have been assembled together by accident; (2) translation of a long (potentially alternate) 5' UTR; (3) amplification of a larger section of C region due to mispriming; (4) sequencing of intronic regions from incompletely processed RNA that was amplified; or other explanations. The median observed CDR3 lengths are 42 (15 to 72) nt, or 14 (5 to 24) aa (amino acids) for alpha chain; and 42 (18 to 66) nt, or 14 (6 to 22) aa for beta chain. (**Suppl Figure 9**)

Suppl Figure 9. Length distributions of dog TCR alpha chain and beta chain

(A) Distributions of reference lengths for TCR alpha (left) and beta (right) chains. The length of each gene (TRA-V/J, TRB-V/D/J) was inferred from IMGT reference fasta sequences. The reference length for TCR chain was calculated as the sum of TRAV

and TRAJ gene lengths for alpha chain, and the sum of TRBV, TRBD, and TRBJ gene lengths for beta chain. (B) Distribution of observed contig lengths for TCR alpha (top) and beta (bottom) chains. (C) Distribution of observed CDR3 lengths for TCR alpha (left) and beta (bottom) chains. Both observed contig and CDR3 lengths were retrieved from cellranger vdj results. All lengths are reported in nucleotides (nt).

A) Distribution of reference lengths

B) Distribution of observed contig lengths

C) Distribution of observed CDR3 lengths

For the supplemental tables, animal names rather than Dog_A/B are used in the tab names (Rose/Geddy).

Thank you for bringing this to our attention. Our intention was to obscure names to protect client privacy. We believe we have obscured all dog names for the now much larger set of animals throughout all text, figures and tables.

Supplemental Data 1 (the VDJ reference for cell ranger) wasn't available from the manuscript submission system. Are there issues in distributing this as part of the manuscript due to IMGT licensing? Agree that if it is OK to include it should be. It is important to capture the reference used as repeating your well described process for generating the set in the future may result in a different set of reference genes due to changes in the IMGT references.

We apologize if Suppl Data 1 was not made available to you. We did upload it to the manuscript system. It was also available along with our preprint for this manuscript (<https://doi.org/10.1101/2021.06.29.450365>). This set of custom/fixed reference sequences is necessary for others to replicate our analysis, not only because IMGT references might change over time, but also because the fetch-imgt script used to build cellranger reference files does not work (at time of this writing). As described in the methods we had to manually construct missing constant gene entries that the script failed to generate. As a further convenience, and as a result of multiple requests stemming from the preprint, we now also make the complete reference build publicly available for use with the two latest versions of cellranger on our webserver (http://genomedata.org/10X_canine_ref/) and reference this in the paper.

Supplemental data 2 – 4 out of the top 5 paired clonotypes from Dog_A have two TRB chains reported for the clonotypes. Whereas, for Dog_B's expanded clonotypes there's only single TRB chains (as expected). Have you looked at why? Is this caused by doublets, stray RNA in the emulsions picking up barcodes, some quirk of the cellranger software?

This is a good question. We have observed clonotypes with more than one alpha or beta chain (or both) in the same cells in every 10x scTCR library that we have analyzed including for both human and mouse data. So, this issue is not specific to dog. Even 10X's own loupe browser tutorial showing example V(D)J data visualizes multiple clonotypes with with more than one alpha or beta chain (See <https://support.10xgenomics.com/single-cell-vdj/software/visualization/latest/tutorial-chains>). There are a number of different possible explanations. First, there is now

extensive evidence that dual TCRs are a common and natural product of TCR gene rearrangement and thymocyte development (See <https://doi.org/10.4049/jimmunol.1800904> for an extensive review of the topic). It has been reported that ~10% of T cells express dual surface TCR alpha chains (and as much as 30% at mRNA level) and ~1% express dual surface TCR beta chains. This could explain some of the extra TRA/TRB chains we observed. Across all 16 samples now included we detected 55,973 clonotypes and 44,544 had at least one TRA/TRB pair. Of these, ~7.7% had an extra TRA, 4.4% had an extra TRB and 2.0% had extras of both TRA and TRB. This is somewhat less TRA and more TRB than expected. However, it should also be noted that we had approximately twice as many median UMIs per cell for TRB vs TRA from which to derive clonotype sequences so this may have shifted the relative numbers of clonotypes with extra TRB vs TRA chains detected. There are multiple other potential technical explanations for apparent extra TRA/TRB sequences including: Background (extracellular) mRNA, cell doublets, errors in transcription in the cell, errors in reverse transcription to make cDNA, random errors during sequencing, or index hopping in the sequencing process, any of which can introduce noise to the process of contig assembly. We actually have seen a significant increase in “proper” pairing of alpha/beta chains since the first two pilot samples (**Suppl Table 9**). Conversely, the percent of cells with extra TRA or TRB chains is much lower and the relative amount of extra TRB is lower than extra TRA as expected. This suggests at least some of the extra clonotypes observed in the first two dogs were the result of technical issues that have been reduced between v1 (first two dogs) and v2 (all subsequent dogs) kits available from 10x.

Results text:

Across all 16 samples, 44,544 (79.6%) clonotypes had at least one TRA/TRB pair (**Suppl Table 7 and 9**). Of these, ~7.7% had an extra TRA, 4.4% had an extra TRB and 2.0% had extras of both TRA and TRB. Only 8 clonotypes (0.01%) had more than 2 TRA or TRB chains which would be suggestive of unfiltered duplicates or some other problem. On average each sample had a median of 69.6% clonotypes with a single TRA/TRB pair, 2.6% TRA only, 16.6% TRB only, 6.3% extra TRA, 3.5% extra TRB, and 1.3% extra TRA and TRB. In samples processed with 10x v1 5' kits we observed noticeably lower median TRA/TRB pairs (30.9%) compared to samples processed with v2 kits (69.8%). Many v1 clonotypes were TRA only (20.2%) or TRB only (40.9%) compared to only 2.5 and 15.7% respectively for v2. The relative amounts of extra TRA (2.3%) and extra TRB (5.4%) in v1 shifted to 6.6% extra TRA and 3.3% extra TRB in v2, more in line with biological expectations for dual TCR expression³⁷. These numbers are consistent with those from the Eschke et al study which also used the v2 10x protocol (62.1% TRA/TRB , 8.7% TRA only, 21.1% TRB only)¹⁹.

Discussion text:

We observed improved TRA/TRB pairing using 10x v2 5' kits compared to v1 kits with less TRA/TRB only clonotypes and percentages of extra TRA/TRB clonotypes more in line with expectations for dual TCR expression³⁷. Therefore we recommend using v2 kits if possible.

Reviewer #2 (Remarks to the Author):

In this manuscript, the authors present single-cell TCR sequencing profile data for two different dogs under treatment with autologous deglycosylated vaccines derived from tumor cultures, using 10X technology and described the detected TCR sequences and their distribution. To my understanding, a novelty of this work is providing new primer sets to adapt 10X scTCR-seq framework for dog immune cell analysis. Nevertheless, I am not sure how significant the contribution of this work to the research field. If this work has significance in the aspect of methodology development, I suggest the authors highlight more on it.

Thank you for these comments. For the original manuscript, we would argue that we are providing more than just “new primer sets”. Even developing those new primers was not trivial. We describe in detail how we reverse engineered the human/mouse design to replicate for dog samples. This involved creating (in silico) reference VDJ transcripts, designing multiple primer sets, validating these primers in the lab, creating a custom vj reference database for cellranger, generating single cell libraries and sequencing, and then performing comprehensive validation analyses to show these custom reagents and references perform as expected. This was the first scTCR data ever produced for dogs and some of the first scRNA data which we deposit in public databases for others. In the interim we have been contacted by multiple groups seeking to apply this approach in their research.

In the revised paper we take this work significantly further by applying the approach to a large number of additional samples, showing ability to profile not only melanoma PBMCs with possibly expanded clonotypes but also clonal T cell malignancies and highly diverse normal samples. We performed new analyses: (1) investigating VJ gene segment usage in dogs leading to proposed reannotation of several gene segments in the reference database and provide an alternate (more useful) reference sequence for one gene segment (TRAV9-2), (2) surveying germline variation across the cohort, (3) assessing clonality to demonstrate expected differences between healthy normal samples, samples of dogs under immunotherapy treatment, and samples from dogs with T cell malignancies, (4) profiling gene and V(D)J expression at single cell resolution

to demonstrated that expanded clonotypes correspond to specific T cell populations, and (5) characterizing T cell phenotypes for expanded clonotypes in samples from dogs with melanoma. We have summarized these contributions in the discussion.

In this work we were able to successfully adapt existing protocols to perform single cell TCR profiling for 16 samples from 11 individual companion dogs. We were able to detect nearly all known functional V(D)J gene segments for both α and β TCR chains. We also identify several pseudogene or non-functional ORF segments which could be annotated as functional. These are at least in part related to a large amount of germline variation observed in our data, not currently represented in the reference database for dogs. Integration with single cell transcriptome sequencing for 5 samples demonstrated the expected relationship between cells expressing rearranged V(D)J sequences and T cell markers or cell type inferred from global gene expression patterns. A spectrum of T cell clonality was observed in the samples, from monoclonal (T cell lymphoma), to oligoclonal (melanoma) or polyclonal (healthy normal) (**Figure 5**). There was slightly more diversity in healthy lymph nodes than healthy PBMCs but both showed near complete absence of clonal expansion. The PBMC samples from dogs with melanoma showed strong evidence of dominant clonotypes. Furthermore, cells classified as CD8+ T cells with expanded clonotypes showed consistently increased expression of known activation markers and decreased expression of exhaustion markers. As these samples were collected from melanoma patients undergoing active immunotherapy treatments, taken together, these patterns might indicate active T cell populations responding to their tumors. However, such conclusions await further immune studies of these canine patient samples.

As the authors mentioned, there is a value of companion dogs as a model system for human cancer. Then, they should conduct more extensive comparison between human and dogs in terms of TCR diversity. I believe such a comparative analysis will substantially increase the impact of this work.

Assessing TCR diversity/clonality is a common practice in immunogenomic analysis of humans, mice and other organisms. Our intention with this work was to develop and demonstrate an assay (and corresponding informatics) which would make this possible for dogs. At a high level, dogs and humans have highly similar TCR rearrangement systems. Both can produce near infinite levels of diversity. It is not entirely clear what it would mean to perform a “comparison between human and dogs in terms of TCR diversity”. If we wanted to consider the relative usage of different dog V(D)J gene segments, relative to their human orthologs, we would need many more samples to make conclusions. Our motivating use case for dog TCR profiling, as a model for

humans, was to establish that patterns of TCR clonality/diversity and T cell phenotypes could be correlated with clinical correlates. Can we identify/diagnose malignant T cell populations using TCR clonality analysis as done in humans? Can we correlate TCR diversity/clonality with different clinical phenotypes (healthy versus diseased). Can we identify expanded clonotypes representing T cell populations potentially responding to an immune therapy? In the initial paper these were mostly just possible uses of a protocol which we showed to be working. In the revised paper we have now addressed a number of these questions directly. See new text in results and discussion and numerous responses to reviewers above and below.

Currently, the manuscript is descriptive overall. What are characteristics of T cell subsets enriched by dominant TCR clones? Does it similar to the canonical type of T cells that undergo activation and clonal expansion in human immunity?

This is an interesting question, and one that other reviewers also allude to. We have added new analyses to the paper to address this (Methods, Results, Discussion, **Figure 8**, **Suppl Figure 11** and **Suppl Table 11**, **Suppl Table 12**). Briefly, we consulted with immunology experts at our institute. With their help we identified 21 biomarkers of T cell activation or exhaustion. Using Ensembl we identified the subset (n=16) of these markers with one-to-one orthology. We performed a differential expression analysis comparing expanded (clonotypes at >1% of cells) CD8+ T cells to non-expanded CD8+ T cells. We show that the T cell clusters with the dominant clonotype tend to express more activation markers compared to other T cell clusters which tend to express more exhaustion markers (**Figure 8** and **Suppl Figure 11**). Taken together, these patterns are suggestive of a T cell population responding to a tumor (induced or not by their cancer vaccines). However, without functional studies, such conclusions are still very preliminary. We have tried to cautiously present these results in the discussion with the following:

Furthermore, cells classified as CD8+ T cells with expanded clonotypes showed consistently increased expression of known activation markers and decreased expression of exhaustion markers. As these samples were collected from melanoma patients undergoing active immunotherapy treatments, taken together, these patterns might indicate active T cell populations responding to their tumors. However, such conclusions await further immune studies of these canine patient samples.

Reviewers #3-4 (Remarks to the Author):

Skidmore et al. present a primer scheme for the amplification of canine TCR sequences from cDNA generated with 10x-Chromium 5' single-cell gene expression kits. Using this approach, they perform concurrent single-cell RNA and TCR sequencing on PBMC samples obtained from two dogs enrolled in immunotherapy trials for cancer. They demonstrate recovery of TCR sequences that map primarily to clusters of T cells.

As a technical report, this manuscript reports useful advances for the field though some revisions to strengthen and clarify the results are needed prior to publication. The manuscript could be strengthened by the addition of further quality control metrics, and by the addition of an integrated analysis of T cell phenotype and repertoire, which would illustrate the translational potential of such a methodology for profiling both the clonotypes and their gene expression profiles for the same cells.

Major points:

The authors should provide additional quality metrics obtained using their approach, such as:

1) What is the level of TCR sequence consensus produced by cellranger? Are individual reads for the same UMI in agreement with each other?

These are interesting but complicated questions to answer, and arguably beyond the scope of this paper. We are using a custom nested PCR to amplify 10X 5' (v1 or v2) cDNA library fragments, spanning the beginning of V(D)J transcripts to just inside the C gene region. This amplified product, by design of the 10X protocol, contains 16bp cell barcode sequences and 10bp UMI sequences. This product is fragmented and then sequenced on the Illumina platform (2x150bp reads) very deeply. Each clonotype is derived from one or more assembled contigs, itself each composed of multiple UMI read families, each with very many individual reads. Individual reads in UMI families will certainly have sequencing errors. We report sequence qualities at the typical error rate expected of the Illumina platform. An average of 85.7% of Read1 and 87.8% of Read2 bases were at least Q30 (meaning they have less than 0.1% chance of error (summarized in **Table 3** and **Suppl Table 7**). So, not all individual reads for the same UMI will be in perfect agreement with each other. However, only a few reads per UMI read family are typically required to correct for such errors (as few as 3-4 reads per UMI are commonly required). It is also possible for a read to be assigned to the wrong read family, because of errors in the UMI or barcode sequence itself. The cellranger algorithm mitigates against this possibility by requiring reads in a read family to have the same cell barcode, checking barcodes against a white list of allowed sequences, correcting barcode sequences that are at most one Hamming distance from a white list sequence, requiring more than one UMI supporting each barcode and junction region, requiring a minimum number of filtered UMIs, and other filtering steps. During assembly,

reads that don't belong will form distinct branches, paths or components during de bruijn graph simplification and be discarded (<https://support.10xgenomics.com/single-cell-vdj/software/pipelines/latest/algorithms/assembly#graph>). We have many UMIs per clonotype and thousands of reads per UMI family. Specifically, we produced an average of 14.2 median UMIs/cell (range: 11 to 20), and an average of 2,706 median reads/UMI (range: 735 to 4,157) per cell for each clonotype. Assembling each clonotype contig from UMI read family sequences is also a challenging problem and assembly errors are possible. However, the cellranger algorithm incorporates multiple filtering steps to remove problematic contigs. **Overall, we expect the final UMI (and TCR) consensus sequence quality to be very high.** Importantly, the approach to UMI read grouping and clonotype contig assembly is not novel to our paper. Many papers have been published describing the use of the 10X platform for TCR profiling in humans and mouse using this same approach. We are leveraging the 10x protocol for UMI-based correction and the 10x cellranger algorithm for assembling clonotype contigs. **Our advance is to adapt this approach for dogs by designing and optimizing custom primers, creating the necessary reference sequence files, validating the approach (now with positive and negative controls), and demonstrating feasibility to use this approach for tracking immune responses in canine samples.** Nevertheless, we acknowledge that summarizing the overall expected quality of the approach is important. We therefore have added additional summaries of UMI/read QC metrics to the paper to address this point in **Table 3, Suppl Table 7** and the following Results text.

Sequencing data metrics approximated comparable human data for the TCR enriched libraries (**Table 3; Suppl Table 7**)^{34,35}. For scTCR libraries, we observed an average of 4,863 (1,850 to 8,000) estimated V(D)J expressing cells. For these cells, we produced an average of 42,312 (range: 26,772 to 86,352) mean total reads per cell with average fraction of reads in cells of 83.3% (39.8% to 96.0%), an average of 14.2 median UMIs/cell (range: 11 to 20), and an average of 2,706 median reads/UMI (range: 735 to 4,157) per cell for each clonotype (Table 3). Note that 10x only recommends 2,000 read pairs per cell for 150 x 150 sequencing. We have sequenced at approximately 21 times the recommended depth. This is a result of our sequencing core's practice of diverting approximately 1/20 of the scRNA sequencing run towards scTCR. In total, an average of 187.7 (87.0 to 236.8) million reads were generated for each of the 16 V(D)J libraries with 96.2% (94.4 to 97.1%) valid barcodes. Average Q30 bases for barcodes, Unique Molecular Identifier (UMI), Read 1, and Read 2 sequences were 91.5, 90.9, 85.7, and 87.8% and respectively.

2) For expanded TRB sequences, is pairing with TRA sequences consistent from the same dog?

Yes. In general, specific expanded TRB sequences tend to be consistently paired with a specific TRA and vice versa. This is especially true for samples subsequent to the original manuscript submission, after we switched to the 10x v2 protocol, and see higher fraction of cells with the expected paired TRA::TRB clonotypes (See Reviewer #1, last response). For example, if we consider Melanoma_C, the most dominant clonotype is TRA CDR3:CAMGPVYSGVGSQTLTF (TRAV9-8::TRAJ28) paired with TRB CDR3:CASAGQGDPHTQYF (TRBV28::TRBJ2-5) inferred for 147 cells. We do not see this specific TRA sequence matching any other TRB sequences or vice versa. In other words, the TRA (CAMGPVYSGVGSQTLTF) clonotype and TRB (CASAGQGDPHTQYF) are only seen with each other. This is true for at least the top 5 clonotypes for this sample with only minor exceptions. In some cases a beta chain is matched with different alpha chains in separate clonotypes but the CDR3 amino acid sequences are identical and only small nucleotide level differences are observed for a single cell. In other cases there are cells which express multiple alpha and/or beta chains. See response to Reviewer #1 for an explanation of these observations. We have added the following text to address TRA/TRB pairing rates observed in our data as below.

Results text:

Across all 16 samples, 44,544 (79.6%) clonotypes had at least one TRA/TRB pair (**Suppl Table 7 and 9**). Of these, ~7.7% had an extra TRA, 4.4% had an extra TRB and 2.0% had extras of both TRA and TRB. Only 8 clonotypes (0.01%) had more than 2 TRA or TRB chains which would be suggestive of unfiltered duplicates or some other problem. On average each sample had a median of 69.6% clonotypes with a single TRA/TRB pair, 2.6% TRA only, 16.6% TRB only, 6.3% extra TRA, 3.5% extra TRB, and 1.3% extra TRA and TRB. In samples processed with 10x v1 5' kits we observed noticeably lower median TRA/TRB pairs (30.9%) compared to samples processed with v2 kits (69.8%). Many v1 clonotypes were TRA only (20.2%) or TRB only (40.9%) compared to only 2.5 and 15.7% respectively for v2. The relative amounts of extra TRA (2.3%) and extra TRB (5.4%) in v1 shifted to 6.6% extra TRA and 3.3% extra TRB in v2, more in line with biological expectations for dual TCR expression³⁷. These numbers are consistent with those from the Eschke et al study which also used the v2 10x protocol (62.1% TRA/TRB , 8.7% TRA only, 21.1% TRB only)¹⁹.

In general, specific expanded TRB sequences tended to be consistently paired with a specific TRA and vice versa. This was especially true for samples processed with the 10x v2 protocol. For example, if we consider Melanoma_C, the most dominant clonotype was TRA CDR3:CAMGPVYSGVGSQTLTF (TRAV9-8::TRAJ28) paired with TRB CDR3:CASAGQGDPHTQYF (TRBV28::TRBJ2-5) inferred for 147 cells (**Suppl Table 9**). We do not see this specific TRA sequence matching any other TRB

sequences or vice versa. In other words, the TRA (CAMGPVYSGVGSQTLTF) clonotype and TRB (CASAGQGDPHTQYF) are only seen with each other. This is true for at least the top 5 clonotypes for this sample with only minor exceptions. In some cases a beta chain is matched with different alpha chains in separate clonotypes but the CDR3 amino acid sequences are identical and only small nucleotide level differences are observed for single cells.

Discussion text:

We observed improved TRA/TRB pairing using 10x v2 5' kits compared to v1 kits with less TRA/TRB only clonotypes and percentages of extra TRA/TRB clonotypes more in line with expectations for dual TCR expression³⁷. Therefore we recommend using v2 kits if possible.

3) *For expanded TRB/TRA sequences, is mapping consistently to either CD8+ or CD4+ T cells?*

In all cases (the original two melanoma PBMC samples, and now two additional melanoma PBMC samples, and T-zone lymphoma samples) the expanded TRA/TRB sequences (clonotypes) are almost exclusively mapping to CD8+ T cells. This is shown in the revised **Figure 7** and **Suppl Figure 10** (see below).

(Figure 7) The single most dominant clonotype in each sample (red population in panel D) largely overlaps with CD8+ population (teal population in panel B).

(Suppl Figure 10) The single most dominant clonotype in each sample is usually assigned to CD8+ population (teal population in panel A and B).

This has been clarified in the results section with the following text:

For melanoma PBMCs, cells corresponding to the single most dominant V(D)J clonotypes largely overlapped with CD8+ T cells (Figure 7B,D; Suppl Figure 10). For the T zone lymphoma, the vast majority of cells correspond to a single dominant

clonotype, which also was almost entirely CD8+ T cells (**Figure 7B,D; Suppl Figure 10**).

4) For cells with multiple UMI mapping to TRB/TRA, are these sequences in agreement with each other? It would be expected that some fraction of sequences are artifacts generated by factors such as mispriming during the RT reaction or the generation of chimeric PCR byproducts. These technical features would strengthen the understanding of the quality of the pairings recovered and the methods presented.

This has been addressed in detail in response to question #1.

The authors should explicitly mention the level of agreement between the TRBV/TRAV/TRBJ/TRAJ sequences they obtain and those currently in canine reference genomes and on IMGT. The authors are correct that these references are incomplete for non-model species (i.e. human and mouse). The annotation of novel V/J segments and alleles in this species would be a valuable addition to the field. Are these gaps in the reference genome a factor in the limited coverage of TRBV gene segments observed in the data shown in Figure 4? It would appear to be consistent gaps across both dogs evaluated.

Thank you for pointing out this limitation in the presentation of our results. The gaps in Figure 4 are not explained by possible gaps in the reference as these gene segments all come from IMGT which is based on the same reference genome. Most of the gaps that were previously observed can be explained by considering the annotation/status of known V(D)J gene segments. We did not previously annotate pseudogene and non-functional gene segments. Most gaps overlapped with these pseudogene/non-functional gene segments. Some others were filled in by increased sampling. We have now added 14 additional samples to this study and added the appropriate V(D)J gene annotations. We show that 31 TRAV, 47 TRAJ, 24 TRBV and 12 TRBJ total gene segments were observed representing 29/34 (85.3%), 40/40 (100.0%), 22/22 (100.0%) and 9/9 (100.0%) of known functional gene segments respectively. A few pseudogene or otherwise defective gene segments were also observed to be expressed and after sequence analysis we propose re-annotation of several. This is summarized in a new **Figure 3** for TRB and **Suppl Figure 4** for TRA. The possibility of identifying novel gene segments is addressed in detail in response to Reviewer #1.

Figure 3 in revised manuscript. The gaps in coverage are observed in pseudogenes (gene names in blue).

Suppl Figure 4. The gaps in coverage are observed in pseudogenes (gene names in blue) and nonfunctional ORF (gene names in red).

“As expected, individual clonotypes were characterized by evidence of germline variation, V(D)J recombination diversity, as well as somatic hypermutation and/or recombination-related mutations at V(D)J junctions (See Figure 6 for a representative example clonotype).”

The authors should quantify and present these factors rather than showing data for a single representative clonotype. Expanding the analyses in Figure 6 for a plurality of

clonotypes would further address these factors and show further generalizability of the method.

First, we apologize for referencing somatic hypermutation in error as pointed out by Reviewer 1. Recombination-related mutations at V(D)J junctions is the basis for CDR3 clonotype diversity which is quantified, analyzed and visualized throughout the paper. The ability to accurately differentiate germline variation from the somatic recombination related changes above (or other non-SHM somatic changes) is highly dependent on the number of cells/clonotypes which express any given V(D)J gene segment in each sample. However, we now include an analysis of germline (donor reference) sequences across our dataset of 16 samples and identified 36 gene segments (23 TRAV, 13 TRBV) with at least one germline difference from reference for a total of 428 alternate donor references. Each sample had on average 26.75 non-reference germline sequences for an average of 21.1 gene segments. Some germline differences are unique to a single or few individuals while others are shared across every individual. Some of these may represent potential errors in the reference as described above. In general, where multiple samples are available the germline differences are highly consistent. We have included the novel germline donor reference sequences as **Suppl Data 3**. We have added a new germline analysis section to the results.

Germline Analysis

On average, each sample had 26.75 (range: 17 to 36) alternate allele sequences compared to reference in 21 (range 11 to 28) TRAV or TRBV genes (**Suppl Table 10**). There were 23 TRAV and 13 TRBV genes with at least one alternate allele observed in at least one sample (**Suppl Data 3**). Note that germline variant assessment for J genes is currently not performed by cellranger.

Can the authors demonstrate a relationship between repertoire and phenotype – for example, do “dominant clonotypes” express greater levels of activation signatures than other clonotypes? This analysis would further motivate the simultaneous profiling of gene expression and TCR repertoire. The data presented in Figure 7 appear suited to consider these connections further.

This is an interesting question, and one that other reviewers also allude to. We have added new analyses to the paper to address this (Methods, Results, Discussion, **Figure 8**, **Suppl Figure 11** and **Suppl Table 11**, **Suppl Table 12**). Briefly, we consulted with immunology experts at our institute. With their help we identified 21 biomarkers of T cell activation or exhaustion. Using Ensembl we identified the subset (n=16) of these markers with one-to-one orthology. We performed a differential expression analysis comparing expanded (clonotypes at >1% of cells) CD8+ T cells to non-expanded CD8+

T cells. We show that the T cell clusters with the dominant clonotype tend to express more activation markers compared to other T cell clusters which tend to express more exhaustion markers (Figure 8 and Suppl Figure 11). Taken together, these patterns are suggestive of a T cell population responding to a tumor (induced or not by their cancer vaccines). However, without functional studies, such conclusions are still very preliminary. We have tried to cautiously present these results in the discussion.

Figure 8. Expression of T cell activation and exhaustion markers in expanded clonotypes versus non-expanded clonotypes for Melanoma_B_PBMC

Heatmap of single cell expression values ($\log_e(x + 1)$) normalized and scaled for all cells in the sample) for expanded CD8+ T cells (left) vs non-expanded CD8+ T cells (right) for known markers of T cell activation (top) and exhaustion (bottom). Expanded cells are those with a clonotype frequency greater than 1%. Marker genes are colored blue if their expression is significantly increased or red if significantly decreased in expanded vs non-expanded CD8+ T cells (adjusted p-value < 0.05) for this dog sample (Melanoma_B_PBMC) (Suppl Table 12).

Methods:

Differential gene expression analysis of expanded versus non-expanded CD8+ T cells
CD8+ T cells (see Cell typing) were categorized as expanded if they had a clonotype represented in >1% of all cells. Otherwise they were considered non-expanded. A set of known human T cell activation (n=8) and exhaustion (n=14) markers was identified from the literature²⁸⁻³² (**Suppl Table 6**). Of these, 4 activation (CD38, GZMA, GZMK, MKI67) and 12 exhaustion (CTLA4, HAVCR2, NFATC1, NR4A1, NR4A2, NR4A3, PDCD1, PRDM1, TCF7, TIGIT, TOX, TOX2) could be reliably mapped to dog orthologs (Ensembl 102, one-to-one). Differential expression was tested for all 16 markers, using the Seurat package, with a Wilcoxon Ranked Sum test. A marker was considered significant if the adjusted p-value (Bonferroni correction) was < 0.05.

Results:

Differential expression analysis showed that expanded CD8+ T cells had significantly increased expression of known T cell activation markers GZMA, GZMK, or CD38 (adjusted p-value < 0.05) and significantly decreased expression of exhaustion-related markers CTLA4, TOX, or NFATC1 compared to non-expanded CD8+ T cells (adjusted p-value < 0.05) (**Figure 8, Suppl Figure 11, Suppl Table 12**). The inverse pattern (significantly decreased activation or increased exhaustion) was not observed. Additionally, we noted decreased expression of TCF7, encoding for TCF1 protein, which when co-expressed with TOX supports a progenitor exhausted fate, and TCF7 is subsequently downregulated upon terminal exhaustion.²⁹

Discussion:

Furthermore, cells classified as CD8+ T cells with expanded clonotypes showed consistently increased expression of known activation markers and decreased expression of exhaustion markers. As these samples were collected from melanoma patients undergoing active immunotherapy treatments, taken together, these patterns might indicate active T cell populations responding to their tumors. However, such conclusions await further immune studies of these canine patient samples.

Discussion: "Indeed, we observed that a number of the more frequent clonotypes could be merged based on CDR3 sequence similarity, even further emphasizing the dominant clonotype pattern (Suppl Table 2-3)" It is not clear how "clonotypes were merged based on CDR3 sequence similarity" from Supplemental Tables 2 and 3. This is currently an active area of investigation in the field, with many proposed solutions (Dash et al Nature 2017, Huang et al Nature Biotechnology 2020, Zhang et al Clin Cancer Res 2020), none

of which seemed to be cited or utilized here. Better justification for this statement needs to be provided and clarification on how the consolidation was performed.

Thank you for bringing this up. We did not mean to imply that we performed clonotype merging. We made a manual observation that in some cases, recurrent clonotypes appeared to be related. In some cases there were TRA_only, TRB_only and TRA_TRB clonotypes, all at high frequency that seem likely to correspond to the same underlying clonotype but for which some cells did not have resolution of both chains. In other cases, there were separate clonotypes with highly similar (or even identical) CDR3 amino acid sequences that for whatever reason were not grouped by the cellranger algorithm. We have seen this issue manifest for all TCR profiling data we have observed for dog, human or mouse, for multiple platforms. As the reviewer indicates, this is an active area of development, but not a focus of this paper. Such improvements would best be implemented in the cellranger algorithm or other TCR analysis tools. We are not refining these algorithms in this work. We have clarified and added a caveat to the discussion that improved clonotype merging is an area for future development.

Discussion

Interpretation of this data may also benefit from advanced tools for TCR sequence analysis such as improved TCR clonotype merging not assessed here ⁵¹⁻⁵³.

Minor points:

The authors should mention how “dominant clonotypes” are defined.

For the new T cell activation/exhaustion analysis we define dominant/expanded clonotypes as those at greater than 1% of all cells. But, otherwise, we refer to the “single/top most dominant clonotype”. This has now been clarified in the methods, results and figure captions throughout the paper. We do not formally define each clonotype as dominant (or non-dominant) or each sample as having a dominant clonotype (or not). However, there is a very clear qualitative and quantitative difference demonstrated between (1) healthy/normal samples which have virtually no expanded clonotypes, (2) melanomas which consistently have a few very expanded clonotypes, and (3) T cell lymphomas which each have a single very highly dominant clonotype (**Figures 5** and **Suppl Table 9**). The inclusion of additional melanoma samples, lymphomas, and healthy normals in this revised paper clarify these differences.

The authors’ approach for TCR library preparation presented here relies on 5`-template switching, which is primarily compatible with 10x Chromium 5` v2 kits, not newer 10x v3

GE kits or other many other approaches for construction of 3'-barcoded libraries. This limitation should be explicitly acknowledged in the Discussion of the results to contextualize this work with other related studies in the literature in other systems (e.g., human, mouse, etc.). What is required to adapt this method to other library chemistries?

Yes. Just like for established human and mouse protocols our dog protocol is limited to 5' 10X (v1 or v2) kits. With 5' based kits you can do 5' GEX and TCR/BCR profiling but not 3' GEX (v1, v2 or v3), Multiome (GEX + ATAC), and any other 3' based approaches. This is a biological limitation not easily overcome. It results from the fact that the C region is at the 3' end of V(D)J transcript. For the 5' protocol, the 10X adaptors (including R1 sequence) are at the 5' (V gene) end. This allows the V(D)J CDR3 region to be amplified with a single set of PCR reactions using the constant C region and the R1 region. For the 3' protocol, the 10x adaptors are at the 3' (C gene) end. With this orientation, primers for the R1 region and C region would only be able to amplify the C region, not the CDR3 region. Adapting the method for 3' chemistry would require a complicated multiplex PCR which used either R1 or the C region AND primers against all possible V genes, similar to approaches like Adaptive. You could probably send your 10X 3' cDNA to adaptive for sequencing and get TCR profiles, but would lose single cell resolution because the 10X cell barcode would be outside the amplified region. If you developed your own multiplex PCR using the R1 region (in order to preserve the 10X cell barcode) and all possible V genes, the products would include the entire (402-531 bp) C region at 3' end of the transcript. Given the expected 3' end-bias it would be extremely hard to get sufficient coverage of the central CDR3 region needed for resolution of clonotypes. This problem is not unique to dogs but rather common to all 10x single cell and we feel that solving this problem is outside the scope of this paper. We have added this caveat to the paper.

It should be noted that the protocols described in this paper, as with human and mouse protocols, are limited to 5'-based 10x (v1 or v2) protocols. They can not be easily adapted to 3' GEX (v1, v2 or v3), Multiome, or other 3'-based 10x approaches

Figures 4 and 5 are not very informative. At the very least, these figures should be presented on a log-scale to better visualize the quantitative spectrum of their data. These could also be consolidated to one figure to provide space for additional analyses as noted above.

Thank you for this feedback. These figures have been reworked. The previous Figure 4 is now represented by a new **Figure 3** and previous Figure 5 is now represented by a new **Figure 5**. The new **Figure 3** shows VJ gene segment usage for the beta chain, aggregated across the now expanded set of 16 samples (**Suppl Figure 4** shows the

results for alpha chain). The new **Figure 5** shows the TRA/TRB clonotype distribution also for all 16 samples with a new visualization approach. With the increased numbers of samples and new presentation we did not feel consolidation of these two figures made sense any more. Also, we felt that unnormalized values (non log scale) were a more accurate reflection of the data. **Figure 3** shows a small number of VJ combinations at high frequency (representing the combinations that happened to be used in expanded/dominant clonotypes) and a large number of combinations at relatively similar, low levels. This is an accurate reflection of our expectations/observations. The main purpose of the figure is to show we have near complete coverage of known/functional gene segments (colored) and absence of most pseudogene/non-functional segments (non-colored, white). However, it also emphasizes the few extremely expanded combinations of two T cell lymphomas which were added to the study as a kind of positive control. The new **Figure 5** now shows proportions of cells represented by different subsets of clonotypes (top 1, top 2-5, etc) and logging these proportions wouldn't make sense.

“As expected, individual clonotypes were characterized by evidence of germline variation, V(D)J recombination diversity, as well as somatic hypermutation and/or recombination-related mutations at V(D)J junctions (See Figure 6 for a representative example clonotype).”

It is unexpected that TCR repertoires would exhibit somatic hypermutation, as this process typically only occurs in B cells. Can the authors provide further evidence for this event? If not, it would make sense to revise the statement in accordance with the current understanding of T cell development.

We thank the reviewer for pointing out this error with regard to the mention of somatic hypermutation (SHM). We have sometimes observed small mutations that differentiate otherwise identical CDR3 sequences in large populations of cells and speculate that these may represent somatic mutations (which any cell is prone to) occurring during T cell expansion. But, we did not mean to attribute these to somatic hypermutation which is a specific kind of somatic mutation mediated by AID in B cells. We have removed mention of SHM and tried to clarify the language around the introduction of recombination related sequence diversity in the Results - V(D)J Clonality and Diversity Assessment.

As expected, individual clonotypes were characterized by evidence of germline variation, V(D)J recombination diversity (usage of different V, D and J gene segments in different combinations), as well as the recombination-related mutations at V(D)J junctions which occur during gene segment joining and contribute to TCR diversity (See **Figure 6** for a representative example clonotype).

Reviewers' comments:

Reviewer #1 (Remarks to the Author):

Thank you for all the hardwork in adressing my previous comments.

All queries have been adequately addressed and I have no further comments.

Reviewer #5 (Remarks to the Author):

Skidmore et al have developed protocols and primer reagents for joint scRNA/TCR-seq profiling in dogs and demonstrated applicability in healthy and cancer settings. This study will likely have an impact on the field of cancer immunology in dogs because a) the data presented here are of high-quality and b) the protocols developed are compatible with the widely installed 10x Genomics platform. This manuscript has undergone a previous round of review, and in my opinion, the authors present a thoughtful and thorough response to these prior reviews. Thus, I have only minor technical comments for the authors' consideration:

1) The results of the comparison of TCR diversity between healthy dogs and dogs with melanoma and lymphoma shown in Figure 5 are as expected. Nonetheless, a proper comparison of clonal diversity requires that the cell numbers across samples be roughly equal, because clonal diversity is confounded with sampling (i.e. diversity increases as cell numbers increase). From Table 3, it appears that the number of cells in which a productive TCR was detected varies over roughly an order-of-magnitude across these samples, which is not unexpected. Thus, it would worth generating a version of Figure 5 where the cell numbers are randomly sub-sampled to the same number (i.e. to the number of clonotyped T cells in the sample with the fewest clonotyped T cells). The authors might also consider showing the breakdown of Figure 5 for CD4 and CD8.

2) Figure 8 shows UMAP embeddings colored by several covariates for the five samples for which scRNA-seq profiling was performed. There appears to be considerable co-clustering of CD4 and CD8 T cells, which is not surprising. In many cases, this is because memory status rather than CD4 vs. CD8 lineage is a dominant contribution to gene expression for T cell heterogeneity. To clarify this, I would suggest that the authors plot these same UMAP embeddings but color the embeddings by expression of the following markers: CD4, CD8, CCL5, and SELL. I'm not sure if there are straightforward orthologs of CCL5 (RANTES) and SELL (CD62-L or L-selectin) in dogs, but in humans, CCL5 transcript is a very highly expressed marker of effector memory T cells (likely enriched among CD8s in this setting), whereas SELL transcript markers naïve or central memory T cells.

Re: COMMSBIO-21-2112A

First, we would like to thank the editor and reviewers for their careful reading and thoughtful suggestions. In response to this feedback, we have now completed another revision that we believe addresses the remaining concerns. We will first briefly summarize the nature and extent of these revisions. Then we will address each comment individually. The original reviewers' comments are in *black italics*, our responses in plain blue text, and any new text added to the manuscript is in **plain black text, highlighted teal**. The manuscript itself has been highlighted with changes in teal as well. New figures, referred to in this response, are embedded in this document for convenience of the reviewers and also included in the revised submission.

Summary of response:

In response to the reviewers comments several new analyses were performed. The text of the methods and results were supplemented or revised as needed. Supplementary Table 6 was updated to include T cell markers of effector memory and naive status. New **Suppl Figure 10** shows clonality analysis after random downsampling to account for different total numbers of T cells across the sample cohort. New **Suppl Figure 11** shows clonality analysis for CD4 and CD8 subsets of T cells. New **Suppl Figure 14** projects effector memory and naive status marker expression on existing t-SNE plots. New **Suppl Figure 15** shows the same marker expression but focused on a subset of CD8 T cells co-clustering with CD4 cells.

Reviewers' comments:

Reviewer #1 (Remarks to the Author):

Thank you for all the hardwork in addressing my previous comments.

All queries have been adequately addressed and I have no further comments.

We sincerely thank the reviewer for their re-review of our paper.

Reviewer #5 (Remarks to the Author):

Skidmore et al have developed protocols and primer reagents for joint scRNA/TCR-seq profiling in dogs and demonstrated applicability in healthy and cancer settings. This study will likely have an impact on the field of cancer immunology in dogs because a) the data presented here are of high-quality and b) the protocols developed are compatible with the widely installed 10x Genomics platform. This manuscript has undergone a previous round of review, and in my opinion, the authors present a thoughtful and thorough response to these prior reviews. Thus, I have only minor technical comments for the authors' consideration:

We thank the reviewer for their review of our revised paper and additional comments. We appreciate this effort and have responded to the specific technical comments below.

1) *The results of the comparison of TCR diversity between healthy dogs and dogs with melanoma and lymphoma shown in Figure 5 are as expected. Nonetheless, a proper comparison of clonal diversity requires that the cell numbers across samples be roughly equal, because clonal diversity is confounded with sampling (i.e. diversity increases as cell numbers increase). From Table 3, it appears that the number of cells in which a productive TCR was detected varies over roughly an order-of-magnitude across these samples, which is not unexpected. Thus, it would worth generating a version of Figure 5 where the cell numbers are randomly sub-sampled to the same number (i.e. to the number of clonotyped T cells in the sample with the fewest clonotyped T cells). The authors might also consider showing the breakdown of Figure 5 for CD4 and CD8.*

Thank you for your thoughtful comments. We agree that an ideal comparison of clonal diversity requires that the cell numbers across samples be roughly equal. To address this concern, we have generated a new supplementary figure using average results from data randomly sub-sampled, with 100 permutations, with replacement, to the lowest number of clonotyped T cells (n=1,850) observed in any sample. As shown in the new **Suppl Figure 10**, the downsampled data show that the patterns of TCR diversity are preserved and extremely similar to those presented in **Figure 5**. To maintain simplicity of presentation for the reader we have retained the original observed cell numbers and fractions (to avoid having to describe averages of averages, etc) but add new text explaining that we considered the potential effect of different numbers of cells and pointing to the new Suppl Figure. We have also stratified by CD4 and CD8 T cells as suggested and present this data in the new **Suppl Figure 11**. These data show that for the 4 melanoma PBMCs and 1 T zone lymphoma for which we had expression data needed to classify CD4 vs CD8 T cells, the dominant/expanded clonotypes are largely CD8 T cells. This is consistent with the previously included Figure 7 which showed that cells with the most dominant V(D)J clonotype largely cluster together in the CD8 T cell clusters. These results have been described in the paper with the following text and figures.

A potential limitation of the analysis of clonotype diversity (**Figure 5**) is that cell numbers across samples were not equal (range 1,850 to 8,000). To address this concern, we computed average cell proportions, across 100 permutations, randomly downsampled, with replacement, to the lowest number of clonotyped T cells observed in any sample (n=1,850 cells). As shown in **Suppl Figure 10**, the downsampled data show that the patterns of TCR diversity are highly concordant with the non-downsampled data. This analysis suggests that our conclusions regarding different clonality/diversity between sample groups (healthy vs melanoma vs lymphoma) are not confounded by differences in T cell number.

We also assessed the clonotype diversity for CD4+ and CD8+ subsets (defined by singleR cell typing) for the 5 samples with gene expression data (**Suppl Figure 11**). The CD8+ subset of the T zone lymphoma sample (n=6,938 cells) is characterized by a single dominant clonotype that accounts for 92% of the cells. In contrast, the CD4+ subset of the T zone (n=158 cells) was more diverse with the most dominant clone accounting for only 19.6%. This suggests that the T zone lymphoma malignant clone is predominantly CD8+. Similarly, for three of four melanoma PBMCs, the CD8 subset of cells were generally more oligoclonal (less diverse) compared to CD4 cells. However, both

had some evidence of clonotype expansion as well as a diversity of different clonotypes.

Melanoma_D_PBMC had similar patterns of CD4 and CD8 expansion. As above, downsampling to a common minimum number of cells did not affect these conclusions.

Suppl Figure 10. Single cell clonotype distribution for TRA and TRB chains for all samples after downsampling

The proportion of total barcodes, for all clonotypes, is shown for the lymph node aspirates (Normal_LN) and PBMCs (Normal_PBMC) from five healthy dogs, PBMCs from four dogs with melanoma (Melanoma_PBMC) and lymph node aspirates from two dogs with T cell lymphoma (Lymphoma_LN). Proportion was estimated by downsampling the number of cells to the smallest number of cells with a detected clonotype (Melanoma_A_PBMC, n=1850), calculating the fraction of cells in each bin (Clonotype 1, Clonotype 2 - 5, etc where the clonotypes are sorted in descending order of cell counts), repeating 100 times, and calculating the average. The healthy normal samples are characterized by highly diverse clonotypes, with even the most frequent clonotype observed in only a very small proportion of cells. The melanoma PBMC cases are characterized by a small number of dominant clonotypes with higher frequency. The T cell lymphoma cases are characterized by one dominant clonotype in each case.

Suppl Figure 11. Single cell clonotype distribution for TRA and TRB chains for all samples for CD4+ and CD8+ T cell types

The proportion of total barcodes, for all clonotypes, is shown for the PBMCs from four dogs with melanoma and lymph node aspirate from one dog with T cell lymphoma for CD4+ cells (panels A, C) and CD8+ cells (panels B, D) according to singleR cell typing. In panel A and B, the proportion was estimated by calculating the fraction of cells in each bin (Clonotype 1, Clonotype 2 - 5, etc where the clonotypes are sorted in descending order of cell counts). In panel C and D, the proportion was estimated by downsampling the number of cells to the smallest number of cells with a detected clonotype (Tzone_LSA_LN, n=158 for CD4+ and Melanoma_A_PBMC, n=773 for CD8+), calculating the fraction of cells in each bin, repeating 100 times, and calculating the average. For the CD8+ subset of cells (panels B and D), the T cell lymphoma (Tzone_LSA_LN) is characterized by one dominant clonotype. The melanoma PBMC cases are characterized by a small number of dominant clonotypes with higher frequency. This is similar to the pattern observed for all T cells (Figure 5). In contrast, for the CD4+ subset of cells (panels A and C) a greater diversity of clonotypes was observed for the T cell lymphoma sample and three of four melanoma samples. Melanoma_D_PBMC had similar patterns of CD4 and CD8 expansion.

2) Figure 8 shows UMAP embeddings colored by several covariates for the five samples for which scRNA-seq profiling was performed. There appears to be considerable co-clustering of CD4 and CD8 T cells, which is not surprising. In many cases, this is because memory status rather than CD4 vs. CD8 lineage is a dominant contribution to gene expression for T cell heterogeneity. To clarify this, I would suggest that the authors plot these same UMAP embeddings but color the embeddings by expression of the following markers: CD4, CD8, CCL5, and SELL. I'm not sure if there are straightforward orthologs of CCL5 (*RANTES*) and SELL (*CD62-L* or *L-selectin*) in dogs, but in humans, CCL5 transcript is a very highly expressed marker of effector memory T cells (likely enriched among CD8s in this setting), whereas SELL transcript markers naïve or central memory T cells.

This is an insightful comment. We have completed an analysis of memory status. First, we identified a set of potential markers of effector memory and naïve status from a recently published canine peripheral blood TCR $\alpha\beta$ T cell atlas (Eschke et al 2023). This paper suggested five markers of effector memory status (CCL5, GZMB, ZEB2, GZMK, CCL4) and four markers of naïve status (LEF1, TCF7, CCR7, SELL). Of these, we dropped GZMB and CCL4 because we could not identify clear gene annotations in the canine reference we were using (**Suppl Table 6**). We did not use SELL because it was nearly uniformly expressed in all cells and cell types and therefore not useful for discriminating between specific subsets of T cells (**Response to Reviewer Figure 1**). Eschke et al also showed low specificity of this marker for dogs in their T cell atlas. Using the remaining 6 markers (3 effector memory and 3 naïve) we plotted their expression on the existing t-SNE as suggested (**Suppl Figure 14**). It was difficult to visualize the expression of these markers for just the CD8 T cells co-clustering with CD4 T cells (as defined by singleR cell typing). Therefore we created an additional set of plots focused on expression of the same markers, but just for those cells (**Suppl Figure 15**). As shown in the new **Suppl Figure 15**, co-clustering of CD4 and CD8 does indeed seem to be influenced by naïve vs effector memory status.

Response to Reviewer Figure 1.

These results have been described in the paper with the following text and figures.

We next sought to characterize the expanded and non-expanded (CD4/CD8) cell populations by memory status. We visualized the same t-SNE projections with CD4, CD8, and six markers of effector memory (CCL5, ZEB2, GZMK) or naïve status (LEF1, TCF7, CCR7) (**Suppl Figure 14**). Individual CD4 and CD8 marker expression was largely concordant with singleR cell typing. For the melanoma samples, the expanded (dominant clone) CD8+ population mainly expressed effector memory markers, whereas the non-expanded population predominantly expressed naïve markers. In contrast, for the T zone lymphoma sample, the expanded population mainly expressed naïve markers. This hints that the malignant CD8+ T cells expanded but never matured.

We next attempted to determine why a subset of non-expanded CD8+ cells seemed to co-cluster with CD4+ cells (**Figure 7B**). **Suppl Figure 15** shows that the subset of CD8+ T-cells co-clustering with the CD4+ population primarily expressed naive markers (LEF1, TCF7, CCR7). A portion of this CD8+ subset was also CD4+. Hence, co-clustering of CD4+ and CD8+ populations can be explained by at least two independent phenomena. First, it marks the existence of double positive (CD4+/CD8+) T-cells. Second, it demonstrates CD4+ and CD8+ populations with shared naive status drives clustering more than CD4 vs. CD8 lineage.

Supl Figure 14. Expression of CD4, CD8, markers of effector memory T-cells, and markers of naive T-cells in scTCR dataset.

Expression of CD4, CD8, markers of effector memory T-cells (CCL5, ZEB2, GZMK) and markers of naive T-cells (LEF1, TCF7, CCR7) were projected onto t-SNE maps (See Figure 7 for details) for scTCR samples of dogs with melanoma or T zone lymphoma. Overall, for melanoma samples, the expanded CD8+ population (defined by dominant cluster in Figure 7D) mainly expressed effector memory markers, whereas the non-expanded, predominantly CD4+ population mainly expressed naive markers. For the T zone lymphoma sample, the expanded CD8+ population (presumed malignant clone) predominantly expressed naive markers. The small non-expanded, CD4+ cluster mainly expressed naive markers but also some effector memory.

Suppl Figure 15. Expression of CD4, CD8, markers of effector memory T-cells, and markers of naive T-cells in a subset of CD8+ T-cells that co-clusters with CD4+ T-cells

Expression of CD4, CD8, markers of effector memory T-cells (CCL5, ZEB2, GZMK) and markers of naive T-cells (LEF1, TCF7, CCR7) for a subset of CD8+ T-cells that co-cluster with CD4+ T-cells (defined by singleR cell typing) were projected onto t-SNE maps (See **Figure 7** for details) for scTCR samples of dogs with melanoma or T zone lymphoma. As expected, a subset of cells classified as CD8+ by singleR did not display CD8 expression. A portion of this CD8+ subset was also CD4+. The subset of CD8+ T-cells co-clustering with the CD4+ population primarily expressed naive markers and not effector memory markers.

REVIEWERS' COMMENTS:

Reviewer #5 (Remarks to the Author):

The authors have done an excellent job of responding to my previous comments. I have no further concerns.